# Landslides as geological hotspots of CO₂ to the atmosphere: clues from the instrumented Séchilienne landslide, Western European Alps.

Pierre Nevers[1], Julien Bouchez[2], Jérome Gaillardet[2, 4], Christophe Thomazo[3], Delphine Charpentier[1], Laeticia Faure[2] and Catherine Bertrand[1]

[1]UMR Chrono-Environnement 16 route de Gray, 25000 Besançon, France
[2]Université de Paris, Institut de physique du globe de Paris, CNRS, F-75005 Paris, France
[3]UMR CNRS/uB6282 Biogéosciences 6 Boulevard Gabriel, 21000 Dijon, France
[4]Institut Universitaire de France, 75231 Paris, France

*Correspondence to*: Pierre Nevers (*pierre.nevers@univ-fcomte.fr*)

**Abstract.** This study makes use of a highly instrumented active landslide observatory (9 years of data) in the French Alps, the Séchilienne slope. Here we use a combination of major element chemistry and isotopes ratios ($^{87}Sr/^{86}Sr$, $\delta^{34}S$) measured in different water types of the stable and unstable part of the Séchilienne instability to assess the contribution of the different lithologies of the slope and the chemical weathering mechanisms. Chemical and isotopic ratios appear useful to characterize weathering processes and the origin of waters and their flowpaths through the massif. A mixing model allows us to allocate the different major elements to different sources, to identify secondary carbonate formation as a major process affecting solutes in the subsurface waters of the instability, and to quantify the involvement sulfuric and carbonic acids as a source of protons. As a consequence of the model, we are able to show that the instability creates favorable and sustained conditions within the failure, through the opening of new fractures bringing fresh and reactive surfaces allowing for the production of sulfuric acid by pyrite oxidation. We clearly identify the contribution of each mineral phase dissolution in the chemistry of the waters. The contribution of remote gypsum dissolution to the sulfate budget in the waters is evidenced. We are also able to refine the pre-existing hydrogeological views on the local water circulation and water flow paths in the instability by showing the hydrological connectivity of the different zones. Overall, our results show that the Séchilienne landslide, despite its role in accelerating rock chemical and physical weathering, acts, at a geological time scale (i.e. at timescales longer that carbonate precipitation in the ocean) as a source of CO₂ to the atmosphere. If generalizable to other large landslide complexes in mountain ranges, this study illustrates the complex coupling between physical and chemical erosion and their impact on the carbon cycle and global climate. The study also highlights the importance of deciphering between sulfite oxidation and gypsum dissolution as a source of sulfate ions to rivers, particularly in mountain ranges.

## 1 Introduction

The weathering of rocks plays a key role in the chemical and climatic evolution of the Earth surface and is one of the geological processes that impacts atmospheric $CO_2$ concentration. When carbonic acid is the proton supplier, silicate weathering removes carbon dioxide from the atmosphere (Lerman et al., 2007, Berner and Berner, 2012). However, the oxidative dissolution of sulfides (*e.g.* pyrite $FeS_2$) produces sulfuric acid that can act as an alternative proton supplier to chemical weathering reactions. Although not directly influencing atmospheric $CO_2$, silicate weathering by sulfuric acid does reduce the potential of rock weathering for $CO_2$ sequestration by "removing" silicates from the Earth surface and thus limiting their weathering by carbonic acid. When sulfuric acid reacts with carbonate minerals, dissolved inorganic carbon is added to ambient waters which leads on the long term to $CO_2$ release towards the atmosphere (Lerman et al., 2007, Calmels et al., 2007, S.-L. Li et al., 2008, Torres et al., 2014). The relevance of this process for the global carbon cycle is two-fold. First, even though carbonate rocks do not constitute the major fraction of the rock types exposed at the Earth surface, the dissolved products of carbonate dissolution dominate global weathering fluxes (Gaillardet et al., 1999) as carbonate minerals dissolve several orders of magnitude faster than silicates (Lasaga 1984). Second, because weathering by sulfuric acid is limited by the supply of sulfide minerals to the Earth surface, it is particularly prominent in active mountain belts characterized by high erosion rates (Calmels et al., 2007; Torres et al., 2016; Blattmann et al., 2019). Within these tectonically active environments, landslides are likely to be hotspots of sulfuric acid production, carbonate weathering, and $CO_2$ release (Emberson et al., 2015, 2018). Indeed, slope instability leads to sustained grain comminution and fractures opening, thereby providing a continuous supply of contact surfaces between water, air, and minerals that can in particular allow for sulfuric acid production and carbonate mineral weathering (Binet et al., 2009; Bertrand et al., 2014).

Here we explore the hypothesis that slope instability can constitute a mechanism promoting coupled sulfide oxidation and carbonate weathering, in a contribution to the study of the role active mountain ranges play on the global carbon cycle (Raymo and Ruddiman 1992). We focus on the Séchilienne slope instability located in the French Alps. This site of active, slow landsliding serves as an observatory for landslide processes and has been the subject of previous hydrogeological and geophysical investigation (Vengeon, 1998; Guglielmi et al., 2002; Meric et al., 2005; LeRoux et al., 2011; Vallet et al., 2015; Lajaunie et al., 2019). The Séchilienne site offers the opportunity to study the role of erosion on sulfide oxidation and carbonate weathering under climatic conditions that differ from those of previous studies (such as Taiwan, the Himalayas, or the Andes), and thus to improve the knowledge on the global impact of landsliding on atmospheric $CO_2$ and climate. We combine measurements of the concentration of major elements and of the isotope composition of strontium and sulfur ($^{87}Sr/^{86}Sr$, $\delta^{34}S$) dissolved in groundwater and springs to estimate the contribution of different rock types to the dissolved species produced by weathering reactions in the landslide. In particular, we estimate the relative role of different acid types (carbonic *vs.* sulfuric) and of two rock types (silicates *vs.* carbonates), and evaluate the role played by secondary carbonate formation on the solute budget of percolating waters. Besides shedding light on the global impact of landsliding on atmospheric $CO_2$, Sr and S isotopes coupled to water chemistry allow for a quantitative analysis of the sources of solutes in natural waters and of the chemical

evolution of natural waters, which in turn opens the possibility to improve existing hydrogeological model in complex environments such as landslides.

**2 Study area**

**2.1 Geological setting**

The "Séchilienne" site hosts a highly-instrumented, continuously-monitored landslide, part of the French National Landslide Observatory (OMIV, http://www.ano-omiv.cnrs.fr/). The Séchilienne massif is located at the SW border of the Paleozoic crystalline Belledonne mountain range in the French Alps, 20 km southeast of Grenoble (Isère, France; Fig. 1). The active
zone of the site is a gravitational instability affecting $60.10^6$ m$^3$ of material, with a maximum depth of about 150 m, located on a south-facing slope of the massif (Le Roux et al. 2011). The most active part of the landslide, referred to as "les Ruines", is located on the eastern border of the unstable zone. Long-term monitoring (extensometers, geodetic measurements, tacheometers and microwave radar) shows that the displacement velocity is around 300 cm y$^{-1}$, while the less active parts of the site are moving at a mean of 10 cm y$^{-1}$ (Le Roux et al. 2011; Dubois et al. 2014).

Geological and structural information are provided by the geological map and by two boreholes drilled in 2010 in the unstable area at depths down to 150 m. Baudement et al. (2013) has integrated these information in a GOCAD® 3D model, recently used by Lajaunie et al. (2019) to propose a new vision of the Séchilienne slope based on a 3D resistivity model. The basement of the massif is mainly composed of micaschists showing a N-S trending sub-vertical foliation. Stratigraphically discordant deposits dating from the Carboniferous to the Liassic periods cover the micaschists on the top north – northeast of the massif
(Mont Sec) and along the Sabot Fault (Fig. 1). The slope is locally covered by Quaternary (Würm) glacio-fluvial deposits made of material reworked from the surrounding formations (Vengeon 1998, Vallet 2014). The micaschists consist primarily of quartz, biotite, phengite, and chlorite with occurrence of carbonate veins and pyrite in fractures. Carboniferous deposits are made of black shales, sandstones, and conglomerates with quartz and serpentine pebbles. Triassic rocks correspond to sandstone, quartzite, dolomite and locally of black shales, argilites, and gypsum. Liassic deposits are limestones with
intercalation of layers rich in breccia consisting of micaschist, dolomite, and coal (Barféty et al. 1972, Vengeon 1998, Vallet 2014). Strong local heterogeneities exist in terms of lithology and fracturation and are induced by the gravitational deformation (Lajaunie et al. 2019).

The part of the slope affected by the landslide extends from 400 m to 1100 m above sea level (a.s.l.; Le Roux et al. 2011; Fig. 1d). Above the elevation of 1100 m a.s.l. the morphology of the Mont Sec corresponds to a plateau of glacial origin underlain
by moraine deposits concentrated in small topographic depressions. The landslide is delimited at its northern border by a major head scarp of about 10 meters high and several hundreds of meters wide, which separates the glacial plateau of Mont Sec from the unstable zone. Eastward, N-S faults scarps limit the landslide whereas the western and southern parts are not well defined by geomorphological evidences. The motion of the landslide consists in a deeply-rooted, toppling movement with N50-70 slabs toward the valley, coupled with the sagging of the upper zone of the slope near the Mont Sec (Vengeon 1998). The

Séchilienne instability is assumed to originate from the decompression of the basement rocks after the Romanche glacier

retreated at the last glaciation (15 kyr ago). Decompression caused the opening of fractures and then the collapse of the summit

of the Mont Sec (Montjuvent and Winistorfer 1980, Vengeon et al. 1999, Potherat and Alfonsi, 2001,). The Séchilienne slope

is thus affected by a dense network of near-vertical, open fractures trending N70 and N110/120, controlling the deformation

of the Séchilienne landslide which is characterized by a deep progressive deformation (about 100-150 m) and the absence of

a well-defined basal sliding surface. Two N20 major fractures are also crossing the Séchilienne massif, the Sabot and the

Séchilienne faults. Open fractures are locally filled with detrital material resulting from the erosion of the massif (Vallet et al.

2015a).

Borehole logs available within the instability (Lajaunie et al. 2019) show that the rock formations below the slope are relatively

unstructured, and that pyrite is heterogeneously distributed therein. Rock samples along the boreholes seem to have been

subjected to oxidizing conditions, albeit with no clear sulfide reaction front at the scale of the instability. In addition,

petrological observations on thin sections from these boreholes, combined with mineralogical analyses obtained from X-ray

diffraction (XRD; Supplementary Material) show that pyrite is disseminated within the rocks, with no particular association

with calcite. Gypsum was not detected from XRD analyses in the sampled rocks, consistent with results from inverse modeling

by Vallet et al. (2015) suggesting that sulphates in waters from the unstable zone (UZ) originate essentially from pyrite

oxidative weathering.

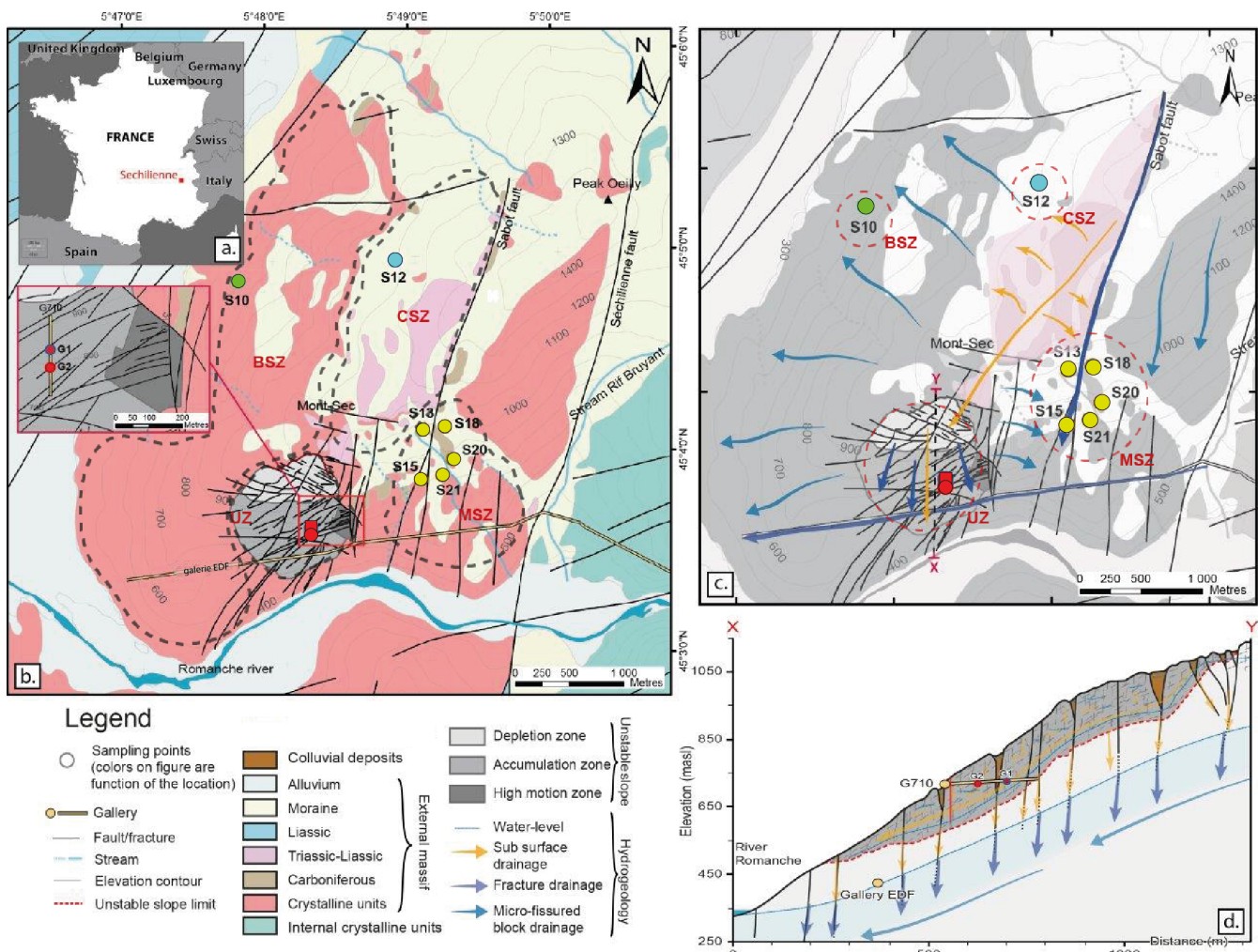

Figure 1: Map of the Séchilienne site a. Location of the Séchilienne massif in the French Alps, b. Simplified geological map of the Séchilienne massif and sampling locations, c. schematic hydrogeological model of the Séchilienne massif, d. hydrogeological cross-section of the instability, modified after Vallet et al. (2015).

## 2.2 Hydrogeological setting

The high degree of fracturation and heterogeneity of the Séchilienne massif leads to distinct and complex patterns of hydrological flow paths. At Séchilienne, water pathways are characterized by different transit times related to a dual permeability behavior that is typical of fractured rock aquifers where conductive fractures play a major role in the drainage (Fig 1.c): rapid transit of infiltration waters through fractures reflects the functioning of a so-called "reactive" hydrological component, whereas slower transit of water through the micro-fissured, less permeable rock matrix, resulting in a smeared response of flow rate to rainfall typifies an "inertial" circulation (Maréchal 1998, Cappa et al. 2004, Vallet et al. 2015a). In

particular, the Sabot and Séchilienne faults play an important role on fluid flow through the massif by draining waters from the sedimentary cover at a fast rate (0.7 km day$^{-1}$), and bypassing the less pervious and more inertial micro-fissured matrix characterized by lower flow velocity (0.08 km day$^{-1}$) (Mudry et Etievant 2007; Vallet et al. 2015a). Local perched aquifers develop during high-flow periods and discharge downwards to the main aquifer, due to the contrast of permeability between the decompressed zone at the surface and the unaltered rock (Lajaunie et al., 2019). An underground tunnel for the production of electricity in a local hydropower plant, named "Galerie EDF", built by Electricité de France (EDF) and located at the base of the slope, acts as a major westward drain for groundwater (Vallet et al. 2015a).

Difference in hydraulic conductivity between the highly fractured unstable zone (thickness about 150-200 m, Le Roux et al. 2011) and the basement situated under the landslide (Fig 1.d) led to the build-up of a two-layer aquifer system. Those two layers are connected to one another through major fractures (Vengeon 1998, Meric et al. 2005; Le Roux et al. 2011, Guglielmi et al. 2002, Vallet et al. 2015a). A temporary and discontinuous shallow perched aquifer is present in the landslide with extension and connectivity varying according to short-term recharge variations. This aquifer is almost dry during the low flow periods, with numerous disconnected saturated pockets (such as open fractures filled by colluvial deposit and altered material) linked to the heterogeneity of the landslide (Guglielmi 2002, Cappa et al. 2004, Vallet et al. 2015a). The recharge of this aquifer is mainly local (through trenches and counterscarps, limiting the runoff) with a contribution of remote groundwater through near-surface drainage at high-flow periods from the sedimentary cover above the landslide (near the Mont Sec summit) (Guglielmi 2002, Vallet et al. 2015a). The deep aquifer, which extends all over the massif (altitude around 550 m asl), corresponds to a saturated layer hosted by the fractured metamorphic bedrock and to an overlying, 100-m thick vadose layer (Vallet et al. 2015a). The deep aquifer level is controlled by the constant water heads of the Romanche alluvium in the valley, and of the Galerie EDF (425 m a.s.l.).

## 3. Samples and analytical methods

Nine outflows draining the whole massif were investigated for physico-chemical parameters and dissolved load chemistry (Fig 1.a). Two of these outflows (G1 and G2) are located within the Unstable Zone (UZ) and correspond to seep water collected in a tunnel excavated to monitor the landslide at 710 m a.s.l. (G710). The remaining outflows (S10, S12, S13, S15, S18, S20, S21) correspond to springs draining the Stable Zone (Fig 1.a). These outflows can be differentiated regarding the dominant local lithology: S10 is located in the Bedrock Stable Zone (BSZ), S12 in the Carbonate Stable Zone (CSZ), and S13, S15, S18, S20, and S21 are located in an area of the Stable Zone characterized by mixed lithology (Mixed Stable Zone, MSZ). Samples were collected every three months over the period 2010-2019. Between 2014 and 2017, waters were sampled once a year for Sr isotopes, while samples for S isotope were collected in 2019. In total, 360 water samples were collected and analyzed for this study. Four local rocks samples were taken, reflecting the main lithological types encountered at Séchilienne: basement micaschist, carbonate (both calcite-rich and dolomite-rich) from the sedimentary cover, and a recrystallized vein in micaschist.

Field measurements of water temperature, pH, and electrical conductivity (EC) were made with a WTW pH/Cond 340i (Xylem Inc.) sensor, with a precision of 0.1 unit and 0.1 µS cm$^{-1}$ for pH and EC, respectively. Water samples were collected in polyethylene bottles and filtered with a 0.45 µm pore diameter nylon filter, before being preserved in cold conditions for measurements of major element concentration and Sr and S isotopes. Analyses of dissolved major elements were all carried out at the research laboratory Chrono-Environnement at the University of Franche-Comté. Dissolved major cation concentrations were measured by atomic absorption spectrometry (AA 100 Perkin-Elmer) with detection limits of 0.5, 0.1, 0.01, and 0.1 mg L$^{-1}$ for Ca$^{2+}$, Mg$^{2+}$, Na$^+$ and K$^+$, respectively. Dissolved anion concentrations were determined using high-pressure ion chromatography (Dionex DX 100) with detection limits of 0.1, 0.1, and 0.05 mg L$^{-1}$ for Cl$^-$, SO$_4^{2-}$ and NO$_3^-$, respectively. The concentration of HCO$_3^-$ was measured by acid titration (N/50 H$_2$SO$_4$) within 48 hours after sampling, with 1% accuracy. Dissolved silica concentration was analyzed with a spectrophotometer (Spectroquant, Pharo 300, Merck) using a silica-test kit (Merck) with 3% accuracy. Only analyses with a charge balance better than 10% were taken into account.

Strontium isotope analyses were carried out at the High-Resolution Analytical Platform (PARI) of the Institut de Physique du Globe de Paris (IPGP). For water samples, dissolved Sr was first isolated from the water sample matrix by automated ion chromatography following the method of Meynadier et al. (2006). For rocks, in addition to bulk samples analysis after digestion in concentrated HF and HNO$_3$. a three-step sequential leaches were conducted using H$_2$O, 1M acetic acid, and 1M HCl. The first step is designed to recover the exchangeable fraction adsorbed onto the solid surface; step 2 was for extracting Sr from carbonates, amorphous hydroxides, and phosphate minerals (Tessier et al., 1979); step 3 was to dissolve any high-order Fe–Mn oxide/oxyhydroxide phases that might be present after HCl leaching (Tessier et al., 1979). The leachate solutions and residual samples were measured for major and trace elements by Quadrupole ICP-MS (Agilent 7900) with a precision better than 5% and processed for $^{87}$Sr/$^{86}$Sr ratio analysis following the same procedure used for bulk samples. To that effect, 3M HNO$_3$ aliquots of digestion solutions were loaded on columns loaded with 0.2 mL of Sr-SPEC resin (Eichrom). Then, 3M HNO$_3$ was used to elute the sample matrix before Sr was eluted in H$_2$O. Strontium isotope ratios were then measured using a Multi-Collector Inductively Coupled Plasma Mass Spectrometer (MC-ICP-MS; Thermo-Fisher Neptune) in low resolution mode (Hajj et al., 2017). Purified Sr solutions were introduced using an APEX desolvation unit and a PFA nebulizer at a rate of 50 to 100 µL min$^{-1}$ depending on the measurement session, and at Sr concentrations between 50 and 150 ppb. The accuracy and reproducibility of the $^{87}$Sr/$^{86}$Sr analysis was assessed using repeated measurements of the international isotope Sr carbonate standard (SRM987, NIST). The obtained values for SRM987 standard NIST was 0.710249 ± 0.000025.

Sulfur isotopes measurements were performed at the Biogéosciences Laboratory, University of Bourgogne, Dijon, France on both sulfates from water samples (Table A1) and sulfides from basement micaschist (Table B2). Nine samples were treated with an excess of 250 g l$^{-1}$ BaCl$_2$ solution to precipitate BaSO$_4$. After centrifugation, the BaSO$_4$ precipitate was washed several times with deionized distilled water and dried at 60°C for 24 hours in an oven. Five hundred micrograms of purified barite samples were poured into tin capsules and homogeneously mixed with 1/3 of vanadium pentoxide before isotopic measurements ($^{34}$S, $^{32}$S) using a Vario PYRO cube (Elementar GmbH) connected online *via* an open-split device to an IsoPrime IRMS system (Isoprime, Manchester, UK). Sulfur isotope data are expressed in delta notation and reported in units per mille

(‰). The $\delta^{34}S$ data are reported with respect to the international standard Vienna Cañon Diablo Troilite (VCDT). Analytical errors are ±0.3‰ (1σ) based on replicate analyses of the international barite standard NBS-127, which was used for data correction assuming a $\delta^{34}S$ value of +20.3 on the VCDT scale

Sulfur contained in sulfides was extracted from eight rock samples of the basement micaschist formation on aliquots of 3 grams (4 sub-samples on each of two rock samples, including unaltered pyrite and iron oxides; Table B2) following the method

described by Canfield et al. (1986). Dried and rinsed $Ag_2S$ precipitates recovered after wet chemistry sulfides extraction are weighted for gravimetric quantification of sample sulfur content. Five hundred micrograms of silver sulfides precipitates were then mixed with an equivalent weight of tungsten trioxide in tin capsules before combustion in a Vario pyro cube (Elementar GmbH$^{TM}$). Sulfur isotope compositions ($\delta^{34}S$) were measured using an IsoPrime IRMS device (Isoprime, Manchester, UK). International standards (IAEA-S-1, IAEA-S-2, IAEA-S-3) were used for calibration and results are reported in the δ-notation

relative to the Vienna Canyon Diablo Troilite (VCDT) standard. Reproducibility (1σ) is better than 0.2‰ based on duplicate analyses of standard materials and samples.

## 4. Results

The concentration of major and trace elements, as well as Sr and S isotope composition are given in Table A1 and A2. The concentration of major elements, EC, pH, temperature, and Sr and S isotope composition of water samples are given in Table

B1.

### 4.1 Major elements

*Rock samples*

The two limestone samples show distinctive response to the leaching procedure, with most of the Ca of the "Laffrey" limestone

located in the acetic acid leachate (67%) and the rest in the HCl fraction (11%), which is indicative of the calcitic nature of this rock sample (Table A1). By contrast, most of the Ca of the "Lias" sample is hosted in the residue, while a significant fraction (37%) is HCl-soluble, suggesting that the sample is dolomitic. The "micaschist" and "vein" samples have higher bulk Sr/Ca, Mg/Ca, Al/Ca and Na/Ca ratios than the two limestone samples, confirming that they are mostly made of silicate minerals.

*Spring water samples*

The chemical composition of waters sampled on the Séchilienne site is very diverse, reflecting heterogeneity in rock types and the existence of various groundwater flow paths. Water pH at Séchilienne is relatively high and varies from 6.5 to 9.4 with a mean value of 7.9. Electrical conductivity range between 79 μS cm$^{-1}$ and 1114 μS cm$^{-1}$ (Table B1). From the major ion

perspective, samples can be grouped into 4 main water types (Fig. 2). These water types correspond to those identified

previously by Vallet et al. (2015a). Type 1 corresponds to Ca-HCO$_3$ waters, typical of water draining carbonate formations, typified by the S12 spring draining the carbonate cover at the top of the Séchilienne slope (CSZ). S12 has low EC values ranging from 79 µS cm$^{-1}$ to 147 µS cm$^{-1}$ with a mean of 117 µS cm$^{-1}$. The second group corresponds to Mg-Ca-HCO$_3$ rich waters, which have circulated through the sedimentary cover (carbonate and dolomite) and the micaschists bedrock and is represented by the S10 spring (BSZ). All S10 samples have higher electrical conductivities ranging from to 308 µS cm$^{-1}$ to 509 µS cm$^{-1}$ with a mean of 443 µS cm$^{-1}$. Waters sampled in the unstable part of the slope (UZ), include the underground outflows G1 and G2 and show a chemical composition that vary from Mg-Ca-HCO$_3$-SO$_4$ waters to Mg-Ca-SO$_4$ waters and constitute the third hydrogeochemical group. The highest EC values of this study are observed for the G1 outflow with electrical resistivities ranging from 613 µS cm$^{-1}$ to 1114 µS cm$^{-1}$ with a mean value of 824 µS cm$^{-1}$. The other outflow of the unstable zone (outflow G2), in contrast to the previous one, shows a mean electrical conductivity value around 391 µS cm$^{-1}$, with a minimum value of 313 µS cm$^{-1}$ (and a maximum value of 470 µS cm$^{-1}$). The fourth and last type of waters include the S13, S15, S18, S20, S21 outflows, sampled in the stable part of the slope (MSZ) along the Sabot fault, and show Ca-Mg-HCO$_3$-SO$_4$ type waters. The MSZ group exhibits EC values ranging from 357 µS cm$^{-1}$ to 567 µS cm$^{-1}$, with a mean of 479 µS cm$^{-1}$. Waters of the Unstable Zone group (G1,G2) are characterized by the highest concentrations in SO$_4^{2-}$ (from 1.32 to 3.90 mmol L$^{-1}$) compared to the other outflows sampled which have values ranging from 0.57 to 1.34 mmol L$^{-1}$ for the MSZ outflows (S13, S15, S18, S20, S21) and from 0.48 to 0.67 for the S10 (BSZ) outflow (Fig. 2; Tab. B1). Fig. 2 clearly shows that SO$_4^{2-}$ ions are significantly contributing to the electrical balance of the analyzed waters. Dissolved Cl$^-$ concentrations are lower than 50 µmol L$^{-1}$ in springs S10, S12, S13, S18, but can reach values above 100 µmol L$^{-1}$ in springs S15, S20, and S21. Dissolved NO$_3^-$ concentrations are typically below 20 µmol L-1 in springs G1, G2, and S10, but are higher in springs S12, S13, S18, S15, S20, and S21, with concentrations above 100 µmol L$^{-1}$ observed in the latter (Tab. B1).

Rainwater samples (Tab. B1) show very low EC values with a mean of 26 µS cm$^{-1}$ and low concentrations for all elements analyzed. In particular, chloride concentrations range from 3.3 µmol L$^{-1}$ to 20.3 µmol L$^{-1}$.

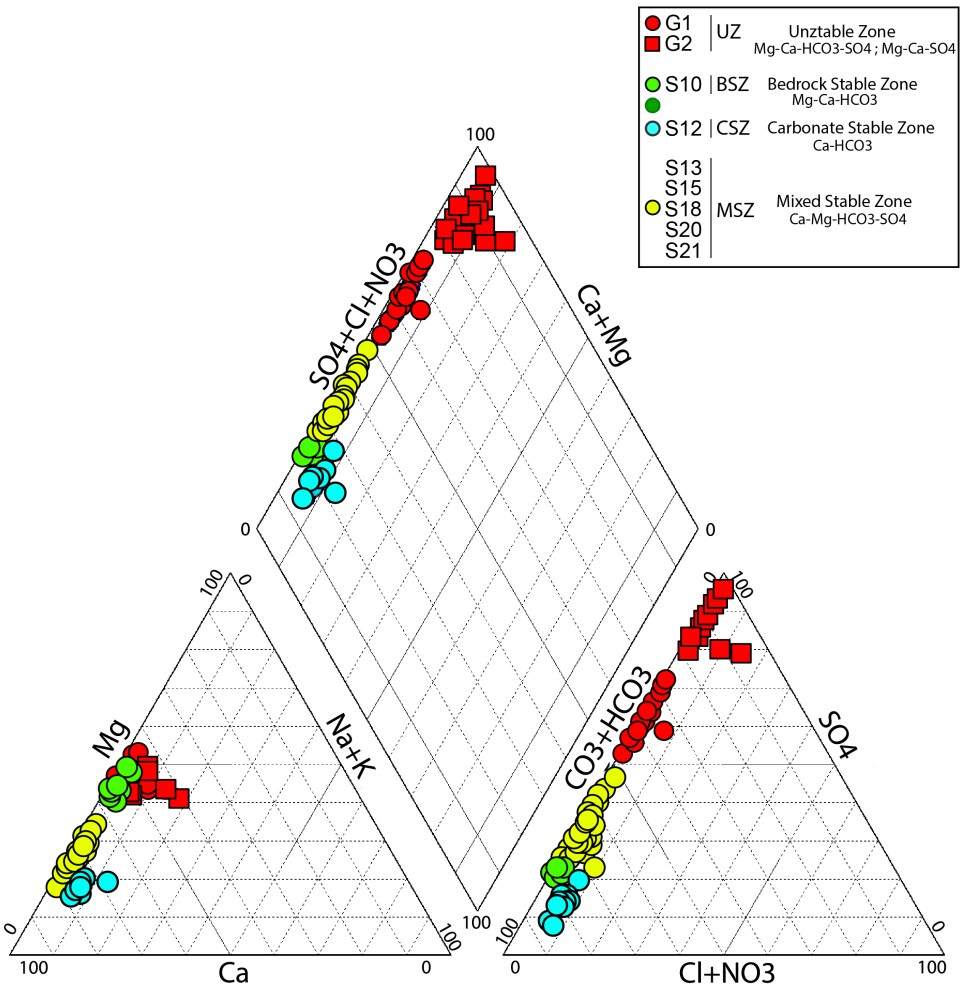

Figure 2: Piper diagram representing the major water types samples in the outflows of the Séchilienne massif

## 4.2 Strontium isotopes

*Rock samples*

The two carbonate rock samples have the lowest Sr isotopic ratios (Table A1), with the lowest value being 0.7095 slightly
higher than the Sr isotopic composition of lower Jurassic seawater (from 0.7065 to 0.7076; Koepnick et al., 1990). The acetic-acid and HCl⁻ soluble fractions of the limestone samples, as well as the bulk analysis of the dolomitic "Lias" sample and its $H_2O$ leachate, are characterized $^{87}Sr/^{86}Sr$ ratios of $\sim$ 0.7105. The $^{87}Sr/^{86}Sr$ ratio of all the leachates of the calcitic "Laffrey" sample, as well as of the bulk sample, show a wider range of variation, in the range 0.7104-0.7179. By contrast, the highest

$^{87}$Sr/$^{86}$Sr ratios were found in the micaschists samples with a value of 0.7351 (Table A1), typical of silicate rocks (0.73±0.01;

Négrel et al., 1993). The veins contained in the micaschists also show high Sr isotopic ratios (0.7277). Such high $^{87}$Sr/$^{86}$Sr ratios are particularly encountered in the residues and bulk samples.

*Spring water samples*

The Sr isotope ratios measured in the spring samples in and around the Séchilienne instability range from 0.7093 to 0.7231 (Table B1). The four main groups of waters have contrasted $^{87}$Sr/$^{86}$Sr isotopic ratios. The highest $^{87}$Sr/$^{86}$Sr values are found in the UZ underground outflows samples (G1, G2) with an average of 0.7210 ± 0.0006 (one standard deviation). The lowest $^{87}$Sr/$^{86}$Sr values correspond to the waters of the MSZ (springs S13, S15, S18, S20, and S21), and average at 0.7095 ± 0.00012, *i.e.* at the value measured in the carbonate rock. The S12 outflow (CSZ) is characterized by $^{87}$Sr/$^{86}$Sr ratios of around 0.7095 ± 0.00005, *i.e.* close to those of the MSZ group. Intermediate values of Sr isotopic ratios are found for the samples of the BSZ group (S10), with an average of 0.7148 ± 0.00019.

## 4.3 Sulfur isotopes

*Rock samples*

Sulfur isotope composition of unaltered rock samples range from -7.9‰ to 17.8‰ with an average value of 1.23‰ ± 11.82‰, whereas weathered micaschists exhibit $\delta^{34}$S values ranging from -13.1 to 9.9‰ and an average of -1.42‰ ± 9.55‰; Table A2). These numbers show the extremely large range of possible sulfur isotope signals co-existing in the various rock types present in the landslide.

*Spring water samples*

Waters show a much narrower range of $\delta^{34}$S values, ranging from -5.5‰ to 6.5‰ (mean 0.43‰ ± 5.12‰). The highest $\delta^{34}$S values are observed for water of the MSZ group (outflows S13, S15, S20, S21) with an average of 6.28‰ ± 0.34‰. Samples of the S12 outflow (CZS group) also exhibits a high $\delta^{34}$S value of 6.03‰. The G1 and G2 outflows (UZ group) exhibit negative $\delta^{34}$S values with an average of -3.74 ‰ ± 1.75‰), with lower $\delta^{34}$S values for G2, (averaging at -5.33‰ ± 0.22‰) than for G1 (2.2‰ ± 0.06‰). The BSZ group (outflow S10) is characterized by $\delta^{34}$S values that are intermediate between those of the MSZ and UZ groups (2.4‰).

## 5. Discussion

### 5.1 Identification of sources to dissolved species

#### 5.1.1 Atmospheric and anthropogenic sources

Rainwater is potentially a significant source of elements to the water sampled in the different springs at Séchilienne. To assess the importance of atmospheric inputs, we use $Cl^-$ concentrations. Chloride is not significantly involved in chemical reactions at the Earth surface and its presence in waters has three main origins: rainwater (through the dissolution of seasalt aerosols), dissolution of saline rocks or inclusions, and anthropogenic inputs. On the other hand, at Séchilienne, $NO_3^-$ is most likely to be derived from human activity, through fertilizer input and/or domestic waste. Therefore, the correlation between $Cl^-$ and $NO_3^-$ concentrations in springs S15, S20, and S21 ($R^2 = 0.65$; Fig. C1) suggests that beyond rainwater, anthropogenic inputs are a significant $Cl^-$ source to these springs. This inference is consistent with the presence of villages upslope from these springs. High $Cl^-$ concentrations ($> 100$ µmol $L^{-1}$) are also found in some samples from underground outflows G1 and G2 (Table B1). However, in the case of these two springs such high $Cl^-$ concentrations are not accompanied with high $NO_3^-$ concentrations but correlate to some extent with dissolved $Na^+$ and $K^+$ concentrations (Fig. C1). This observation could be indicative of the dissolution of salts (NaCl and KCl) as a significant process delivering $Cl^-$ and cations to the springs. Although the origin of these salts is unclear, we note that Zn-Pb ore deposits are reported discovered in the bedrock of Séchilienne landslide exploitation (Barnes, 1997). The presence of fluid inclusions containing alkali elements and Cl in such ores is likely, and leaching of such fluid inclusions could have occurred because of the exposure of new mineral surface during ore. To summarize, the excess of Cl in Séchilienne springs is most probably due to a combination of human activity (road salts and agriculture) and to dissolution of fluid inclusions of hydrothermal origin.

The expected concentration of $Cl^-$ derived from precipitation in spring waters (hereafter called $[Cl]_{crit}$ for "critical chloride", Stallard et al., 1983) of the Séchilienne massif can be estimated by multiplying the mean $Cl^-$ concentration found in rainwater by the mean evapotranspiration factor (P/ETP: 4.02 with P: precipitation, ETP: evapotranspiration calculated from temperature and latitude of the study site, according to Oudin et al. 2005). Alternatively, $[Cl]_{crit}$ can be estimated as being equal to the lowest $Cl^-$ concentrations in the sample set (S10, S12 and S21 springs). Both methods concur to fix the atmospheric contribution of $Cl^-$ to the Séchilienne waters at a maximum of 30 µmol $l^{-1}$. Above this concentration, additional sources must be involved. Once $[Cl]_{crit}$ is known, it is possible to correct all cation concentrations from the atmospheric seasalt input by using:

$$[X]^* = [X] - [Cl]_{crit} \times \left(\frac{X}{Cl}\right)_{seawater} \qquad (1)$$

In this equation, $[X]^*$ denotes the concentration of an element X in the water sample, corrected from the atmospheric input, and $(X/Cl)_{seawater}$ is the seawater elemental ratio. This correction is only significant for $Na^+$, due to the relatively high concentrations found in the Séchilienne waters.

As explained above, significant excess of $Cl^-$ ($[Cl] > [Cl]_{crit}$) is found for the S15, S20, and S21 springs where this excess is the highest (about 30 to 60 µmol l$^{-1}$) and due to domestic and/or agricultural activity; and in the G1 and G2 underground

outflows (between 15-20 µmol l$^{-1}$ of excess), where this excess can be attributed to salt dissolution. For the first group of springs, it is most likely that input of $Cl^-$ is associated with input of $K^+$, through the use of fertilizers. For the second group of springs each mole of $Cl^-$ released by salt dissolution can be associated with one mole of $K^+$ or one mole of $Na^+$. Because of the challenge associated to assessing the exact cause for the observed $Cl^-$ excess in these springs, in the quantitative source apportionment (section 5.1.4) we use a stochastic approach to reflect the uncertainty linked to the nature of the cations delivered

to the springs by these $Cl^-$ sources. However, we emphasize that for the springs where silicate weathering is the most prominent process in terms of cation production (G1 and G2; see section 5.1.2 below), the correction for the solute sources causing the $Cl^-$ excess (human activity and salt dissolution) is relatively negligible.

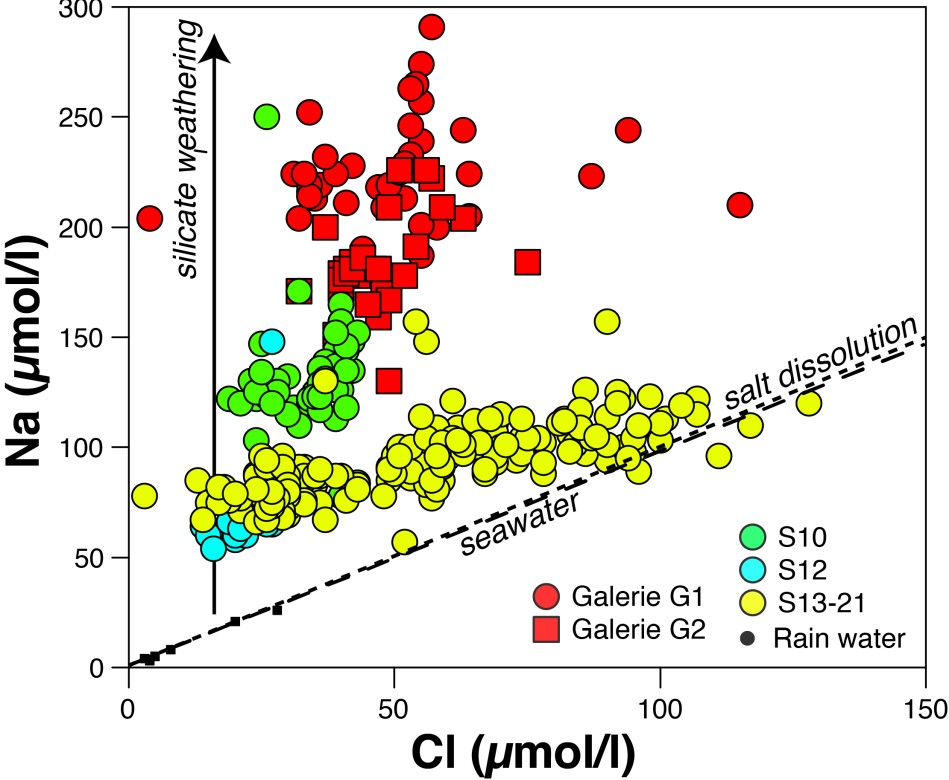

Figure 3: Na vs Cl concentrations measured in the different groups of water outflow from the Séchilienne massif. Rainwater data points are aligned along the seawater composition (seasalts). Na* is defined as the difference Na-Cl$_{crit}$ were Cl$_{crit}$ = 30 µmol/l (see text) where Cl$_{crit}$ is the concentration of chloride only derived from rainwater and concentrated by evapotranspiration.

### 5.1.2 The importance of silicate weathering

The concentration of Na$^+$, once corrected from atmospheric and anthropogenic inputs, can be used as a proxy of silicate
weathering reactions, if the dissolution of silicate minerals is a dominant source of Na to surface waters compared to salt
dissolution and human activity. The underground outflows G1 and G2 have the highest Na* concentrations (mean values
200±30 µmol L$^{-1}$ and 158±21 µmol L$^{-1}$ for G1 and G2, respectively) (Fig. 3). This observation suggests the importance of
silicate weathering reactions in the Unstable Zone, made of fractured micaschists. Although as explained above the source of
excess chloride could also be contributing Na (were this additional source NaCl inclusions), the Na* concentrations remain
the highest found in the Séchilienne landslide area assuming that all Cl$^-$ release to waters is associated to an equivalent Na$^+$
release (in moles). Despite their excess of Cl$^-$, the MSZ outflows chemistry also reveals that silicate weathering reactions are
releasing Na$^+$ to those waters (Fig. 3). However, the Na* concentrations of the Mixed Stable Zone (60 µmol L$^{-1}$ on average)
are equal to around half of those encountered in the Unstable Zone. This contrast between the stable (MSZ) and unstable zone
(UZ) illustrates that the importance of silicate weathering is linked to the fracturation degree at Séchilienne. Finally, the low
Na* concentration in the S12 outflow (Fig. 3) can be attributed to the fact that it mainly drains the carbonate cover. The most
plausible explanation for the non-zero Na* concentration in S12 is the release of Na from silicate material disseminated in the
carbonate rocks.

### 5.1.3 Identifying sources to solutes in the springs of the Séchilienne massif

In the following we use dissolved elemental and isotopic ratios to quantitatively constrain the contribution of various rock
sources (silicates, carbonates, and gypsum) for solutes in the Séchilienne springs. As shown above, strong contrasts exist in
$^{87}$Sr/$^{86}$Sr ratios between the sedimentary carbonate cover and the crystalline rocks of the basement. The isotopic ratio of
dissolved Sr released by water-rock interaction reflects that of the minerals undergoing dissolution and is not affected by the
reincorporation of Sr in secondary minerals (*e.g.* Negrel et al. 1993). Sr isotopes can thus be used to trace the provenance of
dissolved Sr, and by extension of the different cations, in the waters of Séchilienne. By contrast, elemental ratios such as Ca/Sr
and Mg/Sr may be affected by the precipitation of secondary minerals and in particular by the formation of secondary
carbonates (Bickle et al., 2015) and should be used more carefully to identify the provenance of cations.

A series of plots using $^{87}$Sr/$^{86}$Sr as a common Y-axis are shown in Fig. 4. In $^{87}$Sr/$^{86}$Sr *vs*. Ca/Sr or Na/Sr plots (Fig. 4a and 4b)
conservative mixing between reservoirs is indicated by straight lines joining the end members. In this preliminary analysis, the
Ca/Sr ratios for the carbonate, silicate, and evaporite end members are taken from Négrel et al. (1993) and Gaillardet et al.
(1997). The corresponding Sr isotopic ratios are those measured in the rock samples from the Séchilienne massif (Table B1).
The position of the data points corresponding to springs S13, S15, S18, S20, S21 (MSZ group) in Fig. 4a and 4b shows that
their relatively low Sr isotopic composition cannot only be derived from the dissolution of carbonates. Another unradiogenic
end member with low Ca/Sr and Na/Sr ratios needs to be invoked. As indicated by Fig. 4c this end member is enriched in
sulfate as shown by its high SO$_4$/Na ratio. Although gypsum outcrops are not visible at Séchilienne, gypsum is known to exist

in the local Triassic formations present in the upper part of the slope as indicated by the regional geological map (Fig.1). More generally, the presence of gypsum is well documented in the Triassic strata of the the "external Alps" where it plays a major role in large-scale deformation and thrusting (Barféty et al. 1972). The occurrence of gypsum and carbonate dissolution inferred from the chemistry of the Séchilienne springs indicates that the Sabot Fault, which lies at the North-East of the MSZ outflows (Barféty et al. 1972), plays a major role in draining aquifers hosted by sedimentary rocks to the MSZ and BSZ outflows. The

Sr and S isotope composition of Triassic seawater (between 0.7075 and 0.708 and 15±3 ‰, respectively; Burke et al. 1982, Fanlo and Aroya 1998, Kampschutte and Strauss 2004) and the typical Ca/Sr ratio of waters draining gypsum (Gaillardet et al., 1997; Meybeck et al., 1986) are entirely consistent with the contribution of gypsum dissolution. Fig. 4 also shows first that the S12 spring, reported by Vallet et al. (2015a) to be supplied by rapid flowpaths through the sedimentary cover, in addition to being solute-poor compared to springs of the MSZ group, is not influenced by gypsum dissolution despite its geographical

position on the sedimentary part of the slope. The relatively high Na/Sr ratios observed in the S12 spring is probably due to anthropogenic influence, as revealed by the high nitrate concentrations measured in this spring (section 5.1.1 and Fig. C1c). By contrast, $^{87}Sr/^{86}Sr$ and chemical ratios of waters from the UZ outflows (G1, G2) are clearly influenced by a silicate end member. However, their Sr isotopic signature is lower than those of the local micaschist, indicating the additional contribution of Sr from a carbonate and / or evaporitic source to the G1 and G2 spring (Fig. 4a). The $^{87}Sr/^{86}Sr$ ratios of waters of the S10

outflow also exhibit intermediate values between the silicate and carbonate-gypsum mixing line, but with $^{87}Sr/^{86}Sr$ ratio lower than those of the UZ outflows, supporting the idea that water-silicate interaction in the BSZ were less intense than in the UZ. These inferences based on Sr isotopes are in full agreement with those made above based on Na* concentrations and can be interpreted as reflecting the lesser degree of fracturation of the stable zone compared to the unstable zone. Fig. 4c and Fig. 4d show that the higher $^{87}Sr/^{86}Sr$ ratios observed in springs of the UZ (G1, G2), and to a lesser extent of the BSZ (S10) are

associated with sulfate enrichment. However, unlike for samples of the MSZ (S13-S21), dissolved sulfate in UZ and BSZ samples has a relatively low S isotope composition (Fig. 4d). This observation is compatible with a significant influence of sulfide oxidation, despite the very wide range of $\delta^{34}S$ values measured in the bedrock micaschists (between -13,14‰ and 17,77‰, average -0,10‰, S.D: 10,05; Table B2). The presence of pyrite has been reported in the unstable zone of Séchilienne (Bertrand et al. 2014, Vallet et al. 2015a). The concomitant increase in $SO_4^{2-}$ and radiogenic Sr (Fig. 4c), combined with the

decrease in $\delta^{34}S$ (Fig. 4d) suggests a coupling in the unstable zone between sulfide oxidation and silicate weathering. Indeed, the oxidative weathering of pyrite, possibly by $O_2$ or water, leads to the release of sulfate to waters (e.g. Spence and Telmer, 2005):

$$2FeS_2 + \frac{15}{2}O_2 + 7H_2O \rightarrow 2Fe(OH)_3 + 4H_2SO_4 \tag{2}$$

$$FeS_2 + 14Fe^{3+} + 8H_2O \rightarrow 15Fe^{2+} + 2SO_4^{2-} + 16H^+ \tag{3}$$

The oxidation of sulfide to intermediate sulfur species or to sulfate appears to produce only small isotope effects (Fry et al., 1986, 1988; Zerkle et al., 2009; Balci et al., 2012). The significance of these reactions in the unstable zone of Séchilienne can be related to the role of fracturation and grain comminution in favoring the contact between water, air, and minerals, which is

the rate-limiting factor for a fast chemical reaction such as pyrite oxidation.

Altogether, our analyses shows that the composition of the water outflows from the Séchilienne site can be interpreted by a variable contribution of waters having interacted with micaschists and sedimentary rocks, and by a dual origin (sulfide oxidation and gypsum dissolution) of sulfate ions. The chemical and isotopic characteristics of the MSZ and UZ waters show that they have percolated through the sedimentary cover before reaching their outlet in the massif through the Sabot fault.


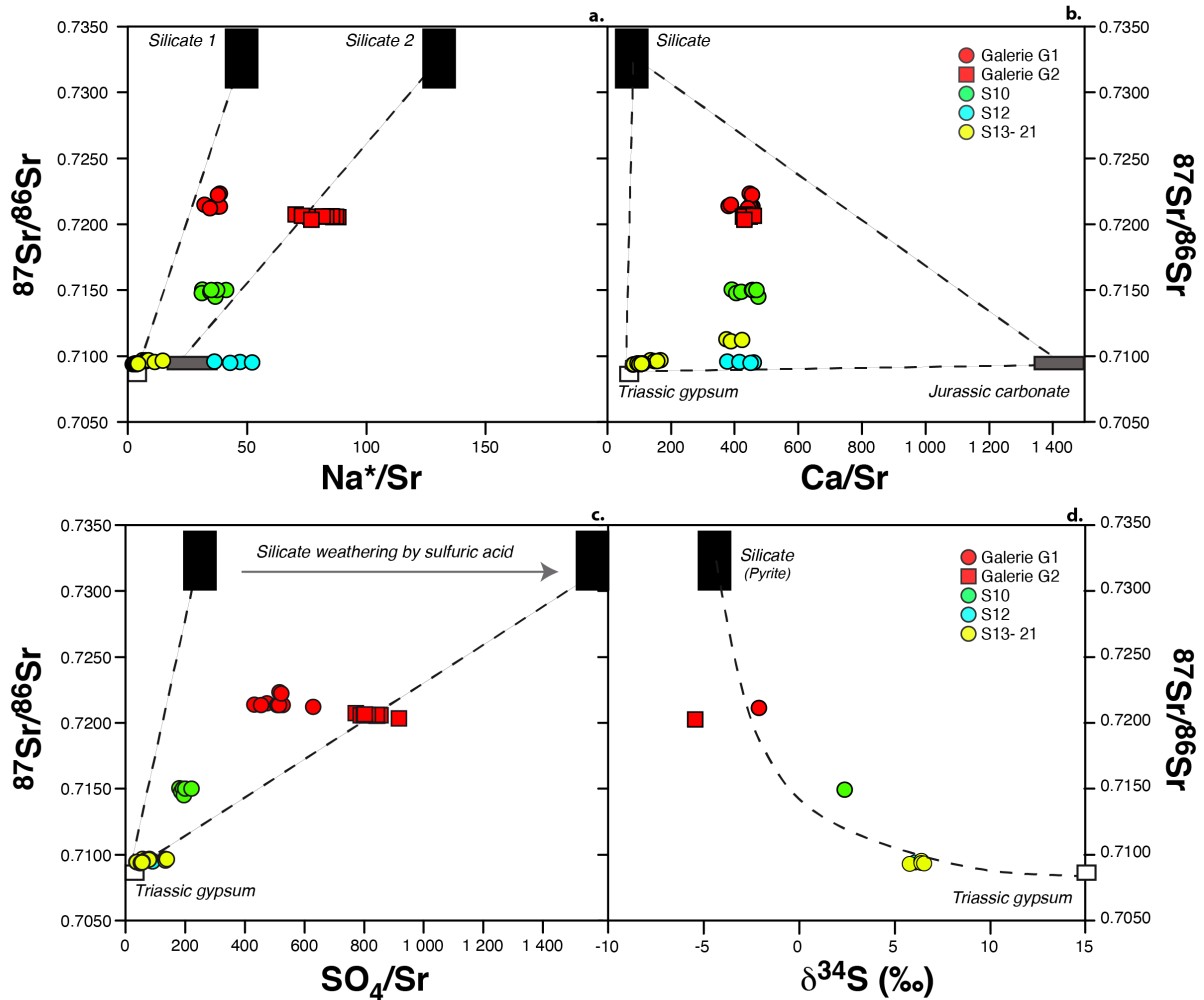

Figure 4: Sr isotopic composition of the different groups of water outflows from the Séchilienne massif as a function of Sr-normalized ratios (a, b and c) and S isotopic composition (d). Mixing end members are discussed in the text (Table C1). Straight lines indicate a mixing process

in Fig. 4a, 4b and 4c. In Fig. 4c, the dashed line is a mixing hyperbola calculated based on the composition of the end members ($^{87}Sr/^{86}Sr$, $\delta^{34}S$, and $SO_4/Sr$ ratios of the silicate and carbonate end members).

### 5.1.4 Quantitative apportionment

Spring dissolved Sr isotopes and major element chemistry make it possible to estimate the relative contribution of each identified end member to the different cations. The details of these calculations are given in Appendix D. Following the above discussion, mixing equations can be written for the conservative trace element Sr:

$$\left(\frac{^{87}Sr}{^{86}Sr}\right)_{mix} = X_{sil}^{Sr} \cdot \left(\frac{^{87}Sr}{^{86}Sr}\right)_{sil} + X_{carb}^{Sr} \cdot \left(\frac{^{87}Sr}{^{86}Sr}\right)_{carb} + X_{gyps}^{Sr} \cdot \left(\frac{^{87}Sr}{^{86}Sr}\right)_{gyps} \tag{4}$$

$$\left(\frac{Na}{Sr}\right)_{mix} = X_{sil}^{Sr} \cdot \left(\frac{Na}{Sr}\right)_{sil} + X_{carb}^{Sr} \cdot \left(\frac{Na}{Sr}\right)_{carb} + X_{gyps}^{Sr} \cdot \left(\frac{Na}{Sr}\right)_{gyps} \tag{5}$$

$$1 = X_{sil}^{Sr} + X_{carb}^{Sr} + X_{gyps}^{Sr} \tag{6}$$

Where the subscripts *mix*, *sil*, *carb*, and *gyps* denote the mixture (water), the silicate, carbonate, and gypsum end members respectively. Proportions of Sr derived from each of those end members $i$ are denoted $X_i^{Sr}$. All ratios are corrected from atmospheric and salt inputs according to the above method. Because the carbonate and gypsum end member add relatively few Na compared to Na* (Na from silicates), equation 5 simplifies into:

$$\left(\frac{Na}{Sr}\right)_{mix} = X_{sil}^{Sr} \cdot \left(\frac{Na}{Sr}\right)_{sil} \tag{7}$$

This assumption is supported by the positions of the different springs in Fig. 4a, which indicates that the low-$^{87}Sr/^{86}Sr$ component of the springs - encompassing both carbonate and gypsum weathering - has a negligible Na content. The proportions of Sr in the different mixing reservoirs can then be estimated and the contribution of each of these end members to the load of the dissolved major species $SO_4^{2-}$ and $Mg^{2+}$ then calculated following:

$$X_i^E = X_i^{Sr} \left(\frac{E}{Sr}\right)_i / \left(\frac{E}{Sr}\right)_{spring} \tag{8}$$

with $i = sil$, *carb*, or *gyp*, and $E = SO_4$, or Mg (corrected from rain inputs). Full discussion is given in Appendix D on the choice of the $(Na/Sr)_{sil}$ (eq. 7) and more generally of the $(E/Sr)_i$ ratios (eq. 8), based on regression of the spring hydrochemical data and independent constraints from our geochemical analyses of the rock samples.

In carbonate-rich contexts like that of Séchilienne, dissolved $Ca^{2+}$ concentrations can be affected by precipitation of secondary carbonates which tend to scavenge significant amounts of dissolved Ca relative to Mg and Sr (Bickle et al., 2015). For this reason, in principle eq. 8 cannot be applied for $E$ = Ca and $i$ = carb. The relatively high Mg/Ca ratios (around 0.9 mol/mol in springs G1, G2, and S10; and 0.2-0.4 mol/mol in springs S10 to S21; Table B1) and Sr/Ca ratios (around 2 mmol/mol in springs G1, G2, S10, S12, and S18; and above 6 mmol/mol for springs S13, S15, S20, and S21) compared to the estimated Mg/Ca and Sr/Ca ratios of the calcite end member at Séchilienne (below 0.1 mol/mol and 1 mmol/mol, respectively) determined from our geochemical analyses of rock samples (Table A1) are suggestive of a significant role of secondary carbonate formation. We quantify the role of secondary carbonate formation using the method proposed by Bickle et al. (2015), which is based on the comparison (in the Sr-Ca-Mg-Na compositional space) of the measurements in springs and the composition predicted from conservative mixing between the rock end members (Appendix D). In this analysis we contend that secondary carbonate formation affected waters containing solutes derived from the three rock end members identified for the Séchilienne springs (silicates, carbonates, and gypsum). We estimate that along the water flowpath secondary carbonate formation scavenges around 60% of the Ca initially released to solution by the combined dissolution of silicates, carbonates, and gypsum for springs G1, G2, and S10, whereas the effect of secondary carbonate precipitation is negligible for the other springs (Appendix D). These results highlight the potential role of lithological diversity, a characteristic of the bedrock material drained by springs G1, G2, and S10 in the UZ and BSZ (compared to other springs mostly influenced by the carbonate cover), in promoting secondary carbonate formation through mixing of compositionally different waters.

Another challenge in using eq. 8 at Séchilienne is the fact that both calcite and dolomite are reported to occur as carbonate minerals at Séchilienne - as confirmed by our own chemical analyses of rock samples (Tab. A1). Therefore, we take into account the presence of dolomite in our quantitative source apportionment, in particular regarding the $(Mg/Sr)_{carb}$ ratio used in eq. (8) (Appendix D). Based on arguments linked to the extent of secondary carbonate precipitation needed to explain the spring data (see above), we estimate that the contribution of dolomite dissolution to the overall Ca released by carbonate weathering at Séchilienne is about 10 to 20% (Appendix D).

In order to quantify the uncertainty associated to our mixing model, a Monte Carlo approach was used with 10,000 simulations. Results are given in Tab. D2 and represented in Fig. 5 as a stacked bar plot. In the following text, results on mixing proportions $X_i^E$ are reported as $D_{50}{}_{-D_{16}}^{+D_{84}}$ ($D_n$ is the $n^{th}$ percentile of the output distribution over the 10,000 simulations; $D_{50}$ is thus the median).

Carbonate dissolution appears as the major contributor to dissolved Ca, and to about half of dissolved Mg in all the springs sampled on the different parts of the studied zone (UZ, BSZ, MSZ), making this process a major supplier of cationic charges to waters at Séchilienne (Tab. D2; Fig. 5). In the most active part of the landslide (G1, G2), despite the silicate-dominated lithology, carbonate contribution is significant (about 40%), indicating the waters percolating though the unstable zone acquired part of their chemical and isotopic composition from above the hillslope. Calculations of the proportions of sulfate derived from the different end members show a minor but non negligible contribution of gypsum dissolution (reaching 88% of the total sulfate in spring S15) and a very clear contribution of pyrite oxidative weathering particularly important in the

fractured zone. In the G1-G2-S10 group of springs, most of the anionic charge (> 80%) is provided by the oxidative weathering of pyrite. Springs from the BSZ exhibit a lower proportion from silicate end member with a median of $0.48_{0.24}^{0.76}$ against $0.61_{0.38}^{0.84}$, $0.56_{0.31}^{0.81}$ for G1 and G2 (UZ), respectively. This contrast can be attributed to the unstable context of G1 and G2
compared to that of the stable part of the slope at the BSZ outflow.

In waters of the MSZ (S13-S21), significant contribution of gypsum is evidenced in the results of the mixing equations with values ranging between $0.20_{0.15}^{0.30}$ and $0.36_{0.29}^{0.48}$. These values are consistent with the mixing relationship presented in Fig. 4d. Therefore, and although significant uncertainty exists regarding the S isotope composition of the pyrite endmember, S isotope data lend support to our inference from the mixing model that waters are derived essentially from pyrite oxidation in the
unstable zone, and from gypsum dissolution the MSZ waters (S13, S15, S18, S20, S21). The relative contribution of carbonates is also significant in MSZ waters with proportions ranging from $0.32_{0.09}^{0.58}$ for S18 to $0.33_{0.13}^{0.56}$ for S15.

Based on results of the mixing model, we can estimate a value for $\delta^{34}S$ of the pyrite endmember. Indeed, a significant linear negative relationship ($R^2 = 0.8$) exists between the $\delta^{34}S$ measured in springs across the Séchilienne massif and the modal estimates of their $X_{sil}^{SO4}$ (Fig. D5). The intercept of this relationship at $X_{sil}^{SO4} = 1$ (equivalently at $X_{gyp}^{SO4} = 0$) gives an estimate
for $\delta^{34}S_{sulfur}$ of -3.1‰. Such estimates are consistent with the range of measurements of solid sulfur reported in this study (ranging between -13.1‰ to 17.8‰) and reflect to an average value of the S isotope composition of sulfides for the Séchilienne unstable zone.

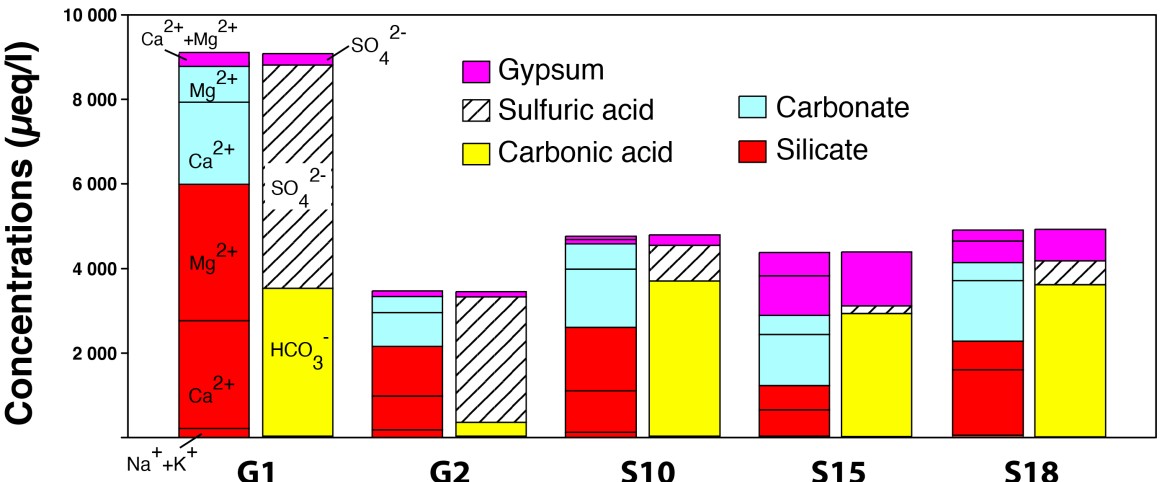

Figure 5: Concentration (in charge equivalents) calculated for major dissolved species and for each end member from the mixing model presented in the text and in the Appendix (Table D2). Silicates contribute for Na+K+Ca+Mg (Na+K not indicated), carbonates for Ca+Mg, gypsum dissolution for Ca+Mg (not distinguished). Note that ultimately the hydrogenocarbonate ion originates from respiration in soils. Sulfuric acid is generated by the oxidation of sulphide minerals, a process that occurs preferentially in the fractured zone. Relative contributions of different end members were obtained by solving a set of mixing equations using a Monte Carlo approach. Note that the
relative contribution of each rock end member to the $Ca^{2+}$ load here refers to that calculated for the Ca "initially" released into solution, that is before secondary carbonate precipitation (Appendix D).

## 5.2 Implications for hydrogeological processes at the Séchilienne site

Water plays an important role in the dynamics of slope instabilities, first as a physical (hydrogeological) process that can lead to aggravation of the instability. Secondly, water is a geochemical agent, which weather rocks and makes them less cohesive (Rutqvist and Stephansson, 2003; Binet, 2006; Cappa et al., 2004). These two categories of processes interact with each other in time as weathering leads to modifications in subsurface permeability and porosity, and thus in water flowpaths through the massif. Hydrological triggering is the most usual mechanism of initiation and reactivation of landslides but water flows in the subsurface have also been shown to have a major impact on the destabilization of a slope (de Montety et al. 2007; Guglielmi et al. 2002, Vallet et al. 2015b). However, landslides constitute very heterogeneous media due to their intense fracturation, which makes hydrogeological investigation complicated. The use of hydrochemistry and isotopic investigation is a good substitute to classical investigation. For example, groundwater dissolved $^{87}Sr/^{86}Sr$ ratios have proven to be useful in determining the sources of solutes in natural waters (Négrel and Deschamps, 1996, Négrel et al., 2001, Dotsika et al., 2010), investigating mineral weathering reactions (Brass, 1975; Åberg et al., 1989; Bullen et al., 1996; Clow and Drever, 1996; Bullen and Kendall, 1998), and identifying mixing processes involving groundwaters of different sources (Woods et al., 2000; Frost and Toner, 2004; Singleton et al., 2006) also inside an unstable context (Deiana et al., 2018). Values of groundwater dissolved $\delta^{34}S\text{-}SO_4^{2-}$ have also been used in aquifer studies to identify sulfate sources (Moncaster et al., 2000; Cortecci et al., 2002; Gammons et al., 2013). In particular, the Séchilienne hydrogeological model proposed by Vallet et al. (2015a) use sulfates as a tracer of waters flowing through the instability with the assumption that all $SO_4^{2-}$ measured in groundwaters is sourced from pyrite oxidation. High sulfates concentrations in MSZ waters were indeed inferred by Vallet et al. (2015a) to be derived from a mixture of 30% of waters from the UZ (drained through the micro-fissured matrix) and 70% from the sedimentary cover (drained through both micro-fissured matrix and larger fractures), thereby establishing a hydraulic connection between the UZ and the MSZ waters (Fig. 1c).

Results from the present study partly support the hydrogeological model established by Vallet et al. (2015a) but allow us to refine this model through the identification of the contribution of another, unexpected end member corresponding to the dissolution of gypsum with a remote origin. Based on the local geological map (Barféty et al. 1972) gyspum occurrence has been reported but outside of the study zone, upstream along the Sabot fault which lies at the North-East of the MSZ outflows. As the fault is a major flowpath (Lajaunie et al., 2019), draining aquifers hosted by the sedimentary cover to the MSZ outflows, it contributes to the enrichment in $SO_4^{2-}$ of those waters (Fig. 6). Our study therefore indicates a significant evaporitic origin for the sulfates in the MSZ waters, challenging the interpretation of Vallet et al. (2015a) of a hydrogeological connection between waters of the unstable and stable zones. Sulfate in outflows draining the UZ and BSZ is not strongly sourced from evaporites, but the part of those sulfates with evaporitic origin can be explained by a contribution of water flows through the Sabot fault towards the sedimentary cover and the basement formations (Fig. 6).

In addition, the systematic differences in elemental concentrations observed between the UZ and BSZ outflows (Tab. A2) can be linked to the structure of the slope and water flowpaths in the subsurface. Indeed, the S10 outflow drains a stable area (BSZ) just above the lowly-weathered, only slightly fractured basement. By contrast, the G1 and G2 outflows drain the unstable part of the slope (UZ), where the basement is highly fractured. This leads in turn stronger weathering degree of rocks and minerals there, and in particular of pyrite which is a major contributor to dissolved sulfate in the G1 and G2 outflows - more so than in the S10 outflow (stable and lowly weathered) characterized by lower sulfate contents.

Improving numerical and predictive models requires the incorporation of hydrological processes such as the dynamics of water circulation within a slope (directly dependent on fracturation, volumes of water involved, etc.). This study shows that isotopic proxies such as Sr and S isotopes ($^{87}Sr/^{86}Sr$, $\delta^{34}S$) coupled to water chemistry can be a very powerful tool to constrain groundwater origin and flowpaths in landslides and can substitute to tracer surveys, and constitute an alternative for hydrogeological investigation in logistically-challenging field environments such as unstable slopes.

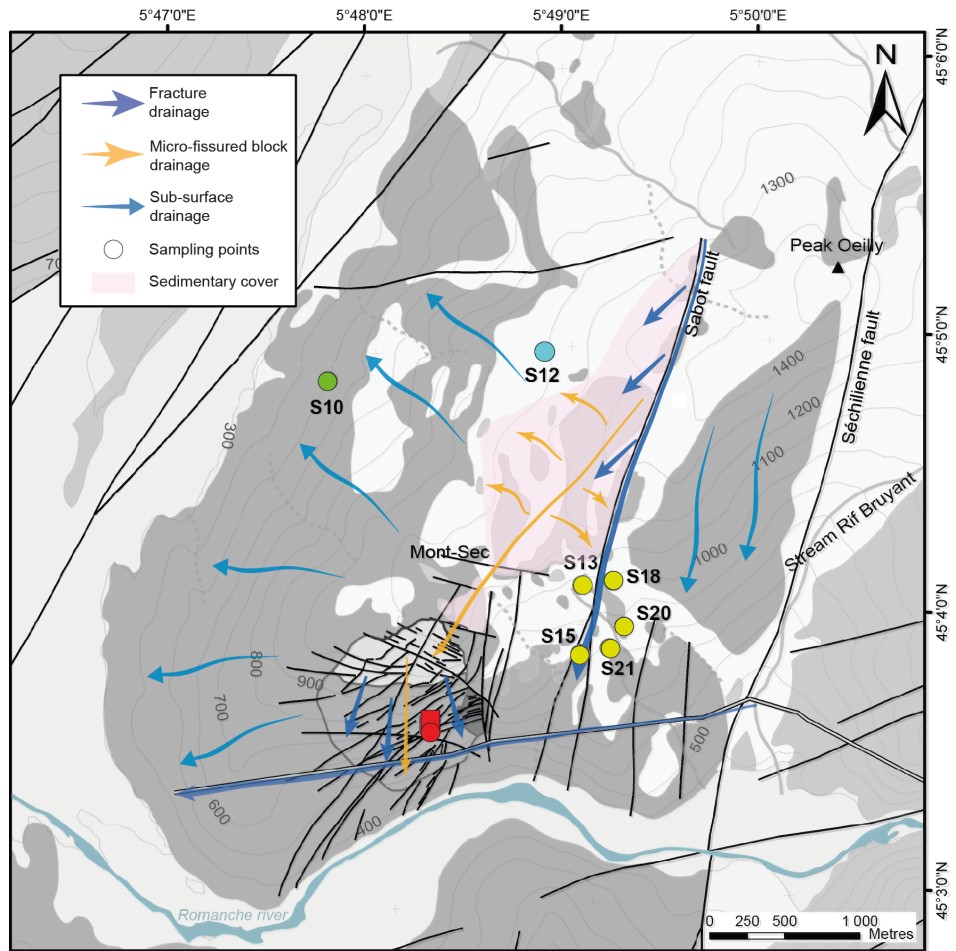

Figure 6: Sketch of the groundwater conceptual model, modified after Vallet et al., (2015).

**5.3 Role of landslides on silicate weathering and CO₂ consumption**

Recent studies have shown the importance of pyrite oxidation, sulfuric acid production, and associated chemical weathering in active landslides (Emberson et al. 2015, Emberson et al. 2016). In the following paragraph we examine the potential implications of the present study of the Séchilienne landslide for the global carbon cycle.

Rock weathering consumes atmospheric $CO_2$ and, associated with the precipitation of carbonates in the ocean, is the mechanism that has allowed for the sequestration of atmospheric $CO_2$ and consequently lower the Earth's surface temperature

on geological timescales (Berner and Berner 2012). Rock forming-minerals uplifted to the Earth surface react with oxygen, carbonic acid produced by soil respiration, and sulfuric acid produced by the oxidation of sulfide minerals. The following reactions describe how carbonic (equations 9 and 11) and sulfuric (equations 10 and 12) acids interact with silicate (here wollastonite $CaSiO_3$) and carbonate minerals, and lead to the production of alkalinity (here $HCO_3^-$):

$$CaSiO_3 + 2CO_2 + H_2O \rightarrow SiO_2 + Ca^{2+} + 2HCO_3^- \tag{9}$$

$$CaSiO_3 + H_2SO_4 \rightarrow SiO_2 + H_2O + Ca^{2+} + SO_4^{2-} \tag{10}$$

$$CaCO_3 + CO_2 + H_2O \rightarrow Ca^{2+} + 2HCO_3^- \tag{11}$$

$$2CaCO_3 + H_2SO_4 \rightarrow 2Ca^{2+} + 2HCO_3^- + SO_4^{2-} \tag{12}$$

It is usually considered that when $Ca^{2+}$ reaches the ocean, over a time period longer than 0.1 to 1 Myr, the precipitation of $CaCO_3$ releases $CO_2$ into the ocean-atmosphere system according to the reaction:

$$Ca^{2+} + 2HCO_3^- \leftrightarrow CaCO_3 + CO_2 + H_2O \tag{13}$$

The influence of the above reactions (equations 9-13) on atmospheric $CO_2$ partial pressure depends on the time scale considered (Torres et al., 2016). At short timescales (typically $< 10^5$ yrs), the chemistry of river discharge is able to influence the carbonate system in the ocean. Indeed, the delivery of alkalinity and Dissolved Inorganic Carbon (DIC) to the ocean to a ratio lower than that of the modern seawater ratio (Alk/DIC ~ 1) leads to increased dissolved $CO_2$ concentration, and consequently higher $CO_2$ content in the atmosphere through re-equilibration (Zeebe and Wolf-Gladrow, 2001). If the Alk/DIC ratio is higher than 1 but

lower than 2, at time scales longer than that typical of carbonate precipitation in the ocean ($10^5$ to $10^6$ yrs) but shorter than that of marine sulfate reduction to sulfide in sea bottom sediments (several $10^6$ yrs), atmospheric $CO_2$ will increase because the precipitation of carbonates releases $CO_2$ to the ocean-atmosphere system that was not consumed on land by weathering reactions (combination of equations. 12 and 13). This mechanism should lead to global warming (Calmels et al., 2007) and has been invoked by Torres et al. (2014) for maintaining atmospheric $CO_2$ levels during the Himalayan orogeny, which

otherwise should have led to a rapid atmospheric $CO_2$ depletion by enhanced silicate weathering. Finally, at timescales longer

than that typical of sedimentary burial of sulfide in the ocean, only silicate weathering by carbonic acid leads to net C sequestration (Berner et Berner, 1996; Calmels et al. 2007).

However, our analysis demonstrates that at Séchilienne reaction (13) also occurs directly on the continent through the formation of secondary carbonates, favored by the addition of calcium and alkalinity derived from silicate weathering, which
results in a "short cut" of the carbon cycle. This short-term $CO_2$ release has to be taken into account when evaluating the overall $CO_2$ effect of weathering reactions at Séchilienne, and more generally in lithologically complex weathering systems where secondary carbonate formation is likely to involve solutes produced by a variety of processes, and in particular by carbonate weathering by sulfuric acid.

To this aim, we use the stoichiometry of reactions (9) to (13), together with the results of our quantitative source apportionment
(section 5.1.4) to calculate the impact of weathering reactions at Séchilienne on atmospheric $CO_2$ over two scales (convoluted spatially and temporally), referred to in the following as "on site" or "local" (*i.e.*, immediately when weathering processes take place in the unstable zone) and "long term" or "global" (i.e., taking into account marine carbonate precipitation in the ocean ensuing solute delivery to the ocean).

Fig. 7 shows that waters produced in the Unstable Zone of Séchilienne (G1 and G2) act as $CO_2$ sources on the "long term",
whereas waters produced in the bedrock stable zone (S10) or the mixed stable zone (S15, S18) are $CO_2$ sinks or $CO_2$-neutral within uncertainty. Our study thus shows that instabilities such as the Séchilienne landslide can act as hotspots of long-term $CO_2$ release to the atmosphere depending on the types of mineral-fluid interactions and also on the flow paths followed by the water drained in the landslide. We suggest that chemical weathering in similar landslides throughout the Alps (i.e Clapière, Super Sauze, and Valabres in the French Alps, Rosone in Italy; Barla et Chiriotti, 1995 ; Follacci, 1999 ; Binet, 2006) have a
similar impact on global biogeochemical cycles and climate. Although it is beyond the scope of the present study to quantify the $CO_2$ fluxes linked to weathering in the Séchilienne landslide - let alone to attempt an extrapolation of such local results to the scale of the Alpine range - our work clearly shows that silicate and carbonate weathering by sulfuric acid generated in landslide zones of active mountain ranges have a climatic impact though a complex set of entangled short-term and long-term effects. Furthermore, this impact contradicts the textbook view that silicate weathering in mountain ranges consumes $CO_2$ from
the atmosphere and cool the global climate (Raymo, 1991), and motivates more detailed studies associating hydrogeological and mineralogical approaches to build a more realistic understanding of the impact of mountains on climate change.

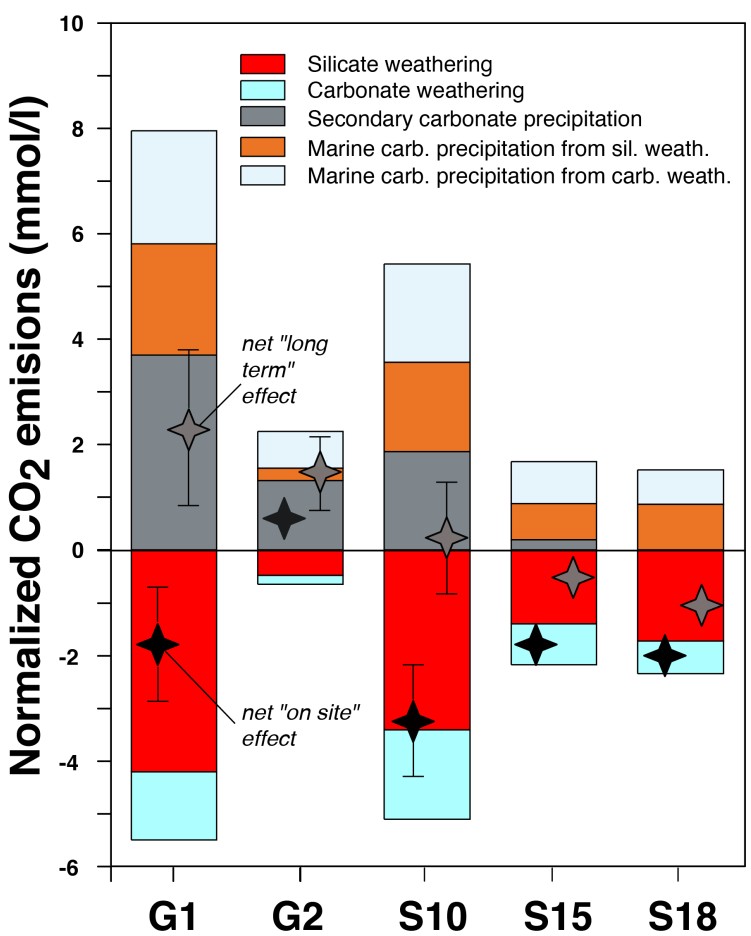

Figure 7: Evaluation of the effect of weathering processes at Séchilienne on atmospheric $CO_2$. The hydrochemistry of the springs G1 and
G2 draining the instability ("Unstable Zone") show that the weathering results in long-term $CO_2$ production to the atmosphere because the
cations are preferentially released in spring waters by the action of sulphuric acid and not carbonic acid from the soils (Table D3). In the
Bedrock Stable Zone (spring S10) and in the Mixed Stable Zone (springs S15 and S18), weathering processes act as $CO_2$ sinks or are $CO_2$-
neutral within uncertainty. Secondary carbonate precipitation returns $CO_2$ to the atmosphere. In particular, in the spring G2 of the Unstable
Zone, this process results in a net "on site" $CO_2$ release because it involves precipitation of Ca and alkalinity derived from carbonate
weathering by sulfuric acid.

## 6. Conclusion

We use measurements of dissolved major element chemistry coupled to Sr and S isotopic ratios in spring waters of the
Séchilienne active landslide site in order to identify the chemical processes at play in the subsurface of the landslide area.
Among these tracers, strontium isotopes allow us to allocate cations to different sources, circumventing issues affecting
elemental ratios related to the precipitation of secondary. Silicate, carbonate, and evaporite weathering all appear to contribute
to the cation load of the Séchilienne waters. Scavenging of dissolved calcium by secondary carbonate formation is identified

as a major process affecting solutes in the subsurface waters of the Séchilienne instability and favored by the mixing of different solution having interacted with a heterogeneous set of minerals. Sulfur isotopes bring a unique qualitative constraint on the

origin of the sulfate ion, which is abundant in the Séchilienne groundwaters, showing the contribution of not only pyrite oxidation but also of gypsum dissolution.

The provenance of dissolved species at Séchilienne also reveals the complex water flow paths there. In particular, waters percolating through the landslide have acquired part of their hydrochemical characteristics far away from the unstable zone itself. For example, sulfur isotopes clearly indicate an unexpected contribution from Triassic sedimentary gypsum dissolution,

that can only occur in the sedimentary layers capping the upper part of the massif and pointing out the importance of water drainage by a major fault of the massif.

The comparison between the stable and unstable parts of the site suggests that silicate weathering is enhanced in the fractured, unstable zone, where the landslide is active. Sulfur isotopes indicate that the production of acidity by the oxidation of magmatic sulfides enhances rock alteration in the unstable zone, this leads us to suggest the following feedback. By favoring the

penetration of oxic waters and allowing contact with silicate minerals, fracturation and grain comminution controls the oxidation of pyrite that in turn rapidly generates sulfuric acid. The weathering of silicate minerals by sulfuric acid weakens the rock structure what in turn favors fracturation in response to the gravitational stress. Fletcher et al. (2006) and recently Behrens et al. (2015) have shown that opening of porosity at the rock-soil interface in soil profiles can be initiated by oxidation of Fe(II) minerals inducing a positive volume budget leading to the production of micro-cracks, inducing further weathering, at the

origin of the opening of fractures, provided that enough carbonate and pyrite is present in the bedrock. At a larger scale, the feedback we propose here exemplifies a similar process of coupling between physical and chemical processes sustaining mass wasting in mountain ranges.

Finally, we demonstrate that the Séchilienne landslide is a hotspot of $CO_2$ release to the atmosphere over the long term. Although it remains difficult to upscale the results of the present study to the entire Alpine range, or to a global scale, landslides

developed on sulfide-hosting sedimentary rocks appear to have a climatic impact opposite to the conventional view that rock weathering in mountains ranges consumes $CO_2$ from the atmosphere, and thus contributes to global cooling. In addition, our study shows a strong control of weathering processes and rates by local hydrogeological features, such as the complexity of flow paths setting the chemistry of the groundwaters within the unstable zone. More work is needed to assess the importance of landslides as hotspots of chemical alteration and geological $CO_2$ emissions, in particular to investigate their hydrological

and hydrochemical response to weather and climate change. More generally, landslides epitomize the coupling between landscape evolution, tectonics and climate-weather. For this reason - as well as that of their societal impact in terms of natural hazard - monitoring landslides over a range of time scales and frequency should become a priority.

# Appendices

## Appendix A: Bedrock analyses

Table A1: Major, trace elements concentration and strontium isotopic compositions of rock samples of the Séchilienne slope

**Concentration in rock samples (in ppm relative to total rock mass)**

| | Li | Be | Na | Mg | Al | K | Ca | Ti | V | Cr | Mn | Fe | Co | Ni | Cu | Zn | As | Rb | Sr |
|---|---|---|---|---|---|---|---|---|---|---|---|---|---|---|---|---|---|---|---|
| **"Lias" limestone** | | | | | | | | | | | | | | | | | | | |
| $H_2O$ leachate | 0,035 | 0,004 | 1,86 | 113 | 0,77 | 12,3 | 61,7 | 0,000 | 0,003 | 0,001 | 0,071 | 1,543 | 0,001 | | 0,002 | | 0,004 | 0,040 | 0,140 |
| Ac. Ac. leachate | 0,392 | 0,113 | 20 | 14242 | 10,3 | 21,1 | 30498 | 0,067 | 0,235 | 0,070 | 147 | 998 | 0,611 | 1,002 | 0,003 | 33,17 | 0,019 | 0,022 | 21,70 |
| HCl leachate | 0,887 | 0,033 | 79 | 45085 | 7,7 | 13,5 | 86109 | | 0,006 | 0,001 | 303 | 5732 | 0,116 | 0,575 | 0,011 | 6,863 | 0,016 | 0,022 | 60,52 |
| Residue | 4,76 | 0,184 | 187 | 51193 | 1085 | 427 | 97029 | 51 | 4,420 | 1,288 | 187 | 5785 | 0,717 | 2,015 | 1,64 | 15,95 | 1,654 | 2,059 | 45,58 |
| Bulk | 9,8 | 0,5 | 501 | 127456 | 1166 | 1329 | 220499 | 55 | 5 | 2,4 | 646,6 | 16782 | 1,9 | 4,8 | 3,28 | 76,5 | 1,92 | 2,67 | 152 |
| Sum of leachate / Bulk | 0,62 | 0,73 | 0,57 | 0,87 | 0,95 | 0,36 | 0,97 | 0,93 | 0,95 | 0,56 | 0,99 | 0,75 | 0,76 | 0,74 | 0,51 | 0,73 | 0,88 | 0,80 | 0,84 |
| **"Laffrey" limestone** | | | | | | | | | | | | | | | | | | | |
| $H_2O$ leachate | 0,066 | 0,012 | | 13,0 | 1,85 | 25,1 | 297 | | 0,007 | 0,004 | 0,109 | 0,745 | 0,002 | | 0,009 | 0,205 | 0,019 | 0,081 | 3,368 |
| Ac. Ac. leachate | 0,189 | 0,242 | 0,383 | 1561 | 38,9 | 58,9 | 240693 | 0,220 | 1,003 | 0,172 | 382 | 1003 | 0,053 | 0,447 | 0,037 | 2,835 | 0,164 | 0,059 | 342,59 |
| HCl leachate | 0,104 | 0,148 | | 1041 | 76,9 | 40,6 | 40863 | 0,119 | 0,803 | 0,126 | 77 | 891 | 0,215 | 1,405 | 0,938 | 5,493 | 0,188 | 0,134 | 61,54 |
| Residue | 3,11 | 0,123 | 150 | 178 | 3263 | 1354 | 877 | 127 | 6,814 | 2,795 | 2,88 | 1710 | 1,634 | 8,214 | 1,895 | 2,316 | 0,977 | 7,240 | 2,853 |
| Bulk | 5,79 | 0,711 | 401 | 3659 | 4522 | 2664 | 344063 | 180 | 10,9 | 5,17 | 553,4 | 4771 | 2,55 | 13,7 | 4,91 | 20,0 | 2,25 | 11 | 532 |
| Sum of leachate / Bulk | 0,60 | 0,74 | 0,38 | 0,76 | 0,75 | 0,55 | 0,82 | 0,71 | 0,79 | 0,60 | 0,84 | 0,76 | 0,75 | 0,74 | 0,59 | 0,54 | 0,60 | 0,70 | 0,77 |
| **Micaschiste** | | | | | | | | | | | | | | | | | | | |
| $H_2O$ leachate | 0,035 | 0,002 | 21 | 1,71 | 7,42 | 64,6 | 0,71 | 0,035 | 0,012 | 0,007 | 0,023 | 4,302 | 0,005 | 0,006 | 0,007 | 0,006 | 0,067 | 0,119 | 0,013 |
| Ac. Ac. leachate | 0,758 | 0,096 | 74 | 273 | 798 | 511 | 474 | 0,228 | 0,838 | 0,795 | 14,9 | 689 | 0,977 | 1,271 | 0,826 | 1,907 | 1,145 | 1,074 | 3,184 |
| HCl leachate | 3,36 | 0,047 | 24 | 654 | 1484 | 199 | 1295 | 3,723 | 1,402 | 2,028 | 12,6 | 2465 | 0,720 | 3,125 | 0,526 | 4,220 | 2,460 | 0,673 | 6,315 |
| Residue | 17,3 | 1,366 | 14459 | 4926 | 57017 | 17831 | 585 | 2791 | 48,62 | 41,37 | 57 | 12151 | 5,303 | 16,440 | 4,636 | 25,346 | 1,706 | 75,619 | 40,181 |
| Bulk | 27,0 | 2,32 | 13436 | 7150 | 70048 | 22490 | 2813 | 3407 | 60 | 64,4 | 94,5 | 18792 | 10,5 | 28,3 | 7,7 | 41,1 | 5,5 | 111 | 59 |
| Sum of leachate / Bulk | 0,79 | 0,65 | 1,08 | 0,82 | 0,85 | 0,83 | 0,84 | 0,82 | 0,84 | 0,69 | 0,90 | 0,81 | 0,67 | 0,74 | 0,78 | 0,77 | 0,97 | 0,70 | 0,84 |
| **Vein** | | | | | | | | | | | | | | | | | | | |
| $H_2O$ leachate | 0,103 | 0,001 | 56 | 4,20 | 0,492 | 13,5 | 17,8 | 0,003 | 0,002 | 0,001 | 0,200 | 1,429 | 0,009 | 0,007 | 0,005 | 0,015 | 0,022 | 0,038 | 0,273 |
| Ac. Ac. leachate | 0,168 | 0,009 | 16,3 | 115 | 63 | 26 | 393 | 0,057 | 0,062 | 0,058 | 14,9 | 220 | 0,898 | 0,468 | 0,153 | 0,690 | 0,293 | 0,093 | 2,681 |
| HCl leachate | 0,258 | 0,004 | 3,57 | 73 | 104 | 23 | 154 | 0,496 | 0,091 | 0,153 | 5,22 | 339 | 0,108 | 0,340 | 0,077 | 0,479 | 0,691 | 0,126 | 1,234 |
| Residue | 5,50 | 0,108 | 1244 | 371 | 4156 | 1223 | 74 | 197 | 3,412 | 3,097 | 6,18 | 966 | 0,340 | 1,090 | 0,387 | 2,044 | 0,157 | 5,520 | 4,015 |
| Bulk | 9,22 | 0,142 | 1497 | 592 | 4631 | 2089 | 648 | 216 | 3,75 | 4,87 | 26,7 | 1704 | 1,66 | 2,57 | 1,94 | 5,51 | 1,41 | 6,50 | 8,93 |
| Sum of leachate / Bulk | 0,65 | 0,85 | 0,88 | 0,95 | 0,93 | 0,62 | 0,99 | 0,91 | 0,95 | 0,68 | 0,99 | 0,90 | 0,82 | 0,74 | 0,32 | 0,59 | 0,83 | 0,89 | 0,92 |

**Concentration in rock samples (in ppm relative to total rock mass)**

| | Y | Zr | Nb | Cs | Ba | La | Ce | Pr | Nd | Sm | Eu | Gd | Tb | Dy | Ho | Er | Tm | Yb | Lu |
|---|---|---|---|---|---|---|---|---|---|---|---|---|---|---|---|---|---|---|---|
| **"Lias" limestone** | | | | | | | | | | | | | | | | | | | |
| H₂O leachate | 0,001 | 0,000 | 0,004 | 0,001 | 0,030 | 0,000 | 0,000 | 0,000 | 0,000 | 0,000 | 0,000 | 0,000 | 0,000 | 0,000 | 0,000 | 0,000 | 0,000 | 0,000 | 0,000 |
| Ac. Ac. leachate | 1,195 | 0,007 | 0,001 | 0,000 | 3,582 | 0,256 | 0,552 | 0,074 | 0,354 | 0,155 | 0,143 | 0,220 | 0,033 | 0,172 | 0,032 | 0,081 | 0,010 | 0,053 | 0,007 |
| HCl leachate | 1,165 | 0,002 | 0,002 | 0,001 | 1,926 | 0,284 | 0,532 | 0,058 | 0,249 | 0,079 | 0,046 | 0,138 | 0,021 | 0,121 | 0,026 | 0,070 | 0,009 | 0,042 | 0,006 |
| Residue | 2,329 | 1,571 | 0,203 | 0,145 | 21,22 | 0,565 | 1,324 | 0,162 | 0,695 | 0,237 | 0,105 | 0,317 | 0,053 | 0,318 | 0,065 | 0,191 | 0,024 | 0,143 | 0,020 |
| Bulk | 5,55 | 2,26 | 0,25 | 0,215 | 32,09 | 1,34 | 3,03 | 0,38 | 1,65 | 0,63 | 0,37 | 0,84 | 0,14 | 0,78 | 0,17 | 0,45 | 0,06 | 0,32 | 0,05 |
| Sum of leachate / Bulk | 0,85 | 0,70 | 0,85 | 0,69 | 0,83 | 0,82 | 0,80 | 0,77 | 0,79 | 0,75 | 0,80 | 0,80 | 0,76 | 0,78 | 0,73 | 0,77 | 0,70 | 0,75 | 0,71 |
| **"Laffrey" limestone** | | | | | | | | | | | | | | | | | | | |
| H₂O leachate | 0,002 | 0,003 | 0,018 | 0,004 | 0,024 | 0,001 | 0,002 | 0,001 | 0,001 | 0,001 | 0,000 | 0,001 | 0,000 | 0,000 | 0,000 | 0,000 | 0,000 | 0,000 | 0,000 |
| Ac. Ac. leachate | 4,463 | 0,103 | 0,004 | 0,002 | 3,243 | 3,435 | 4,811 | 0,774 | 3,329 | 0,691 | 0,416 | 0,711 | 0,101 | 0,557 | 0,117 | 0,333 | 0,044 | 0,280 | 0,042 |
| HCl leachate | 1,108 | 0,092 | 0,006 | 0,030 | 1,478 | 0,704 | 1,084 | 0,190 | 0,851 | 0,184 | 0,100 | 0,191 | 0,027 | 0,152 | 0,029 | 0,085 | 0,012 | 0,074 | 0,010 |
| Residue | 0,248 | 3,223 | 0,464 | 0,490 | 32,50 | 0,477 | 0,703 | 0,074 | 0,227 | 0,019 | 0,016 | 0,026 | 0,004 | 0,030 | 0,007 | 0,027 | 0,004 | 0,028 | 0,005 |
| Bulk | 7,84 | 4,35 | 0,67 | 0,73 | 49,05 | 6,19 | 8,95 | 1,42 | 5,99 | 1,2 | 0,70 | 1,24 | 0,17 | 0,96 | 0,20 | 0,59 | 0,08 | 0,50 | 0,08 |
| Sum of leachate / Bulk | 0,74 | 0,79 | 0,73 | 0,72 | 0,76 | 0,75 | 0,74 | 0,73 | 0,74 | 0,76 | 0,76 | 0,75 | 0,79 | 0,77 | 0,75 | 0,76 | 0,74 | 0,77 | 0,75 |
| **Micaschiste** | | | | | | | | | | | | | | | | | | | |
| H₂O leachate | 0,002 | 0,001 | 0,003 | 0,004 | 0,083 | 0,008 | 0,016 | 0,002 | 0,007 | 0,001 | 0,000 | 0,001 | 0,000 | 0,000 | 0,000 | 0,000 | 0,000 | 0,000 | 0,000 |
| Ac. Ac. leachate | 0,442 | 0,002 | 0,001 | 0,009 | 10,288 | 0,409 | 0,844 | 0,103 | 0,413 | 0,096 | 0,024 | 0,105 | 0,016 | 0,086 | 0,015 | 0,040 | 0,005 | 0,030 | 0,004 |
| HCl leachate | 1,649 | 0,043 | 0,003 | 0,032 | 6,173 | 1,204 | 2,596 | 0,314 | 1,325 | 0,348 | 0,062 | 0,313 | 0,055 | 0,147 | 0,018 | 0,107 | 0,014 | | |
| Residue | 12,881 | 14,56 | 11,173 | 2,476 | 395,42 | 22,44 | 44,752 | 5,237 | 19,85 | 3,43 | 0,686 | 2,978 | 0,402 | 2,083 | 0,404 | 1,156 | 0,155 | 0,920 | 0,120 |
| Bulk | 25,7 | 27,8 | 12,9 | 3,5 | 563 | 28,9 | 59,2 | 7,0 | 26,0 | 5,1 | 1,05 | 4,85 | 0,74 | 4,28 | 0,87 | 2,40 | 0,34 | 2,07 | 0,25 |
| Sum of leachate / Bulk | 0,58 | 0,53 | 0,87 | 0,72 | 0,73 | 0,83 | 0,81 | 0,81 | 0,83 | 0,77 | 0,75 | 0,72 | 0,65 | 0,58 | 0,54 | 0,56 | 0,53 | 0,51 | 0,54 |
| **Vein** | | | | | | | | | | | | | | | | | | | |
| H₂O leachate | 0,001 | 0,000 | 0,001 | 0,007 | 0,018 | 0,001 | 0,001 | 0,000 | 0,001 | 0,000 | 0,000 | 0,000 | 0,000 | 0,000 | 0,000 | 0,000 | 0,000 | 0,000 | 0,000 |
| Ac. Ac. leachate | 0,158 | 0,001 | 0,000 | 0,009 | 1,182 | 0,038 | 0,097 | 0,014 | 0,070 | 0,025 | 0,009 | 0,034 | 0,006 | 0,032 | 0,006 | 0,015 | 0,002 | 0,011 | 0,001 |
| HCl leachate | 0,220 | 0,007 | 0,001 | 0,032 | 0,564 | 0,098 | 0,246 | 0,037 | 0,192 | 0,071 | 0,020 | 0,091 | 0,014 | 0,062 | 0,009 | 0,019 | 0,002 | 0,010 | 0,001 |
| Residue | 1,170 | 1,319 | 0,742 | 0,355 | 28,165 | 1,224 | 2,441 | 0,280 | 1,056 | 0,194 | 0,042 | 0,191 | 0,030 | 0,174 | 0,036 | 0,105 | 0,014 | 0,084 | 0,011 |
| Bulk | 1,06 | 0,66 | 0,73 | 0,47 | 32,2 | 1,49 | 3,09 | 0,386 | 1,51 | 0,33 | 0,08 | 0,32 | 0,05 | 0,22 | 0,04 | 0,09 | 0,01 | 0,07 | 0,008 |
| Sum of leachate / Bulk | 1,47 | 2,02 | 1,02 | 0,85 | 0,93 | 0,91 | 0,90 | 0,86 | 0,87 | 0,88 | 0,89 | 0,98 | 1,07 | 1,21 | 1,32 | 1,54 | 1,45 | 1,53 | 1,61 |

**Concentration in rock samples (in ppm relative to total rock mass)**

| | Pb | Th | U | Contrib. to total Ca (%) | Sr/Ca mmol/mol | Mg/Ca mol/mol | Na/Ca mol/mol | 87Sr/86Sr | s.d. |
|---|---|---|---|---|---|---|---|---|---|
| **"Lias" limestone** | | | | | | | | | |
| H₂O leachate | 0,005 | 0,003 | 0,003 | 0,03 | 1,08 | 3,16 | 0,055 | | |
| Ac. Ac. leachate | 3,469 | 0,057 | 0,558 | 13 | 0,34 | 0,80 | 0,0012 | 0,710395 | 0,000005 |
| HCl leachate | 0,453 | 0,002 | 0,126 | 37 | 0,33 | 0,90 | 0,0017 | 0,709767 | 0,000003 |
| Residue | 7,323 | 0,307 | 2,448 | 42 | 0,22 | 0,91 | 0,0035 | 0,708834 | 0,000004 |
| Bulk | 17,1 | | 4,00 | | 0,31 | 0,95 | 0,0039 | 0,709540 | 0,000014 |
| Sum of leachate / Bulk | 0,66 | | 0,78 | | | | | 0,84 | |
| **"Laffrey" limestone** | | | | | | | | | |
| H₂O leachate | 0,006 | 0,001 | 0,000 | 0,08 | 5,40 | 0,075 | | 0,710629 | 0,000007 |
| Ac. Ac. leachate | 1,493 | 0,439 | 0,012 | 67 | 0,68 | 0,011 | 0,0000029 | 0,710539 | 0,000004 |
| HCl leachate | 2,126 | 0,165 | 0,006 | 11 | 0,72 | 0,04 | | 0,710409 | 0,000008 |
| Residue | 0,709 | 0,247 | 0,084 | 0,24 | 1,55 | 0,35 | 0,31 | 0,717934 | 0,000027 |
| Bulk | 6,0 | 0,48 | 0,10 | | 0,71 | 0,018 | 0,0020 | 0,710573 | 0,000003 |
| Sum of leachate / Bulk | 0,73 | | 1,02 | | | | | 0,77 | |
| **Micaschiste** | | | | | | | | | |
| H₂O leachate | 0,012 | 0,016 | 0,002 | 0,02 | 8,39 | 4,12 | 54,20 | | |
| Ac. Ac. leachate | 0,967 | 1,060 | 0,160 | 16 | 3,20 | 0,99 | 0,28 | 0,728147 | 0,000005 |
| HCl leachate | 1,967 | 7,461 | 0,698 | 44 | 2,32 | 0,87 | 0,034 | 0,720109 | 0,000013 |
| Residue | 4,351 | 10,770 | 1,792 | 20 | 32,7 | 14,5 | 44,9 | 0,735581 | 0,000018 |
| Bulk | 12,1 | 10,3 | 3,4 | | 9,62 | 4,18 | 8,31 | 0,735077 | 0,000014 |
| Sum of leachate / Bulk | 0,60 | | 0,78 | | | | | 0,88 | |
| **Vein** | | | | | | | | | |
| H₂O leachate | 0,003 | 0,001 | 0,000 | 2,6 | 7,31 | 0,41 | 5,75 | | |
| Ac. Ac. leachate | 0,362 | 0,065 | 0,010 | 58 | 3,25 | 0,50 | 0,075 | 0,724583 | 0,000004 |
| HCl leachate | 0,778 | 0,343 | 0,024 | 23 | 3,81 | 0,82 | 0,042 | 0,722884 | 0,000005 |
| Residue | 0,347 | 0,926 | 0,076 | 11 | 25,8 | 8,58 | 30,4 | 0,731125 | 0,000016 |
| Bulk | 1,37 | 0,583 | 0,09 | | 6,29 | 1,50 | 4,02 | 0,727672 | 0,000004 |
| Sum of leachate / Bulk | 1,09 | | 1,30 | | | | | 0,84 | |


Table A2: Sulfate isotopic compositions of bedrock samples of the Séchilienne slope


| | Sample | Sampling dep | $\delta^{34}S$ (‰) | Mean $\delta^{34}S$ | s.d |
|---|---|---|---|---|---|
| *Weathered micaschist* | SC2 19.71 A _ 1 | 19.71 | -7.94 | -0.74 | 9.36 |
| | SC2 19.71 A _ 2 | 19.71 | -7.76 | | |
| | SC2 19.71 B _ 1 | 17.71 | -7.21 | | |
| | SC2 19.71 B _ 2 | 19.71 | -7.01 | | |
| | SC2 108.50 _ 1 | 108.5 | 17.77 | | |
| | SC2 108.50 _ 2 | 108.5 | 17.56 | | |
| | SC2 132.50 _ 1 | 132.5 | 1.67 | | |
| | SC2 132.50 _ 2 | 132.5 | 1.99 | | |
| *Unweathered micaschist* | SC1 84.2 (1) _ 1 | 84.2 | -6.58 | -0.31 | 8.21 |
| | SC1 84.2 (1) _ 2 | 84.2 | -6.56 | | |
| | SC2 128 _ 1 | 128 | -4.12 | | |
| | SC2 128 _ 2 | 128 | -4.28 | | |
| | SC2 106.80 _ 1 | 106.8 | 9.87 | | |
| | SC2 106.80 _ 2 | 106.8 | 10.10 | | |
| | SC1 42.3 (5) _ 1 | 42.3 | -3.32 | | |
| | SC1 42.3 (5) _ 2 | 42.3 | -3.40 | | |
| | SC1 42.3 (5) _ 3 | 42.3 | -3.21 | | |
| | SC1 80.3 (4) | 80.3 | 0.88 | | |
| | SC1 30.60 (2) | 30.6 | -13.13 | | |



# Appendix B: Chemical and isotopic composition of waters from the Séchilienne slope and rainwaters

Table B1: Chemical and isotopic composition of waters samples of the Séchilienne slope

| Outflow | Date | Ca | Mg | Na | Na* | K | F | Cl | NO₃ | SO₄ | HCO₃ | SiO₂ | Sr | Li | EC | T | pH | ⁸⁷Sr/⁸⁶Sr | δ³⁴S |
|---|---|---|---|---|---|---|---|---|---|---|---|---|---|---|---|---|---|---|---|
| | | $\mu mol.L^{-1}$ | $\mu mol.L^{-1}$ | $\mu mol.L^{-1}$ | $\mu mol.L^{-1}$ | $\mu mol.L^{-1}$ | $\mu mol.L^{-1}$ | $\mu mol.L^{-1}$ | $\mu mol.L^{-1}$ | $\mu mol.L^{-1}$ | $\mu mol.L^{-1}$ | $\mu mol.L^{-1}$ | $\mu mol.L^{-1}$ | $\mu mol.L^{-1}$ | uS/cm | °C | | | ‰ |
| G1 | 18/11/2010 | 2495 | 2345 | 187 | 140 | 36 | 11 | 55 | 1.5 | 2729 | 3279 | 114 | | | 613 | 10.6 | 7.6 | | |
| G1 | 03/02/2011 | 2208 | 1954 | 200 | 150 | 30 | 13 | 58 | 0 | 2527 | 3091 | 112 | | | 735 | 10.5 | 8.6 | | |
| G1 | 28/02/2011 | 2358 | 2181 | 257 | 210 | 33 | 12 | 55 | 1.6 | 2723 | 2955 | 115 | | | 797 | 10.06 | 8.2 | | |
| G1 | 01/04/2011 | 2408 | 2057 | 239 | 192 | 30 | 15 | 55 | 0 | 2617 | 3271 | 116 | | | 825 | 11.4 | 8.2 | | |
| G1 | 01/06/2011 | 2508 | 2255 | 274 | 227 | 33 | 16 | 55 | 0 | 2736 | 3391 | 115 | | | 846 | 12.7 | 8.5 | | |
| G1 | 05/10/2011 | 2470 | 2551 | 265 | 219 | 51 | 11 | 54 | 5.3 | 3302 | 3359 | 112 | | | 891 | 11.8 | 8.1 | | |
| G1 | 11/01/2012 | 2745 | 2448 | 96 | 50 | 30 | 12 | 53 | 0 | 3221 | 3039 | 116 | | | 892 | 10.9 | 8.2 | | |
| G1 | 15/02/2012 | 2370 | 2325 | 244 | 189 | 31 | 13 | 63 | 0 | 2940 | 2911 | 111 | | | 837 | 10.6 | 8.42 | | |
| G1 | 27/03/2012 | 2445 | 2530 | 291 | 242 | 35 | 13 | 57 | 5.1 | 3071 | 3299 | 119 | | | 906 | 10.5 | 8.1 | | |
| G1 | 27/06/2012 | 2545 | 2325 | 246 | 201 | 35 | 15 | 53 | 0 | 2768 | 3447 | 119 | | | 843 | 11.2 | 8.0 | | |
| G1 | 25/09/2012 | 2645 | 2325 | 263 | 217 | 36 | 12 | 53 | | 2938 | 3507 | 116 | | | 910 | 11.3 | 7.9 | | |
| G1 | 08/01/2013 | 2196 | 1958 | 233 | 187 | 30 | 11 | 53 | 6.2 | 2266 | 3187 | 118 | | | 750 | 11.1 | 8.2 | | |
| G1 | 16/04/2013 | 2233 | 2016 | 224 | 169 | 42 | 11 | 64 | 39.8 | 2315 | 3583 | 124 | | | 750 | 11.2 | 8 | | |
| G1 | 17/07/2013 | 2174 | 1966 | 213 | 169 | 32 | 7 | 52 | 7.2 | 2224 | 3275 | 120 | | | 719 | 12.4 | 8.0 | | |
| G1 | 17/09/2013 | 2187 | 2230 | 218 | 179 | 31 | 9 | 47 | 3.4 | 2570 | 3123 | 117 | | | 771 | 11 | 8.2 | | |
| G1 | 15/12/2013 | 2237 | 1862 | 201 | 154 | 32 | 11 | 55 | 67.4 | 2584 | 2851 | 121 | | | 806 | 10.6 | 8.2 | | |
| G1 | 09/03/2014 | 1913 | 1647 | 190 | 152 | 25 | 11 | 44 | 2.6 | 1964 | 3471 | 130 | | | 686 | 11.1 | 8.1 | | |
| G1 | 14/06/2014 | 2131 | 2056 | 204 | 201 | 29 | 13 | 4 | | 2414 | 3655 | 136 | 6 | | 762 | 11.4 | 8.2 | 0.7213 | |
| G1 | 09/10/2014 | 2570 | 2313 | 244 | 164 | 85 | 7 | 94 | 23.1 | 2844 | 3518 | 131 | 6 | | 834 | 11.2 | 8.3 | | |
| G1 | 20/04/2015 | 2283 | 2093 | 219 | 177 | 33 | 11 | 49 | 7.1 | 2194 | 3851 | 137 | 5 | | 752 | 11.5 | 7.9 | | |
| G1 | 13/07/2015 | 2157 | 2031 | 210 | 112 | 96 | 8 | 115 | 8.1 | 2755 | 3506 | 127 | 5 | | 795 | 11.9 | 8.1 | | |
| G1 | 15/10/2015 | 2076 | 2139 | 209 | 169 | 31 | 7 | 48 | 8.2 | 2786 | 3063 | 110 | 5 | | 821 | 10.7 | 8 | | |
| G1 | 19/11/2015 | 2446 | 2239 | 229 | 184 | 35 | 11 | 52 | 5.6 | 2986 | 3623 | 115 | 6 | 4.2 | 894 | 11.2 | 7.9 | 0.7214 | |
| G1 | 11/01/2016 | 2365 | 2272 | 228 | 191 | 40 | 12 | 42 | 8.3 | 3907 | 3107 | 265 | 2 | 2.8 | 868 | 10.9 | 7.6 | | |
| G1 | 19/04/2016 | 2243 | 2138 | 211 | 176 | 36 | 9 | 41 | 8.8 | 2986 | 3479 | 121 | 5 | 3.7 | 844 | 11.2 | 7.2 | | |
| G1 | 19/05/2016 | 2210 | 1787 | 204 | 176 | 29 | 12 | 32 | | 2204 | | 125 | 5 | 2.2 | | | | 0.7213 | |
| G1 | 19/05/2016 | 2237 | 1950 | 213 | 184 | 30 | 11 | 35 | | 2543 | | 127 | 5 | 3.4 | 823 | 11 | 7.87 | 0.7213 | |
| G1 | 20/05/2016 | 2286 | 1933 | 219 | 190 | 31 | 11 | 34 | 0.4 | 2610 | | 128 | 5 | 3.4 | 833 | 11.5 | 7.98 | 0.7213 | |
| G1 | 20/05/2016 | 2240 | 1917 | 216 | 186 | 30 | 11 | 34 | 0.3 | 2620 | | 128 | 5 | 3.5 | | | | 0.7213 | |
| G1 | 20/05/2016 | 2260 | 1974 | 219 | 189 | 34 | 11 | 36 | | 2595 | | 128 | 5 | 3.3 | | | | 0.7213 | |
| G1 | 20/05/2016 | 2259 | 1933 | 219 | 189 | 38 | 11 | 34 | 0.4 | 2600 | | 130 | 5 | 3.7 | | | | 0.7223 | |
| G1 | 20/05/2016 | 2253 | 1936 | 214 | 185 | 30 | 11 | 34 | 0.3 | 2592 | | 124 | 5 | 3.5 | 858 | 11.5 | 7.95 | 0.7222 | |
| G1 | 08/09/2016 | 2408 | 2371 | 224 | 190 | 35 | 9 | 39 | 3.5 | 2934 | 3361 | 120 | 6 | 2.1 | 838 | 11.8 | 8.0 | | |
| G1 | 05/12/2016 | 2294 | 2499 | 224 | 197 | 34 | 9 | 31 | 10.6 | 3381 | 2703 | 107 | 6 | 3.5 | 856 | 10.8 | 8.3 | | |
| G1 | 09/03/2017 | 2252 | 2643 | 205 | 150 | 54 | 10 | 64 | 6.1 | 3589 | 2678 | 95 | 6 | 4.6 | 874 | 10.9 | 7.3 | | |
| G1 | 12/06/2017 | 2364 | 2455 | 223 | 149 | 90 | 12 | 87 | 1.9 | 3158 | | 122 | 6 | 5.0 | 900 | 11.4 | 8.2 | | |
| G1 | 25/01/2018 | 2440 | 2174 | 224 | 196 | 32 | 11 | 33 | 3.8 | 3127 | 3442 | 125 | 6 | 5.8 | 893 | 10.9 | 8.4 | | |
| G1 | 19/02/2018 | 2226 | 2028 | 232 | 200 | 41 | 12 | 37 | 3.8 | 2981 | 3230 | 121 | 5 | 5.7 | 849 | 10.6 | 8.4 | | |
| G1 | 03/04/2018 | 2317 | 2054 | 225 | 181 | 48 | 13 | 51 | 4.0 | 2696 | 3672 | 129 | 5 | 5.4 | 837 | | 7.5 | | |
| G1 | 17/09/2018 | 2278 | 2216 | 252 | 223 | 41 | 11 | 34 | 5.4 | 2967 | 3332 | 129 | 6 | 6.8 | 1114 | 11.2 | 8.3 | | |
| G1 | 09/04/2019 | 2035 | 2158 | 185 | 159 | 28 | 11 | 40 | 2.7 | 2901 | 4079 | 97 | 5 | 3.7 | 803 | | 8.1 | 0.7212 | -2.12 |
| G2 | 17/11/2010 | 686 | 667 | 130 | 88 | 34 | 27 | 49 | 3.6 | 1376 | 56 | 364 | | | 313 | | 6.6 | | |
| G2 | 03/02/2011 | 898 | 802 | 204 | 150 | 36 | 29 | 63 | 0 | 1541 | 480 | 267 | | | 361 | 9.2 | 8.5 | | |
| G2 | 01/04/2011 | 886 | 823 | 222 | 173 | 35 | 32 | 57 | 0 | 1615 | 372 | 305 | | | 375 | 10.3 | 8.3 | | |
| G2 | 05/10/2011 | 936 | 831 | 191 | 146 | 97 | 21 | 54 | 5.6 | 1468 | | 213 | | | 385 | 12.1 | 7.4 | | |
| G2 | 11/01/2012 | 848 | 798 | 209 | 166 | 32 | 29 | 49 | 3.3 | 1609 | 232 | 314 | | | 363 | 10.3 | 8.1 | | |
| G2 | 15/02/2012 | 1073 | 864 | 209 | 158 | 40 | 28 | 59 | 14.2 | 1640 | 552 | 252 | | | 389 | 8.6 | 8.1 | | |
| G2 | 28/03/2012 | 898 | 996 | 226 | 182 | 39 | 28 | 51 | 10.7 | 1678 | 604 | 260 | | | 418 | 10.6 | 7.7 | | |
| G2 | 27/06/2012 | 848 | 774 | 178 | 141 | 37 | 28 | 43 | 16.7 | 1356 | 624 | 0 | | | 338 | 12.5 | 8.1 | | |
| G2 | 25/09/2012 | 1160 | 1020 | 226 | 178 | 43 | 27 | 56 | 45.4 | 1855 | | 226 | | | | | | | |
| G2 | 15/12/2013 | 791 | 658 | 159 | 119 | 30 | 26 | 47 | 3.1 | 1437 | 404 | 269 | | | 355 | 10.2 | 8.0 | | |
| G2 | 09/03/2014 | 647 | 630 | 171 | 130 | 30 | 28 | 48 | | 1320 | 196 | 333 | | | 317 | 10.8 | 7.9 | | |
| G2 | 19/11/2015 | 950 | 875 | 181 | 142 | 40 | 24 | 46 | 19.3 | 1694 | 752 | 0 | 2 | 3.1 | 470 | 11.2 | 7.8 | 0.7207 | |
| G2 | 11/01/2016 | 933 | 857 | 184 | 119 | 72 | 23 | 75 | 11.9 | 1769 | 700 | 116 | 6 | 3.9 | 404 | 10.8 | 8.2 | | |
| G2 | 19/04/2016 | 827 | 811 | 172 | 138 | 39 | 21 | 40 | 7.7 | 1676 | 600 | 272 | 2 | 3.1 | 385 | 10.7 | 6.9 | | |
| G2 | 19/05/2016 | 782 | 744 | 181 | 145 | 34 | 25 | 41 | 1.6 | 1530 | | 287 | 2 | 2.8 | 371 | 10.6 | 7.68 | 0.7205 | |
| G2 | 19/05/2016 | 875 | 910 | 179 | 143 | 39 | 23 | 41 | 4.3 | 1542 | | 234 | 2 | 3.0 | 485 | 10.7 | 7.97 | 0.7205 | |
| G2 | 19/05/2016 | 796 | 755 | 184 | 148 | 35 | 25 | 43 | | 1551 | | 286 | 2 | 2.9 | 427 | 10.8 | 7.5 | 0.7205 | |
| G2 | 20/05/2016 | 807 | 764 | 182 | 147 | 34 | 25 | 42 | | 1540 | | 285 | 2 | 2.9 | | | | 0.7205 | |
| G2 | 20/05/2016 | 784 | 765 | 179 | 144 | 33 | 25 | 40 | | 1542 | | 283 | 2 | 2.8 | | | | 0.7206 | |
| G2 | 20/05/2016 | 769 | 751 | 176 | 142 | 34 | 25 | 40 | | 1543 | | 287 | 2 | 2.9 | | | | 0.7205 | |
| G2 | 20/05/2016 | 798 | 770 | 179 | 145 | 34 | 24 | 41 | 2.7 | 1540 | | 278 | 2 | 2.7 | 451 | 11.3 | 7.62 | 0.7206 | |
| G2 | 20/05/2016 | 825 | 787 | 184 | 148 | 35 | 24 | 42 | | 1525 | | 282 | 2 | 2.9 | | | | 0.7205 | |
| G2 | 20/05/2016 | 810 | 772 | 181 | 145 | 34 | 2 | 42 | 2.9 | 1525 | | 280 | 2 | 2.9 | 426 | 11.6 | 7.85 | 0.7206 | |
| G2 | 08/09/2016 | 1015 | 1008 | 186 | 149 | 39 | 27 | 44 | 13.8 | 1763 | 624 | 267 | 2 | 1.4 | 427 | 13 | | | |
| G2 | 09/03/2017 | 885 | 936 | 167 | 125 | 38 | 26 | 49 | 10.9 | 1680 | 584 | 227 | 2 | 2.9 | 390 | 9.9 | 6.6 | | |
| G2 | 12/06/2017 | 731 | 751 | 151 | 117 | 46 | 25 | 39 | 10.4 | 1300 | | 235 | 2 | 2.8 | 372 | 11.6 | | | |
| G2 | 21/08/2017 | 941 | 781 | 171 | 144 | 35 | 25 | 32 | 19.6 | 1638 | 620 | 119 | 2 | 4.2 | 372 | 11.6 | | 0.7206 | |
| G2 | 25/01/2018 | 691 | 691 | 165 | 127 | 45 | 26 | 45 | 2.9 | 1475 | 220 | 318 | 2 | 3.4 | 451 | 9.9 | 8.4 | | |
| G2 | 19/02/2018 | 780 | 742 | 178 | 134 | 58 | 23 | 52 | 3.7 | 1583 | 353 | 297 | 2 | 3.6 | 447 | 10 | 8.4 | | |
| G2 | 03/04/2018 | 683 | 681 | 181 | 141 | 39 | 30 | 47 | 1.6 | 1355 | 160 | 351 | 2 | 3.8 | 406 | | 7.2 | | |
| G2 | 17/09/2018 | 896 | 846 | 200 | 168 | 71 | 26 | 37 | 52.4 | 1377 | 613 | 278 | 2 | 4.4 | 426 | 11.9 | 7.7 | | |
| G2 | 09/04/2019 | 744 | 793 | 159 | 133 | 37 | 28 | 45 | 2.7 | 1582 | 353 | 265 | 2 | 3.0 | 360 | | 7.1 | 0.7203 | -5.45 |
| S10 | 24/09/2010 | 1285 | 1103 | 165 | 131 | 23 | 6 | 40 | 17.6 | 553 | 3751 | 167 | | | 451 | 12.7 | 7.6 | | |
| S10 | 18/11/2010 | 1297 | 1086 | 78 | 44 | 17 | 6 | 40 | 19.0 | 545 | 3835 | 151 | | | 308 | | 6.7 | | |
| S10 | 18/11/2010 | 1335 | 1144 | 91 | 49 | 26 | 6 | 49 | 17.8 | 565 | 3887 | 157 | | | 314 | | 6.8 | | |
| S10 | 02/02/2011 | 1198 | 1070 | 126 | 91 | 15 | 6 | 40 | 22.3 | 508 | 3651 | 143 | | | 422 | 8.4 | 7.5 | | |
| S10 | 03/02/2011 | 1248 | 1059 | 135 | 99 | 20 | 7 | 42 | 26.5 | 531 | 3679 | 152 | | | 435 | 7 | 7.8 | | |


| S10 |  |  |  |  |  |  |  |  |  |  |  |  |  |  |  |  |  |  |  |
|---|---|---|---|---|---|---|---|---|---|---|---|---|---|---|---|---|---|---|---|
| S10 | 28/02/2011 | 1273 | 1111 | 148 | 112 | 21 | 7 | 42 | 23.6 | 541 | 3655 | 154 |  |  | 447 | 9.1 | 7.9 |  |  |
| S10 | 01/03/2011 | 1235 | 1127 | 135 | 101 | 15 | 7 | 39 | 21.1 | 521 | 3683 | 145 |  |  | 445 | 8.8 | 7.4 |  |  |
| S10 | 01/04/2011 | 1235 | 1090 | 135 | 100 | 16 | 8 | 40 | 21.0 | 520 | 3939 | 145 |  |  | 437 | 8.9 | 7.4 |  |  |
| S10 | 01/04/2011 | 1248 | 1111 | 144 | 109 | 21 | 8 | 41 | 22.7 | 541 | 3771 | 154 |  |  | 441 | 9.7 | 7.9 |  |  |
| S10 | 02/06/2011 | 1285 | 1111 | 144 | 110 | 16 | 8 | 39 | 17.5 | 530 | 3783 | 145 |  |  | 440 | 9.2 | 7.8 |  |  |
| S10 | 02/06/2011 | 1310 | 1111 | 157 | 123 | 20 | 8 | 40 | 16.0 | 553 | 3855 | 155 |  |  | 448 | 10 | 8.0 |  |  |
| S10 | 04/10/2011 | 1322 | 1177 | 157 | 122 | 22 | 6 | 40 | 13.5 | 578 | 3999 | 157 |  |  | 473 |  | 8.1 |  |  |
| S10 | 10/01/2012 | 1335 | 1086 | 113 | 80 | 19 | 6 | 39 | 30.6 | 554 | 3807 | 156 |  |  | 462 | 8.5 | 8.2 |  |  |
| S10 | 10/01/2012 | 1273 | 1119 | 139 | 107 | 15 | 6 | 37 | 16.7 | 522 | 3763 | 145 |  |  | 446 | 8.5 | 7.6 |  |  |
| S10 | 28/03/2012 | 1273 | 1185 | 148 | 113 | 16 | 7 | 41 | 17.8 | 541 | 3703 | 148 |  |  | 459 | 7.45 | 9.4 |  |  |
| S10 | 28/03/2012 | 1285 | 1193 | 152 | 116 | 20 | 7 | 43 | 16.6 | 566 | 3699 | 155 |  |  | 462 | 9.4 | 7.9 |  |  |
| S10 | 28/06/2012 | 1205 | 1177 | 137 | 104 | 16 | 6 | 39 | 17.2 | 527 | 3719 | 146 |  |  | 441 | 10.5 | 7.7 |  |  |
| S10 | 28/06/2012 | 1248 | 1185 | 146 | 111 | 20 | 10 | 41 | 15.5 | 552 | 3743 | 154 |  |  | 451 | 11.7 | 8.1 |  |  |
| S10 | 26/09/2012 | 1385 | 1094 | 152 | 119 | 20 | 6 | 39 | 13.4 | 575 | 3991 | 158 |  |  | 466 | 11.4 | 8.2 |  |  |
| S10 | 08/01/2013 | 1322 | 1070 | 126 | 92 | 16 | 5 | 39 | 22.5 | 492 | 3355 | 144 |  |  | 404 | 8.09 | 7.6 |  |  |
| S10 | 08/01/2013 | 1260 | 1136 | 135 | 100 | 21 | 6 | 41 | 29.2 | 541 | 3571 | 150 |  |  | 425 | 9.2 | 8.1 |  |  |
| S10 | 17/09/2013 | 1216 | 1202 | 125 | 95 | 19 |  | 35 | 13.1 | 544 | 3619 | 151 |  |  | 447 | 10.9 | 8.1 |  |  |
| S10 | 16/12/2013 | 1242 | 963 | 118 | 83 | 19 | 5 | 41 | 17.2 | 568 | 3675 | 156 |  |  | 452 | 8.5 | 8.1 |  |  |
| S10 | 10/03/2014 | 1105 | 1013 | 118 | 91 | 19 | 5 | 31 | 26.3 | 542 | 3711 | 160 |  |  | 416 | 9.3 | 8.0 |  |  |
| S10 | 14/06/2014 | 1162 | 1152 | 120 | 91 | 18 | 5 | 34 | 15.8 | 536 | 3675 | 164 | 3 |  | 435 | 11.4 | 8.1 | 0.7150 |  |
| S10 | 14/06/2014 | 1100 | 1113 | 110 | 82 | 14 | 6 | 32 | 21.1 | 508 | 3499 | 157 | 3 |  | 419 | 10.2 | 8.0 | 0.7147 |  |
| S10 | 09/10/2014 | 1291 | 1062 | 136 | 105 | 21 | 5 | 36 | 12.7 | 555 | 3823 | 172 | 3 |  | 432 | 10.6 | 8.1 |  |  |
| S10 | 09/10/2014 | 1204 | 1062 | 122 | 93 | 16 | 5 | 34 | 18.2 | 534 | 3711 | 156 | 3 |  | 440 | 9.4 | 7.8 |  |  |
| S10 | 20/04/2015 | 1097 | 1068 | 117 | 86 | 15 |  | 36 | 22.1 | 481 | 3419 | 153 | 2 |  | 400 | 9.3 | 7.5 |  |  |
| S10 | 20/04/2015 | 1210 | 1151 | 129 | 97 | 20 | 4 | 36 | 21.6 | 532 | 3659 | 160 | 3 |  | 428 | 10.2 | 8.0 |  |  |
| S10 | 13/07/2015 | 1184 | 1053 | 123 | 93 | 18 |  | 35 | 14.3 | 559 | 3792 | 156 | 3 |  | 441 | 11.8 | 8.2 |  |  |
| S10 | 13/07/2015 | 1114 | 1034 | 110 | 83 | 14 | 3 | 32 | 18.3 | 536 | 3621 | 145 | 2 |  | 433 | 10.1 | 7.8 |  |  |
| S10 | 15/10/2015 | 1226 | 1108 | 123 | 92 | 19 |  | 36 | 13.5 | 565 | 4063 | 158 | 3 |  | 461 | 9.7 | 8.2 |  |  |
| S10 | 19/11/2015 | 1193 | 1057 | 123 | 92 | 20 | 5 | 36 | 14.3 | 577 | 4043 | 145 | 3 | 1.3 | 475 | 10.5 | 8.2 | 0.7148 |  |
| S10 | 11/01/2016 | 1279 | 1128 | 131 | 99 | 22 | 5 | 37 | 16.1 | 582 | 3991 | 163 | 3 | 1.2 | 464 | 8.6 | 8.0 |  |  |
| S10 | 19/04/2016 | 1144 | 1068 | 115 | 89 | 15 | 5 | 30 | 14.8 | 522 | 3823 | 147 | 3 | 1.5 | 440 | 9.5 | 7.2 |  |  |
| S10 | 19/04/2016 | 1178 | 1073 | 123 | 100 | 20 | 5 | 27 | 17.1 | 544 | 3823 | 153 | 3 | 1.2 | 448 | 10.3 | 8.0 |  |  |
| S10 | 20/05/2016 | 1198 | 1046 | 132 | 107 | 22 | 4 | 30 | 17.5 | 508 |  | 152 | 3 | 1.2 | 478 | 7.89 | 10.1 | 0.7150 |  |
| S10 | 08/09/2016 | 1267 | 1172 | 122 | 101 | 12 | 5 | 24 | 13.3 | 524 | 3866 | 149 | 3 |  | 454 | 10.7 | 7.5 |  |  |
| S10 | 08/09/2016 | 1249 | 1178 | 123 | 100 | 18 | 5 | 26 | 12.1 | 545 | 3981 | 148 | 3 |  | 464 | 12.1 | 8.1 |  |  |
| S10 | 05/12/2016 | 1307 | 1222 | 130 | 106 | 23 | 7 | 28 | 12.1 | 587 | 4021 | 152 | 3 | 1.2 | 483 | 9 | 8.2 |  |  |
| S10 | 09/03/2017 | 1303 | 1298 | 125 | 101 | 20 | 7 | 28 | 14.6 | 586 | 4085 | 152 | 3 | 2.1 | 474 | 9.7 | 8.2 |  |  |
| S10 | 12/06/2017 | 1263 | 1182 | 130 | 111 | 19 | 6 | 23 | 18.3 | 551 |  | 159 | 3 | 2.1 | 477 | 10.9 | 7.3 |  |  |
| S10 | 12/06/2017 | 1254 | 1174 | 122 | 105 | 15 | 6 | 19 | 13.2 | 539 |  | 153 | 3 | 2.2 | 474 | 11.3 | 7.4 |  |  |
| S10 | 21/08/2017 | 1316 | 996 | 120 | 113 | 14 | 4 | 21 | 10.4 | 545 | 4044 | 70 | 3 | 3.2 | 474 | 11.3 | 7.4 | 0.7145 |  |
| S10 | 21/08/2017 | 1277 | 1031 | 125 | 98 | 19 | 5 | 24 | 9.3 | 560 | 4056 | 72 | 3 | 3.1 | 477 | 10.9 | 7.3 | 0.7150 |  |
| S10 | 20/11/2017 | 1319 | 1235 | 134 | 228 | 21 | 5 | 25 | 8.3 | 581 |  | 165 | 3 | 2.3 | 500 | 9.8 | 8.1 |  |  |
| S10 | 25/01/2018 | 1157 | 1057 | 120 | 83 | 19 | 5 | 27 | 30.3 | 550 | 3751 | 154 | 3 | 2.4 | 476 | 8.7 | 8.3 |  |  |
| S10 | 19/02/2018 | 1182 | 1044 | 250 | 144 | 73 | 5 | 26 | 23.7 | 547 | 3771 | 160 | 3 | 2.4 | 448 | 9.42 | 8.1 |  |  |
| S10 | 07/03/2018 | 987 | 910 | 103 | 125 | 16 | 6 | 24 | 21.2 | 534 | 3752 | 134 | 2 | 1.8 | 410 | 10.8 | 8.1 |  |  |
| S10 | 03/04/2018 | 1532 | 1533 | 171 | 106 | 26 | 6 | 32 | 21.6 | 513 | 3480 | 220 | 4 | 2.8 | 437 |  | 6.5 |  |  |
| S10 | 17/09/2018 | 1284 | 1114 | 147 | 103 | 22 | 8 | 25 | 13.3 | 554 | 3974 | 172 | 3 | 3.0 | 490 | 12.6 | 7.7 |  |  |
| S10 | 09/04/2019 | 1152 | 1124 | 112 | 86 | 18 | 5 | 32 | 16.7 | 543 | 4513 | 125 | 2 | 1.5 | 443 |  | 8.0 | 0.7150 | 2.38 |
| S12 | 23/09/2010 | 412 | 115 | 148 | 124 | 10 | 3 | 27 | 44.5 | 89 | 864 | 144 |  |  | 125 | 11.5 | 7.3 |  |  |
| S12 | 25/09/2010 | 474 | 103 | 74 | 51 | 8 |  | 26 | 41.1 | 51 | 960 | 136 |  |  | 122 | 10.2 | 8.1 |  |  |
| S12 | 18/11/2010 | 449 | 91 | 70 | 48 | 9 | 4 | 26 | 41.1 | 90 | 940 | 137 |  |  | 79 |  | 6.7 |  |  |
| S12 | 02/02/2011 | 387 | 82 | 70 | 47 | 7 | 0 | 27 | 45.3 | 79 | 836 | 130 |  |  | 104 | 4.2 | 7.9 |  |  |
| S12 | 01/03/2011 | 424 | 99 | 65 | 42 | 8 | 0 | 27 | 41.0 | 82 | 900 | 133 |  |  | 114 | 4.7 | 7.8 |  |  |
| S12 | 31/03/2011 | 399 | 93 | 65 | 43 | 8 | 0 | 26 | 39.2 | 80 | 888 | 130 |  |  | 108 | 6.2 | 7.7 |  |  |
| S12 | 02/06/2011 | 437 | 105 | 74 | 50 | 9 | 2 | 28 | 42.5 | 85 | 920 | 132 |  |  | 117 | 7.8 | 7.8 |  |  |
| S12 | 04/10/2011 | 462 | 107 | 74 | 50 | 10 | 0 | 27 | 38.7 | 89 | 656 | 134 |  |  | 120 | 10.8 | 7.8 |  |  |
| S12 | 28/03/2012 | 424 | 91 | 70 | 47 | 8 | 0 | 26 | 34.6 | 84 | 848 | 133 |  |  | 117 | 5.5 | 8 |  |  |
| S12 | 28/06/2012 | 412 | 99 | 74 | 52 | 9 | 0 | 26 | 39.7 | 84 | 904 | 134 |  |  | 116 | 10.8 | 8 |  |  |
| S12 | 08/01/2013 | 399 | 99 | 67 | 46 | 8 | 3 | 25 | 40.2 | 36 | 752 | 133 |  |  |  |  |  |  |  |
| S12 | 17/09/2013 | 416 | 113 | 63 | 45 | 8 |  | 22 | 41.1 | 79 | 840 | 134 |  |  | 113 | 8.4 | 7.7 |  |  |
| S12 | 14/06/2014 | 401 | 104 | 58 | 40 | 7 |  | 20 | 57.1 | 80 | 864 | 143 | 1 |  | 115 | 9.3 | 7.9 | 0.7095 |  |
| S12 | 09/10/2014 | 440 | 100 | 68 | 50 | 9 |  | 21 | 48.2 | 84 | 948 | 149 | 1 |  | 118 | 10.1 | 8.3 |  |  |
| S12 | 20/04/2015 | 382 | 96 | 63 | 47 | 9 |  | 19 | 63.7 | 74 | 824 | 139 | 1 |  | 104 | 6.6 | 7.9 |  |  |
| S12 | 13/07/2015 | 418 | 102 | 65 | 46 | 8 |  | 21 | 54.5 | 84 | 896 | 141 | 1 |  | 119 | 11.5 | 8.9 |  |  |
| S12 | 15/10/2015 | 406 | 101 | 62 | 44 | 8 |  | 21 | 57.8 | 83 | 976 | 132 | 1 |  | 118 | 6.7 | 8.4 |  |  |
| S12 | 19/11/2015 | 425 | 100 | 64 | 47 | 8 |  | 20 | 59.6 | 83 | 916 | 127 | 1 | 0.6 | 123 | 7.8 | 8.5 | 0.7095 |  |
| S12 | 11/01/2016 | 435 | 100 | 66 | 50 | 9 |  | 19 | 57.4 | 82 | 1008 | 135 | 1 | 0.3 | 122 | 6.4 | 8.2 |  |  |
| S12 | 19/04/2016 | 416 | 104 | 63 | 48 | 8 | 3 | 19 | 59.0 | 81 | 1000 | 131 | 1 |  | 120 | 7.2 | 7.3 |  |  |
| S12 | 08/09/2016 | 436 | 113 | 64 | 48 | 8 |  | 19 | 55.6 | 85 | 1016 | 129 | 1 |  | 128 | 11.8 | 7.8 |  |  |
| S12 | 05/12/2016 | 449 | 120 | 67 | 47 | 12 | 3 | 23 | 63.0 | 91 | 1048 | 130 | 1 | 0.3 | 147 | 4.6 | 7.8 |  |  |
| S12 | 09/03/2017 | 426 | 120 | 60 | 41 | 9 | 3 | 22 | 58.8 | 88 | 1016 | 121 | 1 | 0.5 | 124 | 5.1 | 6.5 |  |  |
| S12 | 13/06/2017 | 423 | 111 | 64 | 53 | 7 | 7 | 14 | 64.5 | 81 |  | 133 | 1 |  | 124 | 11.5 | 7.6 |  |  |
| S12 | 21/08/2017 | 419 | 91 | 60 | 47 | 8 | 3 | 15 | 54.8 | 82 | 1008 | 58 | 1 | 0.9 | 124 | 11.5 | 7.5 | 0.7095 |  |
| S12 | 20/11/2017 | 436 | 111 | 66 | 49 | 9 |  | 20 | 63.2 | 85 |  | 136 | 1 | 0.6 | 135 | 4.3 | 8.1 |  |  |
| S12 | 25/01/2018 | 361 | 88 | 60 | 43 | 8 | 6 | 20 | 55.4 | 69 | 847 | 132 | 1 | 0.7 | 137 | 4 | 7.8 |  |  |
| S12 | 19/02/2018 | 382 | 92 | 66 | 50 | 11 | 6 | 19 | 57.8 | 72 | 880 | 135 | 1 | 0.8 | 119 | 5 | 8.2 |  |  |
| S12 | 07/03/2018 | 337 | 82 | 54 | 41 | 7 | 3 | 16 | 62.2 | 75 | 929 | 118 | 1 | 0.7 | 108 | 5.3 | 8.1 |  |  |
| S12 | 03/04/2018 | 393 | 94 | 63 | 45 | 9 | 4 | 21 | 59.9 | 74 | 909 | 133 | 1 | 0.7 | 113 |  | 7.2 |  |  |
| S12 | 17/09/2018 | 424 | 100 | 74 | 57 | 9 | 10 | 21 | 48.5 | 76 | 968 | 145 | 1 | 0.8 | 115 | 12.8 | 8.6 |  |  |
| S12 | 09/04/2019 | 382 | 103 | 53 | 36 | 7 | 3 | 20 | 53.7 | 78 | 1229 | 108 | 1 |  | 118 |  | 7.1 | 0.7094 | 6.03 |
| S13 | 08/09/2010 | 2096 | 757 | 78 | 52 | 17 | 6 | 31 | 25.2 | 1144 | 3247 | 112 |  |  | 508 | 9 | 7.9 |  |  |
| S13 | 23/09/2010 | 2158 | 815 | 130 | 99 | 16 | 7 | 37 | 23.0 | 1206 | 3331 | 126 |  |  | 542 | 8.8 | 7.9 |  |  |
| S13 | 18/11/2010 | 2221 | 831 | 84 | 47 | 20 | 7 | 43 | 27.4 | 1310 | 3339 | 114 |  |  | 534 | 9 | 7.5 |  |  |
| S13 | 03/02/2011 | 1747 | 555 | 83 | 55 | 15 | 7 | 33 | 26.4 | 796 | 3239 | 111 |  |  | 438 | 7.9 | 8.3 |  |  |
| S13 | 01/03/2011 | 1859 | 658 | 83 | 46 | 15 | 6 | 43 | 27.8 | 939 | 3123 | 112 |  |  | 478 | 7.3 | 8.1 |  |  |


| Site | Date | | | | | | | | | | | | | | | | | | | |
|------|------|--|--|--|--|--|--|--|--|--|--|--|--|--|--|--|--|--|--|--|
| S13 | 31/03/2011 | 1884 | 638 | 83 | 52 | 21 | 9 | 35 | 29.0 | 925 | 3319 | 114 | | | 480 | 8.4 | 7.8 | | |
| S13 | 02/06/2011 | 2071 | 699 | 87 | 56 | 17 | 7 | 36 | 23.9 | 1117 | 3287 | 115 | | | 519 | 8.4 | 7.9 | | |
| S13 | 04/10/2011 | 2083 | 732 | 91 | 63 | 16 | 8 | 33 | 18.6 | 1268 | 3299 | 113 | | | 539 | 8.8 | 7.7 | | |
| S13 | 09/01/2012 | 2009 | 691 | 87 | 53 | 18 | 6 | 40 | 28.4 | 1015 | 3383 | 113 | | | 498 | 7.7 | 7.9 | | |
| S13 | 15/02/2012 | 1896 | 601 | 87 | 58 | 16 | 5 | 34 | 24.2 | 891 | 3227 | 112 | | | 462 | 7.7 | 8.1 | | |
| S13 | 27/03/2012 | 1784 | 642 | 91 | 64 | 16 | 6 | 32 | 21.0 | 922 | 3203 | 116 | | | 486 | 8.4 | 7.9 | | |
| S13 | 28/06/2012 | 1921 | 642 | 89 | 58 | 15 | 10 | 36 | 22.8 | 874 | 3199 | 115 | | | 472 | 8.5 | 7.8 | | |
| S13 | 26/09/2012 | 2246 | 724 | 85 | 57 | 14 | 6 | 33 | 17.8 | 1273 | 3351 | 115 | | | 540 | 8.6 | 7.8 | | |
| S13 | 09/01/2013 | 1834 | 568 | 87 | 58 | 15 | 6 | 33 | 29.5 | 675 | 3227 | 114 | | | 442 | 8.1 | 7.8 | | |
| S13 | 16/04/2013 | 1747 | 527 | 87 | 57 | 14 | 7 | 35 | 34.7 | 603 | 3159 | 114 | | | 416 | 8.9 | 7.8 | | |
| S13 | 17/07/2013 | 1851 | 634 | 75 | 44 | 16 | | 36 | 32.2 | 788 | 2931 | 112 | | | 443 | 9.1 | 7.7 | | |
| S13 | 18/09/2013 | 1851 | 754 | 69 | 43 | 17 | | 30 | 23.1 | 1024 | 3283 | 113 | | | 499 | 8.4 | 7.7 | | |
| S13 | 15/12/2013 | 1785 | 539 | 67 | 35 | 16 | 6 | 37 | 30.8 | 904 | 3175 | 111 | | | 477 | 8.1 | 7.8 | | |
| S13 | 10/03/2014 | 1617 | 537 | 72 | 46 | 14 | 4 | 30 | 38.6 | 680 | 3243 | 117 | | | 442 | 8.4 | 8 | | |
| S13 | 14/06/2014 | 1647 | 581 | 70 | 46 | 12 | 5 | 29 | 37.5 | 652 | 3251 | 124 | 16 | | 431 | 8.5 | 8 | 0.7094 | |
| S13 | 09/10/2014 | 1856 | 623 | 76 | 57 | 14 | 5 | 22 | 29.3 | 937 | 3295 | 0 | 19 | | 488 | 8.4 | 7.9 | | |
| S13 | 20/04/2015 | 1758 | 578 | 81 | 56 | 13 | 5 | 30 | 42.4 | 642 | 3343 | 121 | 15 | | 430 | 8.3 | 7.8 | | |
| S13 | 13/07/2015 | 1763 | 606 | 76 | 47 | 14 | 5 | 33 | 32.9 | 911 | 3256 | 116 | 19 | | 471 | 8.8 | 7.5 | | |
| S13 | 15/10/2015 | 1986 | 765 | 76 | 41 | 19 | | 41 | 29.1 | 1235 | 3535 | 116 | 24 | | 547 | 8.02 | 8.4 | | |
| S13 | 19/11/2015 | 1917 | 708 | 74 | 46 | 18 | 6 | 33 | 31.7 | 1143 | 3479 | 107 | 24 | 1.9 | 539 | 8.4 | 7.9 | 0.7093 | |
| S13 | 11/01/2016 | 2002 | 717 | 78 | 75 | 19 | 6 | 3 | 33.0 | 1281 | 3383 | 118 | 24 | 1.7 | 532 | 7.8 | 8.0 | | |
| S13 | 19/04/2016 | 1690 | 590 | 75 | 56 | 15 | 6 | 22 | 35.3 | 821 | 3359 | 114 | 18 | 1.8 | 465 | 8.6 | 7.7 | | |
| S13 | 08/09/2016 | 2019 | 773 | 75 | 54 | 13 | 6 | 24 | 28.0 | 1077 | 3383 | 111 | 22 | | 519 | 9.9 | 7.7 | | |
| S13 | 05/12/2016 | 2126 | 878 | 77 | 54 | 19 | 7 | 27 | 27.2 | 1327 | 3407 | 114 | 26 | 1.7 | 565 | 8.5 | 7.8 | | |
| S13 | 09/03/2017 | 2012 | 915 | 70 | 46 | 16 | 8 | 28 | 30.1 | 1343 | 3428 | 105 | 27 | 2.8 | 567 | 7.8 | 8.1 | | |
| S13 | 13/06/2017 | 1904 | 722 | 88 | 68 | 16 | 6 | 24 | 30.9 | 949 | | 121 | 23 | 3.1 | 564 | 8.8 | 7.6 | | |
| S13 | 21/08/2017 | 2048 | 678 | 71 | 53 | 12 | 6 | 21 | 21.7 | 1289 | 3407 | 50 | 24 | 4.2 | 564 | 8.8 | 7.6 | 0.7093 | |
| S13 | 20/11/2017 | 2166 | 876 | 82 | 58 | 20 | 3 | 28 | 23.3 | 1425 | | 119 | 32 | 3.7 | 586 | 8.3 | 7.8 | | |
| S13 | 25/01/2018 | 1620 | 578 | 78 | 52 | 14 | 6 | 30 | 35.6 | 787 | 3320 | 110 | 18 | 3.1 | 453 | 7.8 | 8.1 | | |
| S13 | 19/02/2018 | 1698 | 553 | 81 | 60 | 17 | 6 | 25 | 35.8 | 775 | 3267 | 117 | 19 | 3.3 | 471 | 9.1 | 7.8 | | |
| S13 | 07/03/2018 | 1545 | 528 | 71 | 47 | 17 | 7 | 29 | 35.9 | 777 | 3326 | 106 | 17 | 2.8 | 436 | 9.2 | 8.1 | | |
| S13 | 03/04/2018 | 1722 | 568 | 87 | 54 | 16 | 7 | 39 | 40.9 | 665 | 3292 | 118 | 18 | 3.1 | 437 | | 7.8 | | |
| S13 | 17/09/2018 | 2034 | 718 | 87 | 67 | 13 | 9 | 24 | 25.8 | 1109 | 3390 | 124 | 25 | 4.4 | 512 | 9.9 | 8.2 | | |
| S13 | 09/04/2019 | 1652 | 619 | 71 | 46 | 15 | 4 | 32 | 34.9 | 805 | 4043 | 93 | 17 | 2.0 | 464 | | 7.8 | 0.7093 | 5.78 |
| S15 | 23/09/2010 | 1734 | 749 | 157 | 80 | 22 | 5 | 90 | 53.5 | 759 | 3223 | 136 | | | 470 | 12.4 | 7.2 | | |
| S15 | 18/11/2010 | 1684 | 691 | 100 | 20 | 17 | 6 | 94 | 108.8 | 721 | 3175 | 121 | | | 454 | 9.5 | 7.9 | | |
| S15 | 02/02/2011 | 1597 | 638 | 113 | 26 | 18 | 6 | 101 | 9.3 | 759 | 3071 | 113 | | | 430 | 8.3 | 7.4 | | |
| S15 | 01/03/2011 | 1697 | 691 | 109 | 36 | 18 | 7 | 85 | 58.6 | 780 | 3099 | 116 | | | 457 | 8.7 | 7.8 | | |
| S15 | 31/03/2011 | 1647 | 658 | 109 | 36 | 18 | 7 | 85 | 57.3 | 739 | 3171 | 116 | | | 439 | 9.3 | 7.8 | | |
| S15 | 02/06/2011 | 1697 | 667 | 113 | 42 | 18 | 8 | 83 | 54.3 | 773 | 3359 | 118 | | | 461 | 9.9 | 7.8 | | |
| S15 | 04/10/2011 | 1747 | 699 | 126 | 52 | 20 | 5 | 86 | 51.4 | 833 | 3251 | 120 | | | 475 | 11 | 7.6 | | |
| S15 | 09/01/2012 | 1597 | 617 | 122 | 30 | 16 | 6 | 107 | 66.4 | 684 | 2943 | 118 | | | 420 | 9.3 | 7.9 | | |
| S15 | 27/03/2012 | 1747 | 749 | 122 | 42 | 18 | 7 | 93 | 56.6 | 832 | 3083 | 117 | | | 470 | 9.4 | 7.7 | | |
| S15 | 28/06/2012 | 1846 | 732 | 113 | 43 | 19 | 7 | 82 | 49.3 | 787 | 3147 | 118 | | | 460 | 10.8 | 7.9 | | |
| S15 | 26/09/2012 | 1772 | 658 | 117 | 44 | 19 | 7 | 85 | 51.1 | 802 | 3259 | 121 | | | 460 | 12.1 | 7.5 | | |
| S15 | 09/01/2013 | 1697 | 683 | 115 | 24 | 18 | 5 | 107 | 76.1 | 727 | 3163 | 115 | | | 445 | 8.9 | | | |
| S15 | 17/07/2013 | 1676 | 740 | 105 | 33 | 20 | | 84 | 55.7 | 655 | 3007 | 117 | | | 431 | 11 | 7.9 | | |
| S15 | 17/09/2013 | 1432 | 768 | 88 | 21 | 16 | | 78 | 49.3 | 698 | 3179 | 117 | | | 450 | 10.7 | 7.7 | | |
| S15 | 15/12/2013 | 1603 | 598 | 94 | 16 | 17 | 5 | 90 | 66.1 | 733 | 3091 | 116 | | | 447 | 9.4 | 7.7 | | |
| S15 | 10/03/2014 | 1509 | 620 | 98 | 16 | 17 | 5 | 95 | 68.7 | 669 | 3131 | 120 | | | 423 | 9.3 | 7.8 | | |
| S15 | 14/06/2014 | 1558 | 706 | 97 | 23 | 17 | 6 | 86 | 56.9 | 694 | 3247 | 126 | 11 | | 431 | 10.8 | 7.8 | 0.7096 | |
| S15 | 09/10/2014 | 1654 | 650 | 106 | 35 | 19 | 4 | 83 | 54.9 | 694 | 3280 | 128 | 11 | | 454 | 11 | 7.9 | | |
| S15 | 20/04/2015 | 1630 | 708 | 109 | 27 | 18 | 4 | 96 | 63.2 | 688 | 3199 | 124 | 11 | | 436 | 9.7 | 7.8 | | |
| S15 | 13/07/2015 | 1589 | 666 | 101 | 29 | 17 | 4 | 84 | 53.3 | 721 | 3205 | 123 | 11 | | 450 | 11.4 | 7.8 | | |
| S15 | 15/10/2015 | 1533 | 635 | 100 | 20 | 16 | | 93 | 132.5 | 666 | 3371 | 123 | 10 | | 453 | 11.1 | 7.8 | | |
| S15 | 19/11/2015 | 1574 | 653 | 103 | 18 | 17 | 5 | 100 | 59.0 | 742 | 3371 | 115 | 11 | 2.2 | 474 | 10.7 | 7.8 | 0.7096 | |
| S15 | 11/01/2016 | 1593 | 651 | 104 | 23 | 18 | 5 | 95 | 105.7 | 765 | 3331 | 119 | 10 | 2.1 | 462 | 9.8 | 7.7 | | |
| S15 | 19/04/2016 | 1536 | 669 | 101 | 24 | 17 | 5 | 90 | 53.8 | 742 | 3279 | 118 | 10 | 2.3 | 453 | 10.5 | 7.6 | | |
| S15 | 08/09/2016 | 1580 | 705 | 102 | 35 | 19 | 4 | 78 | 53.7 | 700 | 3302 | 112 | 10 | 0.6 | 453 | 13.1 | 7.5 | | |
| S15 | 09/03/2017 | 1479 | 744 | 96 | 0 | 15 | 7 | 111 | 150.9 | 684 | 3134 | 106 | 9 | 2.7 | 474 | 9.2 | 7.1 | | |
| S15 | 13/06/2017 | 1709 | 740 | 114 | 37 | 19 | 6 | 90 | 54.5 | 757 | | 127 | 12 | 3.5 | 470 | 13.5 | 7.6 | | |
| S15 | 21/08/2017 | 1557 | 589 | 98 | 27 | 15 | 5 | 83 | 51.1 | 758 | 3272 | 53 | 9 | 4.1 | 470 | 13.5 | 7.6 | 0.7096 | |
| S15 | 25/01/2018 | 1514 | 646 | 110 | 10 | 17 | 5 | 117 | 69.0 | 739 | 3109 | 116 | 10 | 3.4 | 457 | 9.1 | 8.0 | | |
| S15 | 19/02/2018 | 1554 | 643 | 111 | 25 | 19 | 5 | 100 | 70.6 | 754 | 3183 | 119 | 11 | 3.6 | 449 | 9.7 | 7.9 | | |
| S15 | 07/03/2018 | 1256 | 547 | 89 | 6 | 14 | 6 | 96 | 64.0 | 697 | 3167 | 97 | 8 | 2.9 | 427 | 10 | 8.0 | | |
| S15 | 03/04/2018 | 1545 | 651 | 120 | 10 | 17 | 6 | 128 | 69.0 | 638 | 3070 | 119 | 11 | 3.3 | 450 | | 7.7 | | |
| S15 | 17/09/2018 | 1637 | 678 | 125 | 46 | 20 | 9 | 92 | 55.1 | 717 | 3322 | 129 | 12 | 4.3 | 452 | 12.3 | 8.5 | | |
| S15 | 09/04/2019 | 1490 | 712 | 99 | 73 | 19 | 3 | 132 | 64.1 | 742 | 4216 | 96 | 9 | 2.4 | 458 | | 7.7 | 0.7096 | 6.39 |
| S18 | 08/09/2010 | 2046 | 601 | 96 | 74 | 9 | 7 | 25 | 7.4 | 635 | 3895 | 118 | | | 459 | 9.1 | 7.8 | | |
| S18 | 23/09/2010 | 2083 | 642 | 104 | | 15 | 6 | 508 | 8.9 | 636 | 3795 | 121 | | | 480 | 9.5 | 7.4 | | |
| S18 | 18/11/2010 | 1934 | 601 | 86 | 62 | 8 | 7 | 28 | 5.9 | 622 | 3788 | 122 | | | 462 | 8.11 | 8 | | |
| S18 | 02/02/2011 | 1996 | 545 | 87 | 62 | 9 | 7 | 30 | 7.4 | 660 | 3939 | 122 | | | 470 | 7.2 | 7.7 | | |
| S18 | 01/03/2011 | 1984 | 592 | 91 | 67 | 9 | 7 | 28 | 6.9 | 646 | 3827 | 124 | | | 483 | 7.9 | 8.0 | | |
| S18 | 31/03/2011 | 2009 | 566 | 91 | 68 | 9 | 9 | 27 | 6.7 | 644 | 3871 | 121 | | | 472 | 8.6 | 7.7 | | |
| S18 | 02/06/2011 | 1971 | 560 | 96 | 50 | 9 | 7 | 54 | 2.0 | 798 | 3891 | 121 | | | 481 | 8.8 | 7.8 | | |
| S18 | 04/10/2011 | 1934 | 560 | 96 | 71 | 10 | 7 | 29 | 7.2 | 630 | 3807 | 121 | | | 479 | 8.8 | 7.6 | | |
| S18 | 09/01/2012 | 1959 | 568 | 91 | 68 | 8 | 6 | 26 | 6.4 | 637 | 3759 | 123 | | | 464 | 8 | 7.9 | | |
| S18 | 27/03/2012 | 2071 | 625 | 96 | 49 | 9 | 7 | 54 | | 858 | 3815 | 123 | | | 495 | 8.8 | 7.2 | | |
| S18 | 28/06/2012 | 2096 | 658 | 91 | 66 | 10 | 8 | 29 | 5.8 | 703 | 3999 | 123 | | | 499 | 8.6 | 7.8 | | |
| S18 | 26/09/2012 | 2046 | 568 | 94 | 71 | 9 | 8 | 26 | 5.3 | 642 | 3859 | 120 | | | 470 | 8.9 | 7.7 | | |
| S18 | 09/01/2013 | 2221 | 675 | 83 | 57 | 10 | 6 | 30 | 5.9 | 785 | 3919 | 123 | | | 522 | 8.5 | 7.8 | | |
| S18 | 16/04/2013 | 2258 | 708 | 80 | 55 | 10 | 6 | 30 | 7.0 | 824 | 4023 | 122 | | | 522 | 8.9 | 7.6 | | |
| S18 | 17/07/2013 | 2101 | 688 | 70 | 46 | 9 | | 28 | 10.2 | 690 | 3755 | 115 | | | 490 | 9.6 | 7.6 | | |
| S18 | 17/09/2013 | 2052 | 697 | 78 | 56 | 10 | 4 | 25 | 8.3 | 634 | 3727 | 118 | | | 478 | 8.5 | 7.6 | | |
| S18 | 15/12/2013 | 1837 | 512 | 68 | 44 | 8 | 6 | 29 | 10.1 | 637 | 3747 | 118 | | | 466 | 9.2 | 7.7 | | |
| S18 | 10/03/2014 | 1954 | 618 | 66 | 46 | 8 | 6 | 24 | 9.1 | 715 | 4075 | 125 | | | 501 | 8.5 | 7.6 | | |
| S18 | 14/06/2014 | 1951 | 683 | 67 | 45 | 8 | 5 | 26 | 10.8 | 691 | 3983 | 129 | 5 | | 497 | 9 | 7.9 | 0.7097 | |
| S18 | 09/10/2014 | 1930 | 564 | 79 | 65 | 9 | 5 | 16 | 12.7 | 616 | 3875 | 130 | 5 | | 474 | 8.9 | 7.9 | | |
| S18 | 20/04/2015 | 2071 | 701 | 73 | 49 | 9 | 6 | 27 | 9.8 | 746 | 4179 | 127 | 5 | | 504 | 9 | 7.7 | | |
| S18 | 13/07/2015 | 1940 | 593 | 77 | 53 | 10 | 5 | 28 | 12.5 | 648 | 3811 | 126 | 5 | | 471 | 10.7 | 8.0 | | |
| S18 | 15/10/2015 | 1762 | 557 | 72 | 49 | 8 | | 26 | 10.8 | 594 | 3975 | 117 | 4 | | 462 | 8.4 | 8.2 | | |
| S18 | 19/11/2015 | 1739 | 531 | 75 | 51 | 9 | 7 | 27 | 12.1 | 604 | 3915 | 107 | 4 | 2.1 | 469 | 8.6 | 7.9 | 0.7095 | |
| S18 | 11/01/2016 | 1832 | 536 | 80 | 57 | 9 | 5 | 27 | 39.8 | 605 | 3871 | 121 | 5 | 1.9 | 465 | 8.2 | 7.9 | | |

| Site | Date | | | | | | | | | | | | | | | | | |
|---|---|---|---|---|---|---|---|---|---|---|---|---|---|---|---|---|---|---|---|
| S18 | 19/04/2016 | 1815 | 577 | 77 | 62 | 9 | 6 | 18 | 11.9 | 633 | 3999 | 119 | 5 | 2.1 | 478 | 9 | 7.5 | | |
| S18 | 08/09/2016 | 1881 | 637 | 77 | 59 | 11 | 6 | 21 | 15.1 | 583 | 3917 | 112 | 4 | | 464 | 10.2 | 7.6 | | |
| S18 | 05/12/2016 | 1854 | 620 | 81 | 60 | 12 | 8 | 24 | 13.2 | 600 | 3823 | 118 | 4 | 1.8 | 463 | 8.4 | 8.1 | | |
| S18 | 09/03/2017 | 1777 | 634 | 76 | 59 | 9 | 8 | 19 | 12.1 | 589 | 3841 | 109 | 4 | 2.9 | 462 | 8.7 | 7.9 | | |
| S18 | 13/06/2017 | 1897 | 612 | 85 | 74 | 10 | 7 | 13 | 14.0 | 610 | | 127 | 5 | 3.3 | 467 | 11.2 | 7.8 | | |
| S18 | 21/08/2017 | 1761 | 490 | 75 | 62 | 8 | 6 | 15 | 12.2 | 576 | 3864 | 52 | 4 | 4.1 | 467 | 11.2 | 7.6 | 0.7096 | |
| S18 | 20/11/2017 | 1785 | 607 | 82 | 66 | 8 | 6 | 18 | 11.2 | 571 | | 122 | 5 | 3.2 | 458 | 8.5 | 7.9 | | |
| S18 | 25/01/2018 | 1774 | 570 | 75 | 61 | 9 | 6 | 17 | 12.5 | 659 | 3935 | 115 | 5 | 3.5 | 506 | 7.7 | 8.2 | | |
| S18 | 19/02/2018 | 1912 | 578 | 82 | 67 | 11 | 5 | 17 | 12.5 | 675 | 3982 | 125 | 5 | 4.1 | 497 | 9.2 | 7.8 | | |
| S18 | 07/03/2018 | 1626 | 496 | 67 | 55 | 8 | 6 | 14 | 11.7 | 657 | 4010 | 107 | 4 | 3.0 | 480 | 9.1 | 8.6 | | |
| S18 | 03/04/2018 | 1988 | 613 | 79 | 62 | 10 | 7 | 20 | 12.4 | 622 | 4012 | 127 | 5 | 3.8 | 509 | | 7.6 | | |
| S18 | 17/09/2018 | 1899 | 584 | 90 | 60 | 14 | 9 | 36 | 12.1 | 620 | 3942 | 130 | 5 | 4.7 | 458 | 12 | 7.8 | | |
| S18 | 09/04/2019 | 1804 | 611 | 69 | 49 | 8 | 6 | 23 | 11.6 | 663 | 5187 | 101 | 4 | 2.5 | 489 | | 7.6 | | |
| S20 | 08/09/2010 | 2083 | 708 | 96 | 53 | 17 | 7 | 50 | 27.1 | 951 | 3531 | 114 | | | 495 | 11 | 7.7 | | |
| S20 | 23/09/2010 | 2133 | 749 | 148 | 100 | 17 | 7 | 56 | 32.5 | 979 | 3591 | 123 | | | 531 | 12.9 | 7.1 | | |
| S20 | 18/11/2010 | 2196 | 724 | 57 | 12 | 17 | 7 | 52 | 23.3 | 1098 | 3591 | 116 | | | 517 | 9 | 7.9 | | |
| S20 | 02/02/2011 | 1921 | 617 | 104 | 39 | 17 | 7 | 77 | 45.6 | 888 | 3447 | 108 | | | 475 | 7.3 | 8.1 | | |
| S20 | 01/03/2011 | 1959 | 691 | 109 | 51 | 17 | 8 | 67 | 36.2 | 910 | 3355 | 110 | | | 497 | 7.4 | 8.1 | | |
| S20 | 31/03/2011 | 1971 | 638 | 104 | 41 | 17 | 8 | 74 | 34.8 | 881 | 3483 | 110 | | | 489 | 8.4 | 7.9 | | |
| S20 | 02/06/2011 | 2108 | 642 | 109 | 55 | 16 | 9 | 62 | 27.9 | 954 | 3543 | 0 | | | 513 | 8.8 | 7.8 | | |
| S20 | 04/10/2011 | 2096 | 716 | 109 | 59 | 16 | 6 | 58 | 25.8 | 1071 | 3595 | 118 | | | 538 | 10.1 | 7.6 | | |
| S20 | 09/01/2012 | 2046 | 667 | 104 | 30 | 13 | 6 | 87 | 38.9 | 955 | 3439 | 109 | | | 496 | 8 | 7.8 | | |
| S20 | 27/03/2012 | 1896 | 675 | 100 | 56 | 16 | 8 | 51 | 24.4 | 950 | 3175 | 110 | | | 499 | 8.9 | 7.7 | | |
| S20 | 28/06/2012 | 2046 | 691 | 111 | 52 | 16 | 13 | 69 | 28.8 | 876 | 3563 | 116 | | | 505 | 10.8 | 7.7 | | |
| S20 | 26/09/2012 | 2221 | 691 | 109 | 59 | 15 | 7 | 58 | 22.4 | 1046 | 3615 | 117 | | | 519 | 10.7 | 7.8 | | |
| S20 | 08/01/2013 | 1971 | 642 | 120 | 41 | 16 | 6 | 92 | 56.9 | 755 | 3415 | 110 | | | 485 | 8.2 | 7.8 | | |
| S20 | 16/04/2013 | 1846 | 609 | 111 | 42 | 16 | 8 | 81 | 41.7 | 673 | 3371 | 109 | | | 452 | 8.7 | 7.8 | | |
| S20 | 17/07/2013 | 1912 | 626 | 86 | 41 | 15 | | 53 | 32.7 | 731 | 3184 | 112 | | | 460 | 10.9 | 7.6 | | |
| S20 | 17/09/2013 | 2051 | 705 | 91 | 44 | 16 | | 55 | 27.0 | 876 | 3431 | 117 | | | 502 | 10.1 | 7.6 | | |
| S20 | 15/12/2013 | 1810 | 538 | 77 | 28 | 15 | 7 | 57 | 35.1 | 894 | 3199 | 104 | | | 483 | 7.5 | 7.7 | | |
| S20 | 10/03/2014 | 1701 | 561 | 85 | 34 | 13 | 6 | 59 | 42.1 | 696 | 3451 | 112 | | | 448 | 7.5 | 7.9 | | |
| S20 | 14/06/2014 | 1841 | 653 | 94 | 32 | 17 | 5 | 73 | 44.1 | 684 | 3875 | 124 | 19 | | 478 | 10.6 | 7.7 | 0.7094 | |
| S20 | 09/10/2014 | 1993 | 609 | 100 | 49 | 16 | 5 | 59 | 38.7 | 766 | 3763 | 129 | 18 | | 498 | 10.1 | 7.8 | | |
| S20 | 20/04/2015 | 1883 | 647 | 109 | 35 | 16 | 6 | 85 | 56.2 | 709 | 3655 | 119 | 17 | | 480 | 8.8 | 7.8 | | |
| S20 | 13/07/2015 | 1729 | 629 | 85 | 42 | 12 | 6 | 49 | 30.6 | 879 | 3177 | 121 | 18 | | 471 | 14.1 | 8.3 | | |
| S20 | 15/10/2015 | 2012 | 669 | 112 | 42 | 17 | | 82 | 34.6 | 938 | 3843 | 123 | 19 | | 522 | 9.6 | 7.9 | | |
| S20 | 19/11/2015 | 1910 | 636 | 97 | 33 | 17 | 6 | 74 | 41.7 | 986 | 3739 | 108 | 19 | 1.9 | 531 | 9.9 | 7.8 | 0.7093 | |
| S20 | 11/01/2016 | 2031 | 672 | 107 | 48 | 18 | 6 | 69 | 44.5 | 1071 | 3655 | 119 | 19 | 1.7 | 524 | 9 | 7.8 | | |
| S20 | 19/04/2016 | 1857 | 644 | 100 | 41 | 19 | 6 | 70 | 42.9 | 854 | 3959 | 111 | 18 | 1.8 | 491 | 9 | 7.6 | | |
| S20 | 08/09/2016 | 1942 | 667 | 96 | 52 | 14 | 6 | 52 | 35.3 | 842 | 3669 | 113 | 18 | | 506 | 10.9 | 7.6 | | |
| S20 | 05/12/2016 | 2023 | 730 | 102 | 50 | 19 | 8 | 60 | 34.8 | 1019 | 3580 | 112 | 18 | 1.5 | 527 | 9.9 | 7.9 | | |
| S20 | 09/03/2017 | 1958 | 767 | 93 | 33 | 14 | 8 | 69 | 41.4 | 1081 | 3529 | 99 | 18 | 2.8 | 530 | 834 | 8.0 | | |
| S20 | 12/06/2017 | 1915 | 730 | 103 | 48 | 14 | 7 | 64 | 42.5 | 944 | | 115 | 19 | 2.9 | 539 | 11.1 | 7.5 | | |
| S20 | 21/08/2017 | 2014 | 608 | 95 | 50 | 13 | 6 | 53 | 30.1 | 1038 | 3651 | 53 | 18 | 4.1 | 539 | 11.1 | 7.5 | 0.7094 | |
| S20 | 20/11/2017 | 2113 | 769 | 112 | 57 | 16 | 6 | 65 | 32.5 | 1133 | | 122 | 22 | 3.3 | 560 | 9.8 | 7.6 | | |
| S20 | 25/01/2018 | 1738 | 619 | 101 | 39 | 15 | 6 | 71 | 43.5 | 837 | 3339 | 109 | 18 | 3.4 | 521 | 7.1 | 7.9 | | |
| S20 | 19/02/2018 | 1711 | 590 | 100 | 44 | 15 | 6 | 65 | 42.2 | 816 | 3328 | 108 | 18 | 3.2 | 470 | 7.8 | 8.1 | | |
| S20 | 07/03/2018 | 1534 | 533 | 88 | 30 | 12 | 7 | 67 | 42.9 | 801 | 3311 | 98 | 15 | 2.6 | 444 | 8.3 | 8.2 | | |
| S20 | 03/04/2018 | 1723 | 582 | 105 | 30 | 16 | 7 | 88 | 44.6 | 680 | 3307 | 111 | 18 | 2.9 | 460 | | 7.7 | | |
| S20 | 17/09/2018 | 1996 | 647 | 121 | 69 | 18 | 9 | 61 | 34.6 | 852 | 3692 | 127 | 21 | 4.2 | 496 | 12.2 | 8.1 | | |
| S20 | 09/04/2019 | 1651 | 626 | 85 | 60 | 12 | 6 | 63 | 34.5 | 826 | 4533 | 86 | 15 | 1.9 | 469 | | 7.7 | 0.7094 | 6.38 |
| S21 | 08/09/2010 | 1921 | 667 | 78 | 37 | 13 | 6 | 48 | 36.9 | 949 | 3347 | 114 | | | 482 | 13.4 | 7.9 | | |
| S21 | 23/09/2010 | 2021 | 708 | 157 | 110 | 13 | 7 | 54 | 37.4 | 987 | 3323 | 128 | | | 505 | 12.3 | 7.4 | | |
| S21 | 18/11/2010 | 2096 | 699 | 92 | 45 | 13 | 7 | 55 | 57.2 | 967 | 3503 | 116 | | | 515 | 9.3 | 8.1 | | |
| S21 | 02/02/2011 | 1796 | 576 | 96 | 36 | 12 | 7 | 69 | 43.7 | 857 | 3187 | 109 | | | 448 | 6.6 | 8.2 | | |
| S21 | 01/03/2011 | 1834 | 625 | 91 | 41 | 11 | 8 | 59 | 37.4 | 902 | 3107 | 108 | | | 472 | 7.1 | 8.1 | | |
| S21 | 31/03/2011 | 1859 | 617 | 83 | 34 | 12 | 9 | 56 | 47.1 | 896 | 3203 | 107 | | | 467 | 8.03 | 8.6 | | |
| S21 | 02/06/2011 | 1934 | 650 | 100 | 55 | 12 | 10 | 53 | 30.7 | 937 | 3343 | 112 | | | 496 | 10.9 | 7.9 | | |
| S21 | 04/10/2011 | 2271 | 691 | 100 | 49 | 13 | 8 | 59 | 27.4 | 1011 | 2219 | 117 | | | 516 | 12.2 | 7.8 | | |
| S21 | 09/01/2012 | 1996 | 650 | 104 | 40 | 10 | 7 | 75 | 64.4 | 945 | 3319 | 111 | | | 489 | 7.9 | 8.0 | | |
| S21 | 27/03/2012 | 1896 | 683 | 104 | 52 | 12 | 9 | 61 | 30.9 | 941 | 3131 | 108 | | | 484 | 9.1 | 7.9 | | |
| S21 | 28/06/2012 | 1971 | 658 | 104 | 57 | 12 | 7 | 56 | 24.0 | 856 | | 115 | | | 483 | 12.3 | 7.7 | | |
| S21 | 26/09/2012 | 2009 | 617 | 102 | 52 | 11 | 6 | 59 | 25.3 | 963 | 3527 | 118 | | | 498 | 12.8 | 7.9 | | |
| S21 | 09/01/2013 | 1896 | 411 | 102 | 46 | 10 | 7 | 66 | 41.9 | 772 | 3203 | 105 | | | 463 | 7.6 | 7.8 | | |
| S21 | 16/04/2013 | 1784 | 560 | 115 | 55 | 12 | 7 | 70 | 38.9 | 695 | 3123 | 104 | | | 357 | 11.6 | 7.9 | | |
| S21 | 17/07/2013 | 1898 | 647 | 89 | 43 | 11 | 5 | 53 | 30.9 | 717 | 3183 | 112 | | | 448 | 14.5 | 8.2 | | |
| S21 | 17/09/2013 | 1928 | 701 | 88 | 45 | 10 | | 50 | 26.3 | 830 | 3351 | 116 | | | 485 | 11.9 | 7.9 | | |
| S21 | 15/12/2013 | 1807 | 538 | 81 | 31 | 11 | 6 | 58 | 39.4 | 847 | 3255 | 109 | | | 476 | 8.3 | 7.9 | | |
| S21 | 10/03/2014 | 1730 | 571 | 85 | 36 | 10 | 7 | 57 | 58.6 | 714 | 3371 | 112 | | | 445 | 7.7 | 8.1 | | |
| S21 | 14/06/2014 | 1710 | 613 | 81 | 44 | 9 | 6 | 43 | 36.2 | 675 | 3375 | 122 | 18 | | 441 | 12.4 | 7.9 | 0.7094 | |
| S21 | 09/10/2014 | 1890 | 591 | 90 | 48 | 11 | 5 | 49 | 30.4 | 787 | 3439 | 127 | 18 | | 477 | 11.9 | 8.1 | | |
| S21 | 20/04/2015 | 1782 | 613 | 95 | 44 | 12 | 4 | 60 | 42.7 | 688 | 3359 | 116 | 17 | | 438 | 9.6 | 7.9 | | |
| S21 | 13/07/2015 | 1845 | 606 | 101 | 40 | 13 | 4 | 71 | 34.9 | 811 | 3459 | 122 | 18 | | 508 | 12.5 | 7.8 | | |
| S21 | 15/10/2015 | 1863 | 625 | 123 | 39 | 11 | 5 | 98 | 37.5 | 893 | 3727 | 118 | 18 | | 508 | 11.3 | 8.1 | | |
| S21 | 19/11/2015 | 1818 | 601 | 90 | 33 | 11 | 6 | 67 | 36.1 | 937 | 3603 | 107 | 18 | 1.7 | 509 | 10.5 | 8.0 | 0.7093 | |
| S21 | 11/01/2016 | 1919 | 643 | 97 | 43 | 12 | 6 | 63 | 46.2 | 992 | 3507 | 117 | 19 | 1.6 | 500 | 8.8 | 7.9 | | |
| S21 | 19/04/2016 | 1729 | 603 | 92 | 42 | 11 | 6 | 59 | 38.8 | 836 | 3439 | 107 | 17 | 1.6 | 472 | 9.9 | 7.9 | | |
| S21 | 08/09/2016 | 1962 | 666 | 96 | 52 | 12 | 6 | 51 | 33.0 | 819 | 3634 | 119 | 19 | | 493 | 14.1 | 7.8 | | |
| S21 | 05/12/2016 | 1923 | 690 | 101 | 46 | 13 | 8 | 63 | 32.4 | 918 | 3492 | 112 | 18 | 1.4 | 502 | 9.3 | 7.9 | | |
| S21 | 09/03/2017 | 1847 | 743 | 93 | 26 | 8 | 8 | 78 | 45.9 | 965 | 3321 | 98 | 17 | 2.5 | 494 | 8 | 8.1 | | |
| S21 | 13/06/2017 | 1896 | 703 | 113 | 54 | 12 | 8 | 68 | 35.4 | 901 | | 117 | 19 | 2.8 | 520 | 14.5 | 4.7 | | |
| S21 | 21/08/2017 | 1917 | 589 | 96 | 46 | 10 | 6 | 58 | 28.8 | 967 | 3524 | 52 | 17 | 4.2 | 520 | 14.5 | 4.7 | 0.7094 | |
| S21 | 20/11/2017 | 1961 | 702 | 103 | 50 | 10 | 6 | 62 | 28.2 | 1008 | | 119 | 20 | 3.2 | 544 | 9.7 | 7.6 | | |
| S21 | 25/01/2018 | 1797 | 617 | 119 | 30 | 10 | 6 | 104 | 51.9 | 855 | 3350 | 108 | 18 | 3.3 | 471 | 7.4 | 8.1 | | |
| S21 | 19/02/2018 | 1734 | 579 | 113 | 49 | 12 | 6 | 74 | 42.1 | 829 | 3288 | 109 | 17 | 3.0 | 468 | 8 | 8.2 | | |
| S21 | 07/03/2018 | 1525 | 519 | 95 | 15 | 8 | 7 | 94 | 44.0 | 814 | 3303 | 95 | 15 | 2.6 | 446 | 9.5 | 8.3 | | |
| S21 | 03/04/2018 | 1764 | 582 | 110 | 28 | 10 | 8 | 96 | 60.3 | 708 | 3282 | 109 | 17 | 3.1 | 464 | | 7.9 | | |
| S21 | 17/09/2018 | 1912 | 615 | 114 | 67 | 11 | 10 | 55 | 22.9 | 838 | 3559 | 132 | 20 | 4.1 | 494 | 16.3 | 7.7 | | |
| S21 | 09/04/2019 | 1650 | 643 | 92 | 66 | 10 | 7 | 93 | 30.3 | 851 | 4503 | 88 | 15 | 1.9 | 484 | | 7.9 | 0.7094 | 6.54 |
| Rain | 19/05/2016 | 25 | 6 | 26 | | 24 | | 28 | 24.6 | 20 | | 1 | 0 | 0 | 22 | 11.4 | 7.1 | | |
| Rain | 28/05/2016 | 8 | 0 | 5 | | 4 | | 5 | 8.1 | 6 | 72 | 0 | 0 | 0 | 23 | | 5.5 | | |
| Rain | 31/05/2016 | 12 | 2 | 8 | | 5 | | 8 | 9.6 | 6 | 72 | 0 | 0 | 0 | 25 | | 6 | | |
| Rain | 28/05/2016 | 9 | 2 | 21 | | 14 | | 20 | 8.2 | 9 | 72 | 0 | 0 | 0 | 29 | | 5.8 | | |
| Rain | 31/05/2016 | 15 | 2 | 5 | | 5 | | 5 | 11.9 | 6 | 72 | 0 | 0 | 0 | 29 | | 5.9 | | |
| Rain | 19/05/2016 | 2219 | 1953 | 212 | | 28 | | 11 | 26.9 | 26 | | 126 | 5 | 3.5 | 45 | 10.4 | 7.4 | | |
| Rain | 28/05/2016 | 7 | 3 | 3 | | 9 | | 4 | 7.2 | 4 | 72 | 0 | 0 | 0 | 29 | | 5.8 | | |
| Rain | 31/05/2016 | 12 | 2 | 4 | | 4 | | 3 | 9.3 | 4 | 72 | 0 | 0 | 0 | 25 | | 5.9 | | |

**Appendix C: Identifying chloride inputs (atmospheric, anthropogenic, salts): X *vs.* Cl graphs**

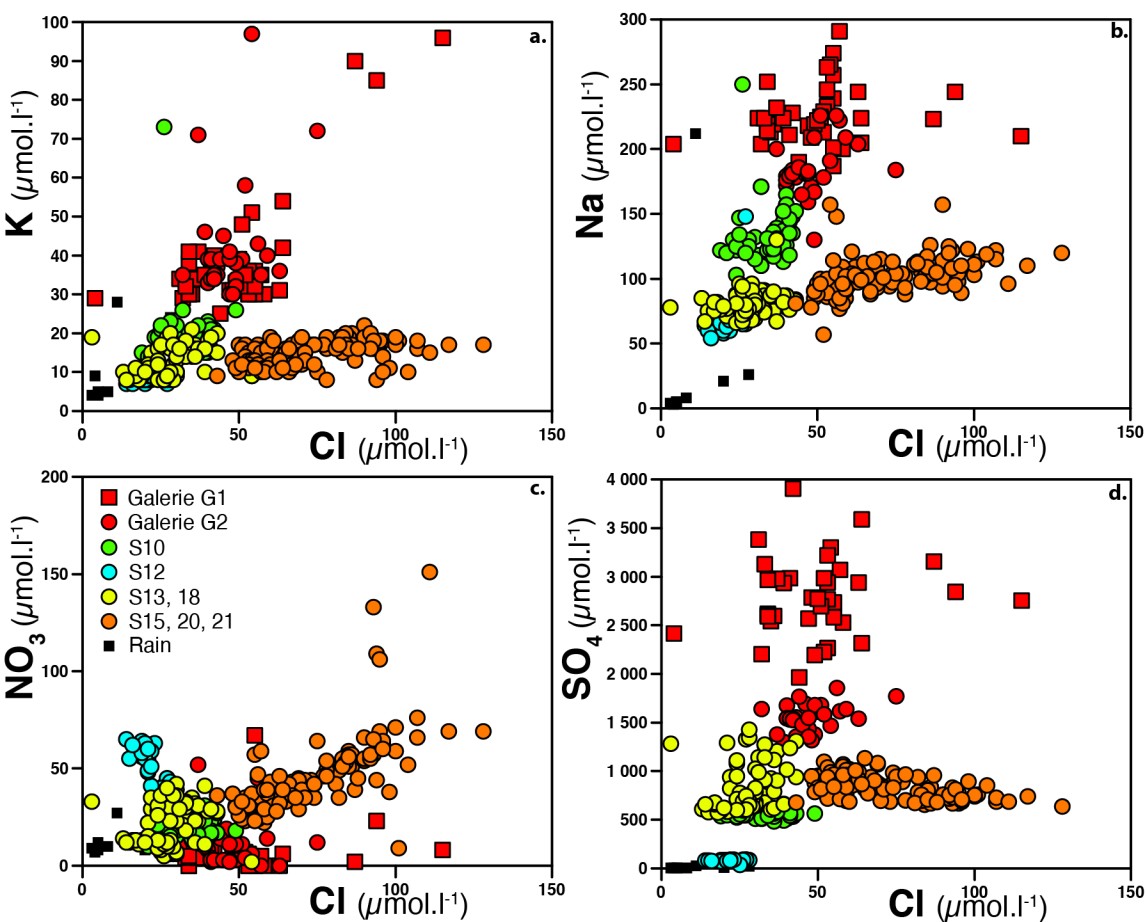

Figure C1: Elemental concentrations (X) of the different groups of water outflow from the Séchilienne massif as a function of Cl concentrations; a. K *vs*. Cl, b. Na *vs*. Cl, c. NO$_3$ *vs*. Cl, d. SO$_4$ *vs*. Cl.



## Appendix D: Solving the mixing equations

*Model set-up and Sr fractions*

Given the discussion of section 5.1 and the geochemical mixing diagrams presented in Fig. 4, three end members can be identified to release solutes to the springs of the Séchilienne area: silicate weathering (*sil*), carbonate weathering (*carb*; including both calcite and dolomite), and gypsum dissolution (*gyps*). The contribution of each of these processes to the solute load of the springs can be estimated using a combination of mixing equations, provided that geochemical tracers that are conservative during mixing of compositionally different waters can be identified, and that the composition of the end members

can be assessed. He we use two such tracers: $^{87}Sr/^{86}Sr$ and Na/Sr ratios (all corrected from rain and anthropogenic inputs, as explained in the main text). Indeed, the $^{87}Sr/^{86}Sr$ ratio is not affected by isotope fractionation due to the way data are reduced after measurements by mass spectrometry; and both Na and Sr are soluble elements unlikely to be scavenged into secondary solids such as clays in most contexts. Two mixing equations, each based on one of the two tracers, can be combined with a third summation equation to solve for $X_{sil}^{Sr}$, $X_{carb}^{Sr}$, and $X_{gyps}^{Sr}$, the relative contribution of the three identified end member to

dissolved Sr:

$$\left(\frac{^{87}Sr}{^{86}Sr}\right)_{spring} = X_{sil}^{Sr}\left(\frac{^{87}Sr}{^{86}Sr}\right)_{sil} + X_{carb}^{Sr}\left(\frac{^{87}Sr}{^{86}Sr}\right)_{carb} + X_{gyps}^{Sr}\left(\frac{^{87}Sr}{^{86}Sr}\right)_{gyps} \tag{D1}$$

$$\left(\frac{Na}{Sr}\right)_{spring} = X_{sil}^{Sr}\left(\frac{Na}{Sr}\right)_{sil} + X_{carb}^{Sr}\left(\frac{Na}{Sr}\right)_{carb} + X_{gyps}^{Sr}\left(\frac{Na}{Sr}\right)_{gyps} = X_{sil}^{Sr}\left(\frac{Na}{Sr}\right)_{sil} \tag{D2}$$

$$X_{sil}^{Sr} + X_{carb}^{Sr} + X_{gyps}^{Sr} = 1 \tag{D3}$$


The simplification of eq. (D2) is made possible by the fact that the gypsum and carbonate end members can be assumed to be devoid of Na. This assumption is supported by the positions of the different springs in a Na/Sr *vs.* $^{87}Sr/^{86}Sr$, which indicates that the low-$^{87}Sr/^{86}Sr$ component, encompassing both carbonate and gypsum weathering of the springs has a negligible Na content, with the exception of spring S12 which we do not treat quantitatively here (Fig. D1).


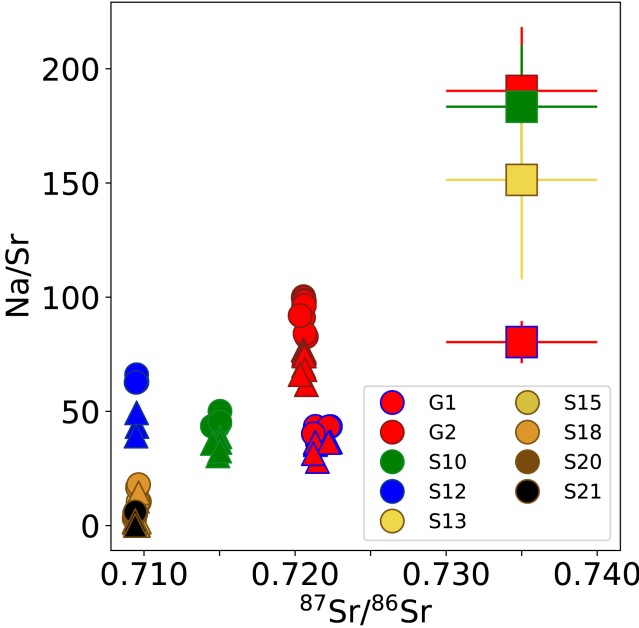

Figure D1: Na/Sr vs $^{87}$Sr/$^{86}$Sr for the different groups of water outflows (circles represent spring water composition corrected for rain and anthropogenic inputs (eq. 1); triangles represent spring water composition corrected for gypsum dissolution (eq. D5)) sampled in the Séchilienne massif. The inferred composition of the silicate end member for each group is shown as squares


The composition of the end members was constrained as follows (values also provided in Table D1):

- for each spring, $\left(\frac{^{87}Sr}{^{86}Sr}\right)_{spring}$ and $\left(\frac{Na}{Sr}\right)_{spring}$ were taken to be equal to the average ratios (corrected from rain inputs for Na/Sr) over all measurements available for the spring (note that one outlier were dismissed for each of the two springs G1 and G2);

- $\left(\frac{^{87}Sr}{^{86}Sr}\right)_{carb}$ was taken between 0.7090 and 0.7095 or 0.7095 and 0.7105, depending on whether the carbonate end member was assumed to be of calcitic or dolomitic nature. These ranges were constrained based on our own geochemical analyses of rock samples (Tab. A1). Note that the first range is consistent with what is known of early Jurassic seawater Sr isotope composition (MacArthur and Howarth, 2004). $\left(\frac{^{87}Sr}{^{86}Sr}\right)_{gyp}$ was taken between 0.7075 and 0.7080, which is typical of Triassic seawater, and in agreement with values reported for Alpine

gyspum by previous studies (Kloppmann et al., 2017).

- $\left(\frac{^{87}Sr}{^{86}Sr}\right)_{sil}$ was assumed to be equal to the highest measured Sr isotope ratio throughout the local bedrock samples ("micaschist", $0.735 \pm 0.005$; Tab. A1).

- The $\left(\frac{Na}{Sr}\right)_{sil}$ ratio was determined for the springs the most affected by silicate weathering (G1, G2, and S10) by extrapolating Na/Sr *vs.* $^{87}Sr/^{86}Sr$ relationships (Fig. D1) to a $^{87}Sr/^{86}Sr$ equal to that of the silicate end member (0.735, see above). Linear extrapolation in this diagram is made possible by the fact that Sr concentration is present on the denominator of both ratios plotted. For each spring, a set of lines was determined, each passing through (1) the carbonate end member ($^{87}Sr/^{86}Sr = 0.7090\text{-}0.7105$ and Na/Sr = 0); (2) one of the samples collected for this spring. After extrapolation of these lines to $^{87}Sr/^{86}Sr = 0.735$ (our central estimate of $\left(\frac{^{87}Sr}{^{86}Sr}\right)_{sil}$, see above), a collection of estimated $\left(\frac{Na}{Sr}\right)_{sil}$ ratios was therefore obtained for each spring, and the average and standard deviation of these ratios was used as spring-specific central estimate and associated uncertainty of the $\left(\frac{Na}{Sr}\right)_{sil}$ ratio, respectively. Note that following the discussion on Cl sources in section 5.1.1 of the main text, for each of the springs G1, G2, and S10 two estimations of $\left(\frac{Na}{Sr}\right)_{sil}$ were performed: one based on Na concentrations corrected from rain inputs only following eq. (1) ([$Na*$]), and the other based on Na concentrations from which an amount equivalent to total Cl concentration was subtracted (hence assuming that the entirety of Cl⁻ release to the springs was associated with Na⁺ release). The average between these two estimates was then used for further calculations. The obtained $\left(\frac{Na}{Sr}\right)_{sil}$ values range from $80 \pm 9$ to $205 \pm 30$ mol/mol, lower than those classically estimated for silicates (e.g. Négrel et al., 1993) but consistently with the fact that the parent rock at Séchilienne is a micaschist, that is a sedimentary rock that has lost soluble elements such as Na in previous weathering episodes compared to igneous silicates. For the springs the least affected by silicate weathering (S12, S13, S15, S18, S20, and S21), as such extrapolation would lack precision, we simply took the average and standard deviation of the $\left(\frac{Na}{Sr}\right)_{sil}$ ratios estimated for springs G1, G2, and S10 (yielding $162 \pm 40$). This strong assumption does not bear consequence on our overall evaluation as these former springs are not significantly affected by silicate weathering anyway.

The uncertainty on the different input parameters was propagated to the output variables ($X_{sil}^{Sr}$, $X_{carb}^{Sr}$, and $X_{gyp}^{Sr}$) using a Monte Carlo method based on 10,000 iterations. For this, at each of the 10,000 iterations a value for each input parameter (end member composition) was randomly picked following either (a) a normal distribution with mean equal and standard deviations to the estimate and uncertainty provided above, respectively, for parameters $\left(\frac{^{87}Sr}{^{86}Sr}\right)_{sil}$ and $\left(\frac{Na}{Sr}\right)_{sil}$; and (b) a uniform distribution bounded by the lowest and highest estimates provided above for parameters $\left(\frac{^{87}Sr}{^{86}Sr}\right)_{carb}$ and $\left(\frac{^{87}Sr}{^{86}Sr}\right)_{gyp}$. The choice of a uniform distribution to reflect uncertainty on the latter parameters is justified by the fact that it is highly unlikely that Sr isotope ratios

of marine carbonates lie outside of the prescribed range, which is unavoidable to occur for a significant number of random draws if one were to be used a normal distribution with a reasonable standard deviation instead. Note that, generally speaking, the composition of the springs was not considered as a random variable here and was simply fixed for all iterations. However, in order to account for uncertainty on the Cl⁻ source to the spring waters (main text section 5.1.1), for each spring and for each simulation we randomly drew values of $\left(\frac{Na}{Sr}\right)_{spring}$ following a uniform distribution bounded by Na concentrations given by eq. (1) ([$Na^*$]) on the one hand, and by Na concentrations from which an amount equivalent to total Cl⁻ concentration was subtracted (hence assuming that the entirety of Cl⁻ release to the springs was associated with Na⁺ release) on the other hand. Finally, Monte Carlo runs yielding $X_{sil}^{Sr}$, $X_{carb}^{Sr}$, and $X_{gyp}^{Sr}$ values outside of the range [0,1] were dismissed (the number of "valid" iterations is reported in Table D2).

The resulting distributions of output $X_{sil}^{i}$ values are reported in Table D2 as their 16th, 50th (median), and 84th percentile. Median estimates of $X_{sil}^{Sr}$ range from 0.06 and 0,08 (springs S15 and S18) to 0.42 and 0.47 (springs G2 and G1, respectively); median estimates of $X_{carb}^{Sr}$ range from 0.04 and 0.07 (springs S15 and S18) to 0.36 (spring S10); and median estimates of $X_{gyp}^{Sr}$ range from 0.25 and 0.29 for G1 and G2 to 0,85-0,90 for S18-S15. S10 median estimate is 0.42. Obtained values differed weakly depending on whether a calcitic or a dolomitic composition was used for $\left(\frac{^{87}Sr}{^{86}Sr}\right)_{carb}$ (Tab. D2).

*Gypsum contribution*

The contribution of each of these end members to the load of the dissolved major species SO₄, Ca, and Mg is necessary to evaluate the impact of the different weathering processes to the CO₂ budget of the Séchilienne area (Torres et al., 2016). In principle, these end member contributions to major dissolved species can be calculated from the corresponding contributions to dissolved Sr following:

$$X_i^E = X_i^{Sr} \left(\frac{E}{Sr}\right)_i / \left(\frac{E}{Sr}\right)_{spring} \tag{D4}$$

with $i = sil, carb,$ or $gyp$, and $E = $ SO₄ and Mg (corrected for rain and anthropogenic inputs). As explained below, because secondary carbonate formation might occur at Séchilienne and thus lead to the preferential scavenging of Ca from waters, eq. (D4) cannot be used indifferently for $E = $ Ca or for $i = carb$. We return to the estimation of $X_i^{Ca}$ values later. Regardless of this issue, eq. (D4) can first be used to calculate $X_{gyp}^{Mg}$ and $X_{gyp}^{SO4}$ provided that the $\left(\frac{E}{Sr}\right)_{gyp}$ ratios are known. The E/Sr ratios of the gypsum end member were determined using the E/SO₄ vs. (Ca+Mg)/SO₄ relationships described by the springs the least affected by silicate weathering (as identified by their low ⁸⁷Sr/⁸⁶Sr ratios, i.e. springs S13, S15, S18, S20, and S21; Fig. D2).

Indeed, for these springs where sulfate can be assumed to be entirely derived from gypsum and not from sulfide dissolution, at (Ca+Mg)/SO$_4$ of 1 mol/mol, the entirety of the dissolved load can be assumed to be derived from gypsum dissolution alone (Ca and Mg being the two major cations likely to be released by gypsum dissolution).

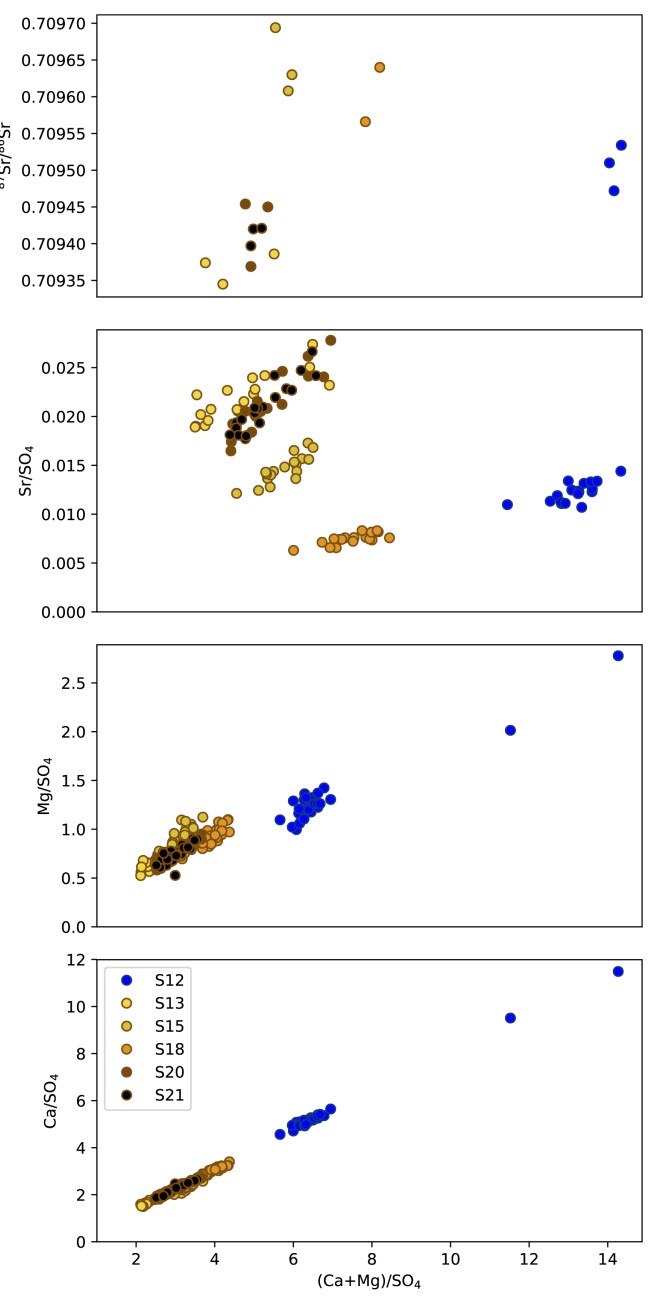

Figure D2: Mixing diagrams of the $^{87}$Sr/$^{86}$Sr ratio and the E/SO$_4$ (with E = Sr, Mg, Ca, and Na) molar ratio *vs.* the (Ca+Mg)/SO$_4$ molar ratios, used to determine the E/Sr molar ratios of the gypsum end member.

Linear extrapolation in Fig. D2 is made possible by the fact that SO$_4$ concentration is present on the denominator of both ratios plotted. Taking the average values obtained through extrapolation of the observed trends to (Ca+Mg)/SO$_4$ = 1 mol/mol suggests 0.31 ± 0.04 and 9.2 10$^{-3}$ ± 4.4 10$^{-3}$ mol/mol for the Mg/SO$_4$ and Sr/SO$_4$ ratios of the gyspum end members. From these two SO$_4$-normalized ratios, Sr-normalized ratios can be obtained to calculate $X_{gyp}^{Mg}$ and $X_{gyp}^{SO4}$ using eq. (D4). Obtained modal values of $X_{gyp}^{Mg}$ are lower than 0.03 for springs G1-G2-S10, and around 0.20-0.40 for the other springs. Modal values of $X_{gyp}^{SO4}$ are lower than 0.05 for springs G1-G2-S10, and range between 0.5 and 0.75 for the other springs

As the SO$_4$/Sr ratio of the carbonate end member was assumed to be equal to 0, consistently with the mixing diagrams of Fig. 4, $X_{sil}^{SO4}$ could be calculated as 1 - $X_{gyp}^{SO4}$ (eq. D3). Consequently, a value for the SO4/Sr ratio of the silicate end member could be estimated using the SO4/Sr mixing equation (similar to eqs. (D1-D2)):

$$\left(\frac{SO_4}{Sr}\right)_{spring} = X_{sil}^{Sr}\left(\frac{SO_4}{Sr}\right)_{sil} + X_{gyp}^{Sr}\left(\frac{SO_4}{Sr}\right)_{gyp} \tag{D5}$$

To solve eq. (D5), the average and standard deviation of all measurements of $\left(\frac{SO_4}{Sr}\right)_{spring}$ were used, thereby yielding a spring-specific value for $\left(\frac{SO_4}{Sr}\right)_{sil}$. This estimate was performed during the Monte Carlo iterations described above, yielding modal values ranging from 666 (spring S20) to 1845 mol/mol (spring G2).

*Secondary carbonate formation and carbonate contribution*

The precipitation of secondary carbonates is known to preferentially scavenge dissolved Ca$^{2+}$ at the expense of other alkali-earth ions Mg$^{2+}$ and Sr$^{2+}$ (Bickle et al., 2015). As a consequence, Ca does not necessarily behave conservatively in carbonate-rich settings such as Séchilienne, where secondary carbonate precipitation is likely to occur. We quantify the role of secondary carbonate formation using the method proposed by Bickle et al. (2015), which is based on the comparison between the chemical composition of springs and that predicted from conservative mixing between the rock dissolution end members. In a first set of attempts we assumed that secondary carbonate precipitation at Séchilienne was affecting only waters and solutes derived from two of the three rock dissolution end members (*i.e.* silicate and carbonate rocks, silicate rocks and gypsum, or carbonate rocks and gypsum). The Rayleigh-type equation proposed by Bickle et al. (2015) links dissolved E/Ca ratios to the extent of Ca scavenging by the formation of secondary carbonates:

$$\left(\frac{E}{Ca}\right)_p = \left(\frac{E}{Ca}\right)_0 \gamma^{K_d^E - 1} \tag{D6}$$

where $\left(\frac{E}{Ca}\right)_0$ is the "initial" E/Ca ratios of waters (*i.e.*, before secondary carbonate precipitation has taken place) with E = Sr,

Mg, or Na, $\left(\frac{E}{Ca}\right)_p$ is the E/Ca ratios of waters after secondary carbonate precipitation has taken place, $\gamma$ is the amount of "initial"

Ca left in solution after secondary carbonate precipitation has taken place, and $K_d^E$ is the partition coefficient of element E into

calcite, defined as the ratio between the E/Ca ratio into the precipitating calcite and that in water. During our Monte Carlo

simulations, we randomly drew values for $K_d^{Sr}$ and $K_d^{Mg}$ following a uniform distribution between 0.02 and 0.10, and

considered that $K_d^{Na} = 0$. As explained by Bickle et al. (2015), results in terms of source apportionment do not depend strongly

on the exact values chosen for the $K_d^E$ parameters. In a scenario where only two rock dissolution end members contribute Ca

involved into secondary carbonate precipitation, $\gamma$ is the solution of (Bickle et al., 2015):

$$\left(\frac{E}{Ca}\right)_p \gamma^{1-K_d^E} = A + B\gamma \left(\frac{Na}{Ca}\right)_p \tag{D7}$$


where $A$ and $B$ are the intercept and slope, respectively, of the binary mixing line between the two rock dissolution end

members in the E/Ca-Na/Ca space, with E = Mg or Sr. Note that eq. (D7) is valid because Na is not significantly incorporated

into secondary carbonates (in other words, $K_d^{Na}$ is assumed to be equal to 0). In Fig. D3, we show the results of such a scenario

where gypsum-derived solutes are thought to be added to waters derived from combined silicate and carbonate weathering

only after these two combined end members have been affected by secondary carbonate precipitation Such a scenario is

compatible with the formation of waters dominated by silicate and carbonate weathering "locally" at the Séchilienne site, to

which solutes derived from gypsum dissolution are admixed after their transport from a distal recharge location, for example

through fluid circulation along the Sabot fault. In this scenario the $\left(\frac{E}{Ca}\right)_p$ and $\left(\frac{Na}{Ca}\right)_p$ ratios in eq. (D7) are those of the spring

waters once corrected from gypsum inputs. However, this approach fails to account for the observed spring data as the $\gamma$ values

retrieved from the Sr/Ca-Na/Ca and Mg/Ca-Na/Ca differ greatly (Fig. 3D), regardless of whether the carbonate end member

is assumed to be of calcitic or dolomitic nature, or any combination thereof. Similar unsatisfactory results were obtained from

different assumptions regarding the two rock dissolution end members contributing Ca involved in secondary carbonate

precipitation.


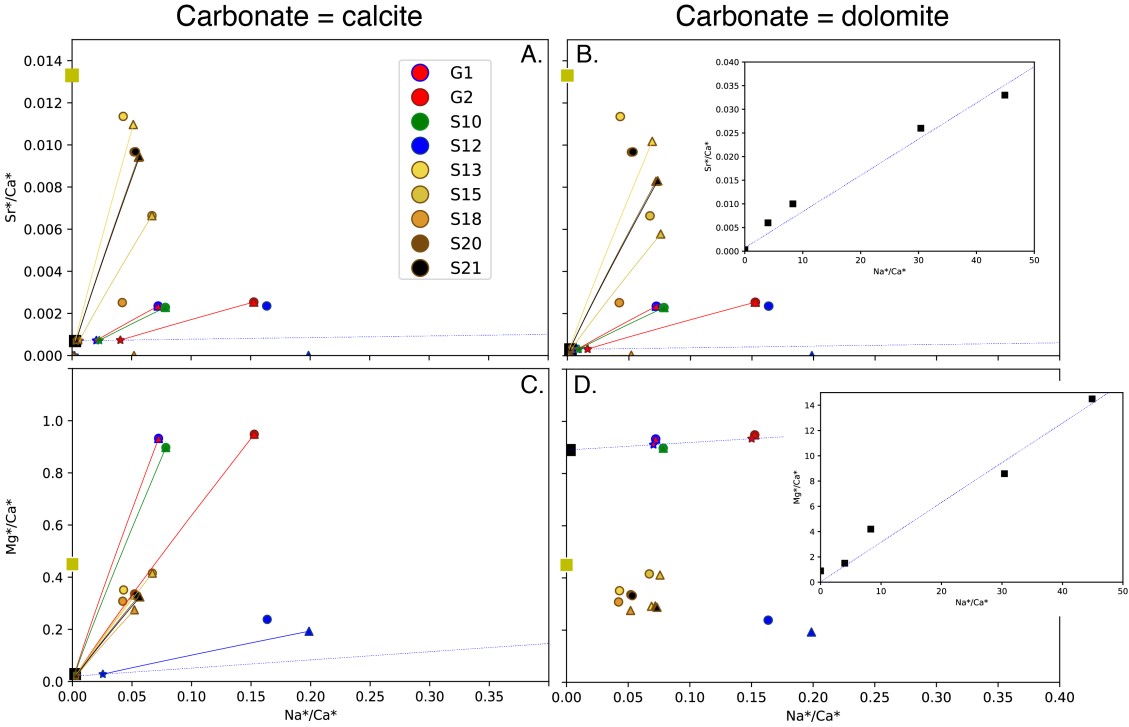


Figure D3: Quantification of the extent of secondary carbonate precipitation on dissolved Ca scavenging, assuming that secondary carbonate precipitation affects only waters derived from the combined weathering of silicate and carbonate rocks, and that gypsum-derived waters are added once secondary carbonate precipitation has occurred. Circles represent spring water composition corrected for rain and anthropogenic inputs (eq. 1); triangles represent spring water composition corrected for gypsum dissolution (eq. D5, which under this assumption can be

used with X = Ca and $i$ = gyps, since gypsum-derived solutes are assumed here to be simply added in a conservative mixing process); and stars represent spring water composition corrected for secondary carbonate precipitation (hence resulting from conservative mixing between the two rock dissolution end members) using the method of Bickle et al. (2015) (eq. D7). The colored lines link the two latter spring compositions and thus correspond to the path followed by the water composition upon dissolved Ca scavenging, such that the length of the line reflects the extent of Ca scavenging as quantified by $\gamma$ (eqs. D6 and D7). The large squares correspond to the composition of the gypsum

(yellow) and carbonate (black) dissolution end members. The blue dotted line reflects the binary mixing between waters derived from carbonate dissolution and waters derived from silicate dissolution, as constrained by our geochemical analyses of rocks samples (Tab. A1), on which spring composition must lie after correction from the effect of secondary carbonate precipitation. Panels A-B and C-D display the Sr/Ca-Na/Ca and Mg/Ca-Na/Ca sub-compositional spaces, respectively. Panels A-C and B-D show the results obtained if the carbonate end member is assumed to be of calcitic and dolomitic composition, respectively (note in particular the higher Mg/Ca ratio of the carbonate end

member in panels D, and the corresponding absence of lines linking triangles and stars, meaning that no Ca scavenging needs to take place through secondary carbonate precipitation to explain the data of springs G1, G2, and S10; and that the data of other springs cannot be explained by this scenario (2) if the composition of the carbonate end member is dolomitic). Inset in panels C and D show the same sub-compositional spaces but with a larger range of ratios, allowing for displaying the composition of rock samples Tab. A1 as black squares. In these insets, the high-Na/Ca data points correspond to silicate samples. To obtain these results, all spring Cl- was assumed to be associated

to Na+ (main text section 5.1.1), and $K_d^{Sr}$ was assumed to be equal to $K_d^{Mg}$ with a value of 0.06 was used. Modification of these assumptions

would not change the general principles that are exemplified by this figure; and uncertainty on these aspects is taken into account in our Monte Carlo simulations.

As a consequence, we assume that secondary carbonate precipitation affects solutes derived from the three weathering sources (carbonate, silicate, and gypsum dissolution) after they are mixed. Such scenario probably does not reflect perfectly the reality of the interplay between water mixing and chemical reactions in the subsurface at Séchilienne, but account for the fact that solutes from the three rock dissolution end members are mixed in a complex porous media, while keeping this problem mathematically tractable.

In such scenario, the contribution of neither rock end member can be corrected before secondary carbonate formation is taken into account. To that effect, the method of Bickle et al. (2015) can be extended to its use in the higher-dimensional Sr/Ca-Mg/Ca-Na/Ca space. In such space, the mixing array between carbonate, silicate, and gypsum dissolution is a plane, to which the composition of the springs can be "brought back" as in eq. (D7):

$$\alpha \gamma^{1-\kappa_d^{Mg}} \left(\frac{Mg}{Ca}\right)_p + \beta \gamma^{1-\kappa_d^{Sr}} \left(\frac{Sr}{Ca}\right)_p + \varepsilon \gamma \left(\frac{Na}{Ca}\right)_p = \delta \tag{D8}$$

where $\alpha$, $\beta$, $\varepsilon$, and $\delta$ are the parameters of the Cartesian equation of the mixing plane in the Sr/Ca-Mg/Ca-Na/Ca space. The ternary mixing relationships in the Sr/Ca-Mg/Na-Na/Ca space were constrained for Séchilienne using our own geochemical analyses of rock samples (Tab. A1) and are shown in Fig. D4. These relationships thus allow us to solve numerically eq. (D8) for $\gamma$ for each spring. We performed this calculation for various values of the relative contribution of dolomite dissolution to the overall Ca released to solution by carbonate weathering (from 0% to 100% dolomite, the rest being delivered by calcite dissolution; the two extreme cases are illustrated in Fig. D4), and constrained the chemical ratios of each of the carbonate components by our own geochemical analyses of rock samples (Tab. A1). We observed as a general trend that increasing the relative contribution of the dolomite end member beyond 20% resulted in an extent of alkalinity consumption through carbonate precipitation that was too large compared to alkalinity production to be sustainable. As a consequence, we fixed the relative contribution of dolomite to the overall Ca release by carbonate dissolution to 10% in the following calculations.

With these constraints, across the Monte Carlo simulations median estimates of $\gamma$ are 0.38 for G1, 0,39 for G2, 0,40 for S10 and 0,90 for S15. For the other springs, $\gamma$ was set to 1 as no Ca scavenging was required to explain the data - in other words, for these springs the hydrochemical data approximately lie on the mixing plane between the three rock dissolution end members in the Sr/Ca-Mg/Ca-Na/Ca compositional space.

The $\gamma$ value found for each spring can then be used to provide a value for the dissolved Sr/Ca, Mg/Ca, and Na/Ca ratios before the carbonate precipitation has taken place $\left(\frac{E}{Ca}\right)_0$ using eq. (D6) with the measured spring ratios as values for $\left(\frac{E}{Ca}\right)_p$. Then, $X_{carb}^{Ca}$ and $X_{carb}^{Mg}$ can be estimated from eq. (D4) for $i = carb$, but using the above-inferred Ca/Sr and Mg/Sr ratios (calculated from Sr/Ca and Mg/Ca ratios) before secondary carbonate precipitation as values of $\left(\frac{E}{Sr}\right)_{spring}$. $X_{sil}^{Ca}$ and $X_{sil}^{Mg}$ are then

determined by difference using eq. (D3) with Ca or Mg instead of Na, Finally, knowing the composition of the carbonate end member (as constrained by our geochemical analyses on rock samples; Tab. A1), values for $\left(\frac{Ca}{Sr}\right)_{sil}$ and $\left(\frac{Mg}{Sr}\right)_{sil}$ can be determined using eq. (D2) with Ca and Mg instead of Na, respectively, as in scenario (2).

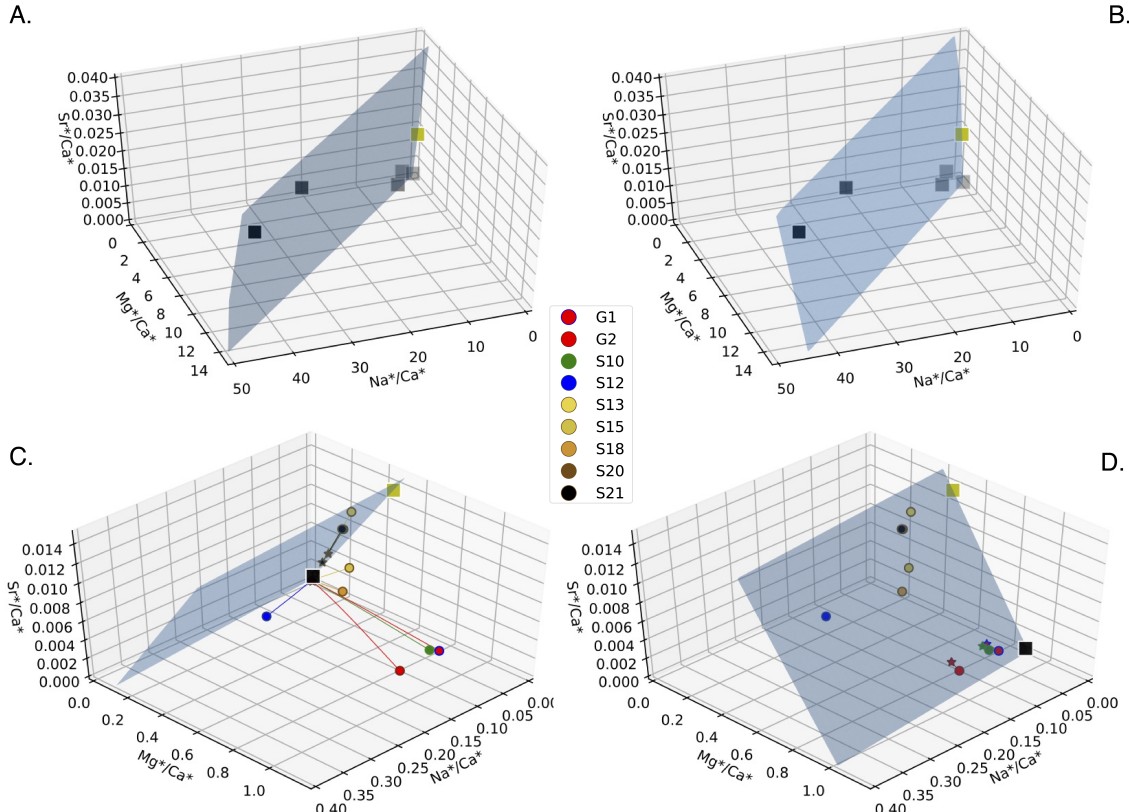

Figure D4: Quantification of the extent of secondary carbonate precipitation on dissolved Ca scavenging, assuming that secondary carbonate precipitation affects waters once they have been mixed from the three rock dissolution end members (silicate, carbonate, and evaporites). Circles represent spring water composition corrected for rain and anthropogenic inputs (eq. 1) and stars represent spring water composition corrected for secondary carbonate precipitation (hence resulting from conservative mixing only) using the method of Bickle et al. (2015), but here extended to the three dimensional Sr/Ca-Mg/Ca-Na/Ca space (eq. D8). The blue plane reflects the ternary mixing between waters derived from carbonate, silicate and evaporite dissolution, as constrained by our geochemical analyses of rocks samples (Tab. A1), on which spring composition must lie after correction from the effect of secondary carbonate precipitation. The colored lines link the two water compositions (before and after secondary carbonate precipitation) and thus correspond to the path followed by the water composition upon dissolved Ca scavenging, such that the length of the line reflects the extent of Ca scavenging as quantified by $\gamma$ (eqs. D6 and D8). The large squares correspond to the composition of the gypsum (yellow) and carbonate (black) dissolution end members. Panels A-B and C-D show the same results but with different scales. In particular, water compositions are made visible only in panels C and D which are "zoom-in" on the low-Sr/Ca, low-Mg/Ca, and low-Na/Ca part of the domains represented in panels A and B. Panels A-C and B-D correspond to results obtained if the carbonate end member is assumed to be calcitic and dolomitic, respectively. Note in particular the higher Mg/Ca ratio of the

carbonate end member in panels B and D, and the resulting "tilting" of the mixing plane (and the corresponding absence of lines linking triangles and stars, meaning that no Ca scavenging needs to take place through secondary carbonate precipitation to explain the data of springs G1, G2, and S10; and that the data of other springs cannot be explained by this scenario if the composition of the carbonate end member is dolomitic). To obtain these results, all spring Cl⁻ was assumed to be associated to Na⁺ (main text section 5.1.1), and $K_d^{Sr}$ was assumed to be equal to $K_d^{Mg}$ with a value of 0.06 was used. Modification of these assumptions would not change the general principles that

are exemplified by this figure; and uncertainty on these aspects is taken into account in our Monte Carlo simulations.

To assess the validity of our findings from the mixing model using independent measurements, our $\delta^{34}S$ measurements can be used. A significant linear negative relationship ($R^2 = 0.8$) exists between the $\delta^{34}S$ measured in springs across the Séchilienne massif and the modal estimates of their $X_{sil}^{SO4}$, consistent with the isotope composition of sulfur being driven by a binary mixture (Fig. D5). The intercept of this relationship ($X_{sil}^{SO4} = 0$, equivalent to $X_{gyp}^{SO4} = 1$) yields an estimate for $\delta^{34}S_{gyp} = 8.4‰$,

while extrapolation to $X_{sil}^{SO4} = 1$ indicates $\delta^{34}S_{sulfur} = -3.1‰$. Such estimates are fully consistent with our own measurements of solid sulfur at Séchilienne, as well as with reported measurements for Triassic seawater where local gypsum might have formed.

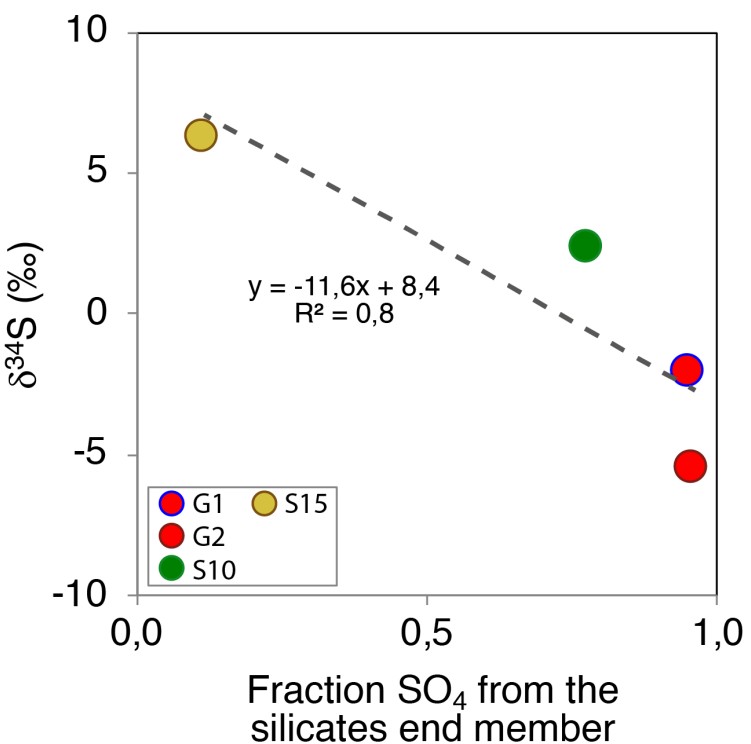


Figure D5: $\delta^{34}S$ measured in springs across the Séchilienne massif vs modal estimates of their fraction of SO₄ from silicate endmember: $X_{sil}^{SO4}$.

The mixing analysis presented above shows that at Séchilienne solutes are released to springs through the partial dissolution of silicate minerals, carbonate minerals, and gypsum, while dissolved Ca is scavenged by secondary carbonate precipitation. Fundamentally, these weathering reactions modify the carbon content in the atmosphere-hydrosphere continuum, which impacts the $CO_2$ concentration in the atmosphere. An approach to evaluate this impact has been recently proposed by Torres et al. (2016), based on the relative change in alkalinity (Alk) and Dissolved Inorganic Carbon (DIC) in ambient waters resulting

from the combination of weathering reactions. Torres et al. (2016) contend that the relevant Alk/DIC ratio against which the shifts in Alk and DIC ensuing weathering reactions have to be evaluated is that of the ocean. However, in the case of Séchilienne, and in particular because secondary carbonate precipitation is a significant process there, this approach cannot be used. Indeed, secondary carbonate precipitation removes Alk and DIC from ambient waters following the reaction:

$Ca^{2+} + 2HCO_3^- \rightarrow CaCO_3 + H_2CO_3$                                (D9)

   And the ensuing change in Alk and DIC occurs at the loci of secondary carbonate precipitation (i.e. at Séchilienne, in the subsurface of the instability), the Alk/DIC of which is not known. However, we still believe that it is essential to assess the long-term (that is, once weathering-derived solutes are delivered to the oceans) effects of these weathering reactions on

atmospheric $CO_2$, especially in contexts where sulfide oxidation is a significant process, such as at Séchilienne.

   As a consequence, we assess the $CO_2$ impact of the various weathering reactions at Séchilienne in the following way:

- For silicate weathering by carbonic acid (eq. 9 of main text), each meq of cation released leads to the consumption of 1 mole of $CO_2$ "on site", meaning immediately when the reaction takes place; and we assume that all meq of cation released are precipitated in the ocean on the "long term" through equation (D9), leading in total to the net consumption

of 0.5 mole of $CO_2$ per meq initially released;

- Silicate weathering by sulfuric acid (eq. 10), has no "on site" net effect on atmospheric $CO_2$; and we assume that it has no net effect on the "long term" either once solutes are delivered to the ocean, based on the fact that this reaction does not produce any alkalinity that is required for carbonate precipitation to occur (eq. D9);

- Carbonate weathering by carbonic acid (eq. 11 of main text) consumes 0.5 mol of $CO_2$ per meq of cation ($Ca^{2+}$ or

$Mg^{2+}$) released "on site", but this effect is negated on the "long term" by carbonate precipitation in the ocean (eq. D9; note that here "long term" refers to a time scale longer than the characteristic time scale for carbonate precipitation in the ocean (around $10^5$ yrs), but shorter that the characteristic time scale for sulfate reduction in the ocean (around $10^7$ yrs; Torres et al., 2016));

- Carbonate weathering by sulfuric acid (eq. 12 of main text) has no "on site" net effect on atmospheric $CO_2$; but is

likely to lead on the long term to the release of 0.25 mol of $CO_2$ (eq. D9) for each meq of cation ($Ca^{2+}$ or $Mg^{2+}$) released initially.

- Secondary carbonate formation happens only "on site" by definition, and leads to the release of 0.5 mol of $CO_2$ for each meq of cation ($Ca^{2+}$) precipitated (eq. D9).

Given that we do not have access to water discharge at Séchilienne, this analysis cannot be performed on a flux basis. Therefore, we use a concentration-based analysis, where the $CO_2$ effect of each process is compared within each spring in a consistent manner and expressed as the consumption or release of $CO_2$ in mol per liter of water (Fig. 7).

We performed this analysis on all springs but spring S12, for which the significant influence of anthropogenic activities precludes quantitative apportionment by the methods outlined above. In addition, in the cases of springs S20 and S21 none of the 10,000 Monte Carlo runs were able to yield $X_i^E$ values in the interval [0,1] for all $i$ and E. This is probably because their hydrochemistry can be readily explained by a simple binary mixture between gyspum and carbonate-derived solutes, such that any small variation in the composition of these end members result in negative $X_{sil}^E$ values for one or more instances of the element E. However, because the hydrochemistry of springs S20 and S21 is similar to those of S13 and S15, we expect that their role on atmospheric $CO_2$ is similar to that played by springs S13 and S15 (Fig. 7).

Table D1: Inputs of the mixing model with silicate, carbonate and gypsum endmembers

| Outflow | \textsuperscript{87}Sr/\textsuperscript{86}Sr | SO4/Sr | Mg/Sr | Ca/Sr | Na/Sr | Na/Sr corrected | Sr/Ca | Mg/Ca | Na/Ca | \textsuperscript{87}Sr/\textsuperscript{86}Sr Mean | s.d. | Na/Sr Mean | s.d. |
|---|---|---|---|---|---|---|---|---|---|---|---|---|---|
| | | | | Spring average composition | | | | | | | Silicate end member | | |
| G1 | 0,72155 | 571 | 418 | 445 | 43 | 32 | 0,0024 | 0,93 | 0,077 | 0,735 | 0,005 | 81 | 9 |
| G2 | 0,72058 | 786 | 397 | 411 | 90 | 62 | 0,0025 | 0,95 | 0,171 | 0,735 | 0,005 | 192 | 28 |
| S10 | 0,71489 | 196 | 400 | 438 | 46 | 36 | 0,0023 | 0,90 | 0,082 | 0,735 | 0,005 | 187 | 28 |
| S15 | 0,70964 | 69 | 64 | 150 | 10 | 1 | 0,0066 | 0,42 | 0,019 | 0,735 | 0,005 | 153 | 40 |
| S18 | 0,70960 | 135 | 126 | 398 | 17 | 12 | 0,0025 | 0,31 | 0,031 | 0,735 | 0,005 | 153 | 36 |

| Outflow | \textsuperscript{87}Sr/\textsuperscript{86}Sr Low | High | Sr/Ca Mean | s.d. | Mg/Ca Mean | s.d. | \textsuperscript{87}Sr/\textsuperscript{86}Sr Low | High | Sr/SO4 Mean | s.d. | Mg/SO4 Mean | s.d. | Ca/SO4 Mean | s.d. |
|---|---|---|---|---|---|---|---|---|---|---|---|---|---|---|
| | Carbonate end member | | | | | | Gypsum end member | | | | | | | |
| G1 | 0,7091 | 0,7097 | 0,00062 | 0,00002 | 0,19 | 0,02 | 0,7075 | 0,708 | 0,0092 | 0,004 | 0,31 | 0,04 | 0,69 | 0,04 |
| G2 | 0,7091 | 0,7097 | 0,00062 | 0,00002 | 0,19 | 0,02 | 0,7075 | 0,708 | 0,0092 | 0,004 | 0,31 | 0,04 | 0,69 | 0,04 |
| S10 | 0,7091 | 0,7097 | 0,00062 | 0,00002 | 0,19 | 0,02 | 0,7075 | 0,708 | 0,0092 | 0,004 | 0,31 | 0,04 | 0,69 | 0,04 |
| S15 | 0,7091 | 0,7097 | 0,00062 | 0,00002 | 0,19 | 0,02 | 0,7075 | 0,708 | 0,0092 | 0,004 | 0,31 | 0,04 | 0,69 | 0,04 |
| S18 | 0,7091 | 0,7097 | 0,00062 | 0,00002 | 0,19 | 0,02 | 0,7075 | 0,708 | 0,0092 | 0,004 | 0,31 | 0,04 | 0,69 | 0,04 |

Table D2: Results of mixing calculations


| Outflow | Fractions Sr ($X^{Sr}_i$) | | | | | | | | | Fractions Mg ($X^{Mg}_i$) | | | | | | | | | percentage of valid iterations |
| --- | --- | --- | --- | --- | --- | --- | --- | --- | --- | --- | --- | --- | --- | --- | --- | --- | --- | --- | --- |
| | Silicate | | | Carbonate | | | Gypsum | | | Silicate | | | Carbonate | | | Gypsum | | | |
| | $D_{50}$ | $D_{16}$ | $D_{84}$ | $D_{50}$ | $D_{16}$ | $D_{84}$ | $D_{50}$ | $D_{16}$ | $D_{84}$ | $D_{50}$ | $D_{16}$ | $D_{84}$ | $D_{50}$ | $D_{16}$ | $D_{84}$ | $D_{50}$ | $D_{16}$ | $D_{84}$ | |
| G1 | 0,47 | 0,42 | 0,52 | 0,28 | 0,09 | 0,46 | 0,25 | 0,08 | 0,43 | 0,75 | 0,63 | 0,88 | 0,22 | 0,07 | 0,36 | 0,02 | 0,01 | 0,05 | 10,2 |
| G2 | 0,42 | 0,37 | 0,47 | 0,29 | 0,11 | 0,48 | 0,29 | 0,10 | 0,47 | 0,73 | 0,58 | 0,86 | 0,24 | 0,09 | 0,40 | 0,02 | 0,01 | 0,05 | 8,9 |
| S10 | 0,23 | 0,20 | 0,26 | 0,36 | 0,13 | 0,55 | 0,42 | 0,23 | 0,64 | 0,67 | 0,53 | 0,83 | 0,28 | 0,10 | 0,43 | 0,03 | 0,02 | 0,07 | 20,9 |
| S15 | 0,06 | 0,05 | 0,07 | 0,04 | 0,02 | 0,07 | 0,90 | 0,87 | 0,92 | 0,43 | 0,15 | 0,56 | 0,19 | 0,08 | 0,33 | 0,41 | 0,28 | 0,55 | 0,4 |
| S18 | 0,08 | 0,06 | 0,09 | 0,07 | 0,02 | 0,13 | 0,85 | 0,79 | 0,90 | 0,56 | 0,37 | 0,73 | 0,19 | 0,05 | 0,34 | 0,22 | 0,15 | 0,38 | 6,2 |

| Outflow | Fractions Ca ($X^{Ca}_i$) | | | | | | | | | Fractions SO$_4$ ($X^{SO4}_i$) | | | | | | Fraction of Ca left in solution after secondary carbonate precipitation ($\gamma$) | | | percentage of valid iterations |
| --- | --- | --- | --- | --- | --- | --- | --- | --- | --- | --- | --- | --- | --- | --- | --- | --- | --- | --- | --- |
| | Silicate | | | Carbonate | | | Gypsum | | | Silicate (sulfur) | | | Gypsum | | | | | | |
| | $D_{50}$ | $D_{16}$ | $D_{84}$ | $D_{50}$ | $D_{16}$ | $D_{84}$ | $D_{50}$ | $D_{16}$ | $D_{84}$ | $D_{50}$ | $D_{16}$ | $D_{84}$ | $D_{50}$ | $D_{16}$ | $D_{84}$ | $D_{50}$ | $D_{16}$ | $D_{84}$ | |
| G1 | 0,55 | 0,28 | 0,82 | 0,42 | 0,13 | 0,71 | 0,02 | 0,01 | 0,04 | 0,95 | 0,9 | 0,99 | 0,05 | 0,01 | 0,1 | 0,38 | 0,37 | 0,39 | 10,2 |
| G2 | 0,48 | 0,17 | 0,78 | 0,48 | 0,18 | 0,81 | 0,02 | 0,01 | 0,05 | 0,96 | 0,92 | 0,99 | 0,04 | 0,01 | 0,08 | 0,39 | 0,38 | 0,40 | 8,9 |
| S10 | 0,40 | 0,12 | 0,74 | 0,56 | 0,20 | 0,85 | 0,03 | 0,01 | 0,06 | 0,78 | 0,59 | 0,88 | 0,22 | 0,12 | 0,41 | 0,40 | 0,39 | 0,41 | 20,9 |
| S15 | 0,19 | 0,05 | 0,42 | 0,37 | 0,15 | 0,66 | 0,35 | 0,24 | 0,50 | 0,12 | 0,03 | 0,2 | 0,88 | 0,8 | 0,97 | 0,90 | 0,89 | 0,90 | 0,4 |
| S18 | 0,40 | 0,12 | 0,67 | 0,37 | 0,10 | 0,67 | 0,18 | 0,13 | 0,29 | 0,43 | 0,21 | 0,57 | 0,57 | 0,43 | 0,79 | 1,00 | 1,00 | 1,00 | 6,2 |

| Outflow | Fractions total cationic charge ($X^{\Sigma+}_i$) | | | | | | | | | | percentage of valid iterations |
| --- | --- | --- | --- | --- | --- | --- | --- | --- | --- | --- | --- |
| | Silicate | | | Carbonate | | | Gypsum | | | | |
| | $D_{50}$ | $D_{16}$ | $D_{84}$ | $D_{50}$ | $D_{16}$ | $D_{84}$ | $D_{50}$ | $D_{16}$ | $D_{84}$ | | |
| G1 | 0,61 | 0,38 | 0,84 | 0,37 | 0,11 | 0,61 | 0,02 | 0,01 | 0,04 | | 10,2 |
| G2 | 0,56 | 0,31 | 0,81 | 0,40 | 0,15 | 0,68 | 0,02 | 0,01 | 0,05 | | 8,9 |
| S10 | 0,48 | 0,24 | 0,76 | 0,48 | 0,17 | 0,73 | 0,03 | 0,02 | 0,07 | | 20,9 |
| S15 | 0,29 | 0,10 | 0,45 | 0,33 | 0,13 | 0,56 | 0,36 | 0,29 | 0,48 | | 0,4 |
| S18 | 0,45 | 0,21 | 0,68 | 0,32 | 0,09 | 0,58 | 0,20 | 0,15 | 0,30 | | 6,2 |

*footnotes:*

- *All elemental ratios in mol/mol*
- *When only one value is provided (spring composition), the value of the parameter was not varied between*


  *different Monte Carlo runs. When two values are provided, in the case of a variable whose value is assumed to follow a uniform distribution "low" and "high" provide the lower and upper bounds of the interval over which the parameter was drawn randomly following a uniform distribution; and in the case of a variable that is assumed to follow a normal distribution "Mean" and "s.d." provide the mean and the standard deviation of the normal distribution from which the parameter was drawn.*


- *The spring elemental ratios reported here have been corrected from rain inputs*
- *"corrected" for spring Na/Sr ratios refers to values once an amount of Na equal to the amount of non-atmospheric Cl has been removed. The actual Na/Sr ratio used in each Monte Carlo run was randomly drawn between "Na/Sr" and "Na/Sr corrected"*


Table D3: Evaluation of the effect of weathering processes at Séchilienne on atmospheric $CO_2$ and $\delta^{34}S$ estimates of the pyrite endmember

| Outflow | "On-site" $CO_2$ consumption (mmol L-1) | | | "Long-term" $CO_2$ consumption (mmol L-1) | | | percentage of valid iterations | $\delta^{34}S$ | | |
|---|---|---|---|---|---|---|---|---|---|---|
| | $D_{50}$ | $D_{16}$ | $D_{84}$ | $D_{50}$ | $D_{16}$ | $D_{84}$ | | Sample | Gypsum | Silicate *(sulfur)* |
| G1 | -1,76 | -2,89 | -0,76 | 2,43 | 0,85 | 4,02 | 10,2 | -2,12 | 8,4 | -3,1 |
| G2 | 0,69 | 0,45 | 0,89 | 1,61 | 1,09 | 2,13 | 8,9 | -5,45 | 8,4 | -3,1 |
| S10 | -3,22 | -4,40 | -2,33 | 0,33 | -0,90 | 1,33 | 20,9 | 2,38 | 8,4 | -3,1 |
| S15 | -1,95 | -2,34 | -1,56 | -0,49 | -0,91 | -0,03 | 0,4 | 6,40 | 8,4 | -3,1 |
| S18 | -2,73 | -3,27 | -2,30 | -1,22 | -1,70 | -0,75 | 6,2 | | | |

**Team list**

Pierre Nevers

Julien Bouchez

Jérome Gaillardet

Thomazo Christophe

Charpentier Delphine

Laëticia Faure

Catherine Bertrand

**Author contribution**

Pierre Nevers measured the Sr isotopic ratios, worked on the interpretation and wrote text. Julien Bouchez and Jérôme Gaillardet helped with the isotopic measurements, in the inversion calculations and in writing text. Christophe Thomazo measured the sulfur isotopic ratios. Delphine Charpentier worked on XRD and mineralogic analyses presented in supplementary materials. Laëticia Faure helped with the rock sample leaching procedure and Sr isotope measurements. Catherine Bertrand organized the sampling strategy and is in charge of the Séchilienne Observatory (SNO OMIV) and helped in writing text.

## Competing interests

The authors declare that they have no conflict of interest.

## Acknowledgments

Part of this work was supported by IPGP multidisciplinary program PARI and by Region île-de-France SESAME Grant no. 12015908. Caroline Gorge, Christophe Loup and Caroline Amiot are acknowledged for analytical support, and Vanessa Stefani

is acknowledged for support on the field. The authors warmly thank: the PEA²t platform (Chrono environment, University of Bourgogne Franche-Comté, UMR CNRS 6249, France), which manages and maintains the analytical equipment used in this study; and OMIV and OSU Theta for supporting the project through financial contributions (SRO_2015).

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
