# Peer review of "Landslides as geological hotspots of CO2 emission: clues from the instrumented Séchilienne landslide, Western European Alps."

_Earth Surface Dynamics, 2020_

## Short Comment (SC1) · 6 Jul 2020

The manuscript by Nevers et al. entitled, "Landslides as geological hotspots of CO2 to the atmosphere: clues from the instrumented Séchilienne landslide, Western European Alps" reports new measurements of groundwaters emanating from a large landslide. In addition to major element concentrations, the authors also report measurements of the isotopic compositions of dissolved sulfate and strontium, which are used to help apportion the solute load between different lithologic sources. The solute source partitioning provides new insights into flow paths through the landslide as well as the effect of local chemical weathering on atmospheric CO2 levels. Overall, I think that this manuscript

is appropriate for publication in *Earth Surface Dynamics* after some minor revisions.

To me, one of the most exciting conclusions of this manuscript was that the detection of gypsum dissolution in some samples implied water flow through the Sabot faults. This result demonstrates that detailed hydrochemical measurements and mixing models can be used to improve our understanding of flow through the subsurface. The interpretation that some proportion of the solute load is sourced from gypsum dissolution also impacts the other main conclusion of the paper: that landslide weathering is a source of $CO_2$ to the atmosphere. Multiple previous papers have come to this same conclusion recently and they are appropriately highlighted in the Nevers et al. manuscript.

Given how important the detection of gypsum dissolution is to the Nevers et al. manuscript, most of my main comments concern the mixing model that the authors use to come to this conclusion. Firstly, I am slightly confused about the rainwater correction that the authors employ in this study. I get the concept of using chloride critical values, but I am slightly concerned that the assumption of spatially and temporally uniform evapo-transpiration is not appropriate for this specific study site. The data presented in Figure 3 show a wide range of chloride concentrations, including some below the chloride critical value. How does such a range arise if all samples are presumed to originate from rainwater with a constant concentration of Cl that undergoes the same amount of evapo-transpiration? It is also well known that rainwater element to Cl ratios can be significantly different than seawater element to Cl ratios especially for sulfate (e.g., Stallard & Edmond 1981), which is an important ion in the present study. The authors also acknowledge that some Cl remains after they perform the correction, which implies that there is an additional solute source. However, this additional source is not included in their subsequent mixing model, which is concerning to me as this might be a source of error.

The 3 component mixing model the authors utilize is based on Na, Sr, and Sr isotopic ratios as these are expected to behave more conservatively. I agree with the authors

about this and trust that their mixing model yields reasonable estimates for the fraction of Sr sourced from each end-member (with the caveat that they ignore the source of excess Cl as mentioned above). However, the extrapolation of the mixing results for Sr to Ca and Mg effectively assumes conservative mixing (Equation 8), which negates the whole motivation for performing the mixing calculations with Na and Sr alone.

The manner in which some of the end-member ratios are defined is concerning to me as there may be some statistical issues and/or circular logic applied. For some end-members, linear regressions of solute data are extrapolated to yield constraints on their characteristic elemental ratios. For example, the text on line 638 describes regressing element to sulfate ratios versus Ca+Mg to sulfate ratios. Since sulfate is the denominator for both ratios, there is potential for spurious correlation that might affect the best fit line and thus the calculated end-member ratio. Similarly, since the sulfate to Sr ratio of the silicate end-member is estimated from the data, the finding that the d34S correlates with the fractional contribution of S from this end-member might be forced to be the case as oppose to being an entirely independent test of the mixing model. Given the large range of d34S values observed in the solid-phase data, I am somewhat surprised that a single d34S value for sulfide mineral weathering can be applied to all of the water samples.

My recommendation would be for the to employ a more robust mixing model in their revised manuscript that (1) includes atmospheric deposition as a solute source (as oppose to performing a fixed correction beforehand), (2) includes an additional end-member besides atmospheric deposition that supplies Cl, and (3) uses all of the major elements and either allows for secondary mineral formation (e.g., Bickle et al. 2015) or assumes congruent weathering. This model should also test broader end-member ranges that rely less heavily on the linear regression approach. My concern is that the current way in which gypsum dissolution is being identified is not the most robust. Since detecting and quantifying gypsum contributions are so important to the main conclusions of the manuscript, I think that a more robust assessment of the data, either
through an improved mixing model or other technique, would greatly strengthen the manuscript.

**Minor comments:**

Line 111 - I do not know what "inertial circulation" means and would appreciate a brief definition here.

Line 395 - I am not sure that figure 5 is a histogram. Is it not a stacked bar plot? Also, one standard deviation of the mixing model results may not be a sufficient representation of the uncertainty. With the Monte-Carlo approach that is applied, there is no reason that one of the more "rare" solutions could not be the most accurate. Focusing on the median (and results nearby) will be sensitive to the assumption of a priori distribution shapes for each end-member.

Figure 7 - A more complete definition of R and Z in the figure caption would be useful.

**References**

Stallard, R. F. & Edmond, J. M. Geochemistry of the Amazon 1. Precipitation chemistry and the marine contribution to the dissolved load at the time of peak discharge. Journal of Geophysical Research 86, 9844–9858 (1981).

Bickle, M. J., Tipper, E. D., Galy, A., Chapman, H. & Harris, N. On discrimination between carbonate and silicate inputs to Himalayan rivers. American Journal of Science 315, 120–166 (2015).

---

## Referee Comment (RC1) · Xin Gu (Referee) · 7 Jul 2020

In this manuscript, Nevers and colleagues present major element chemistry and isotopes ratios (87Sr/86Sr, $\delta$34S) measured in different water types from a highly instrumented landslide (Séchilienne) in France. This study includes a comprehensive dataset and highlights the coupling of pyrite oxidation and carbonate dissolution. I feel the manuscript is interesting and fits the journal. But several points need be addressed or clarified to increase the strength of the paper:

The evidence from the rocks: Nevers and colleagues did a nice job by using the 87Sr/86Sr, $\delta$34S of the rocks to constrain the interpretation on dissolved species in

water. However, more data on chemical/minerology composition of rocks might be necessary, e.g. the pyrite and carbonate abundances. It might be interesting to know how deep is the reaction front of pyrite (if samples from several boreholes are available). Also, whether weathering is driven by carbonic or sulfuric acid (as shown in Fig. 5) is related to the pyrite and carbonate abundances in rock. Could gypsum be a weathering product of pyrite weathering? Since the isotope values were reported, I suspect the solid samples might be still available to make the analysis.

The discussion on hydrology: I like the authors' approach in section 5.2 on how this source identification may help to refine the hydrogeological model. But I feel the authors could discuss more about how the hydrological process may affect the chemistry of different water types. For example, is it possible that outflow S10 with low elemental concentrations represents an interflow, where pyrite has already been depleted in surrounding rocks and samples from G1 and G2 represent deeper groundwater where pyrite oxidation is occurring?

Some specific comments:

Line 35: I agree that silicate weathering by sulfuric acid does not directly influence atmospheric CO2, but I will argue it will reduce the potential for CO2 sequestration by silicate weathering.

Line 51: some references should be added to guide readers.

Line 278: The authors showed a more complicated mass balance approach later, then is the correlation from atmospheric input necessary here? The atmospheric input could be another endmember in the mixing model.

Line 364: I didn't find the label (a-d) in the figure. The gray bar in Fig. 4a (left upper panel) needs explanation.

Line 420: I love this figure. I think some quantitative results should be summarized in the abstract.

Line 460: It is better to use another color for the river

Line 535, 536: These two citations are not listed in references.

Line 538: In general, I agree with the authors. But the significance of such feedback really depends on the pyrite and carbonate abundances in bedrock.

Table A1. Are the dissolved oxygen data available? Given the importance of pyrite oxidation, such data might be interesting.

Figure C1: the y label is unreadable.

---

## Editor Comment (EC1) · Robert Hilton (Editor) · 15 Sep 2020

Dear authors,

Two experts in this field have now reviewed your manuscript. Apologies for the slight delay in the process, a third referee had initially accepted the task and I was awaiting their comments. However, I'm sure you understand the current pressures on people, and so I have decided we can proceed with the reviews in hand.

I agree with the referees that this study will be of interest to the readers at ESurf and provides new insights on this important theme. However, they highlight aspects which

need more work in a revised version.

Please prepare a revised manuscript that address the referee's comments (and provide a point-by-point reply). Overall, it seems moderate revisions are necessary, focused on: 1) more careful discussion of the assumptions that go into the use of Cl for the rain correction (R1); clarifying and better justifying choices of end member compositions used in the mixing analysis (R1); 3) assumptions of conservative nature of ions (R1+R2) and the role of secondary sulfides (R2); 4) clarifications on the rock samples – their bulk geochemistry and presence of key mineral phases (R2); 5) specifics of the hydrological pathways and associated reactions (R2). Please see the referee comments for more details.

I also completed my own review, prior to reading the referees comments, and in addition to the points they raise, I identified a few other comments/edits to address:

14 - Here we use a combination of major element chemistry. . ..

16 – the final two sentences here are very vague – it would be better to use this space to highlight some key results (or examples of being able to do what you say)

20 – Using a mixing model of XXXX(details), we are able to show. . ..

21 – where does it do this – in the failure itself? In the debris it creates? It would be useful to specify here.

23 – "but" => by?

26 – change "instable zones" to "large landslide complexes"

27 – instead of "physical and chemical erosion and climate", is it clearer to say "physical and chemical erosion and their impact on the carbon cycle and global climate"

36 – and indeed when sulfuric acid mixes with natural waters containing HCO3 at neutral pH or higher – this can release CO2.
38 – is this true (that carbonates are a minor fraction)? I think Hartmann's global maps show sedimentary rocks cover ∼65% of the earth;s surface, and I imagine that carbonates could make up a big chunk of that, especially considering interbedded carbonates and shales, and carbonate cement in siliciclastic rocks.

108 – consider splitting this sentence.

Figure 1 – can you show the cross section (d) location on b or c?

118 – can you explain briefly what the 'gallery' is – its not a term I've heard before, and other readers may not be familiar with it either

160 – leach. H2O not H20

179 – Sulfur

183 – typo

Figure 2 – add the notations to the figure legend so the readers can quickly see the water types (e.g. what is UZ BSZ etc.,)

Figure 4 – please add a,b,c,d labels to panels. Can carbonate weathering by sulfuric acid also be identified on part c? on part d, what does silicate end member mean for the x-axis (sulfur isotopes) – I guess pyrite? On d, what was the choice of S and Sr concentrations to make the mixing hyperbola?

---

## Author Comment (AC1) · 18 Nov 2020

Dear Dr. Hilton

We thank you for handling our submission, and we thank the reviewers for the quality of their comments. The manuscript will be thoroughly revised in order to address these comments. You will find in this letter a proposal for changes that we can make to improve our manuscript. The major modifications will be as follows:

- We will consider in more detail the origin of chloride in the different water bodies. At least two additional sources of chloride have been identified through further examination of the data, and these two sources will be included explicity in our mixing analysis (SC1).

- We will use a more realistic composition for the carbonate end member (in terms of Ca/Sr and Mg/Sr ratios) and include an explicit treatment of secondary carbonate precipitation in the inversion scheme based on the method introduced by Bickle et al. (2015) (SC1).

- We will add some discussion material about bedrock composition (including notably new data on major and trace elements on bulk rock, leacheates of the carbonate component, and residue) (SC1 and RC1)

- Further clarification will be provided on the specifics of the hydrological pathways and associated reactions such as the origin of gyspum-sourced solutes in waters, and on the oxidative weathering of the pyrite (RC1).

We believe that these modifications will significantly improve our manuscript, and we hope make it ultimately suitable for publication in E-surf.

Please find below the reviewer comments and our answers.

Answer to Reviewer #1.

- Firstly, I am slightly confused about the rainwater correction that the authors employ in this study. I get the concept of using chloride critical values, but I am slightly concerned that the assumption of spatially and temporally uniform evapotranspiration is not appropriate for this specific study site. The data presented in Figure 3 show a wide range of chloride concentrations, including some below the chloride critical value. How does such a range arise if all samples are presumed to originate from rainwater with a constant concentration of Cl that undergoes the same amount of evapotranspiration? It is also well known that rainwater element to Cl ratios can be significantly different than seawater element to Cl ratios especially for sulfate (e.g., Stallard & Edmond 1981), which is an important ion in the present study. The authors also acknowledge that

some Cl remains after they perform the correction, which implies that there is an additional solute source. However, this additional source is not included in their subsequent mixing model, which is concerning to me as this might be a source of error.

We agree with the reviewer that the way we were quantifying the contribution of non-rock sources to the spring cationic load can be improved. Indeed, some springs show a fairly high Cl concentration (> 100 $\mu$mol/L). A closer look at the data show different sources of chloride, one of them being clearly linked to release of nitrate and therefore to anthropogenic activities (agriculture, domestic input and/or use of road salt). This nitrate-rich component is particularly prominent in the stable part of the slope, where a couple of villages are present. In the unstable zone, an additional source of chloride is also found, but not linked to nitrate. In the most unstable zone of the landslide, we propose an additional source of Cl derived from the leaching of bedrock minerals, and thus to the release of Na or K.

To differentiate these different inputs of chloride, in the revised manuscript, we will include a new scenario in the inversion scheme, in addition to the previous scenario where Cl was just contributed by the atmosphere (subtraction of the critical Cl*): in this new scenario all Cl will associated with Na such that it will be necessary to subtract from the Na budget the whole Cl amount for each spring. This will obviously translate in additional uncertainty in the output parameters ("R" and "Z" parameters).

Regarding the reviewer's suggestion that we should rather "includes atmospheric deposition as a solute source (as oppose to performing a fixed correction beforehand)", we emphasize that this would not necessarily solve the issue of non-cyclic sources of Cl - we would still need to make a strong assumption about the X/Cl ratios of these sources. This is why we prefer to keep this "fixed correction beforehand", and to modify it to address the - fully valid - concern of the reviewer. We also note that the new scenario will place an "upper bound" on the correction that should be performed on Na concentrations, as some Cl could still be derived from other sources than NaCl. As a consequence, we believe that the combination of our previous scenario and of this

new one will constitute a sort of "sensitivity analysis" allowing us to explore the whole range of possibilities for the impact of Cl sources on the cationic load of the springs at Séchilienne.

The 3-component mixing model the authors utilize is based on Na, Sr, and Sr isotopic ratios as these are expected to behave more conservatively. I agree with the authors about this and trust that their mixing model yields reasonable estimates for the fraction of Sr sourced from each endmember (with the caveat that they ignore the source of excess Cl as mentioned above). However, the extrapolation of the mixing results for Sr to Ca and Mg effectively assumes conservative mixing (Equation 8), which negates the whole motivation for performing the mixing calculations with Na and Sr alone.

This is a fair point too. To address this comment, we will significantly modify our inversion scheme by: (1) using the actual composition of local carbonate rocks to convert Sr fractions into Mg and Ca fractions (based on new major and trace element data obtained on HCl and acetic acid leachates of local carbonate bedrock samples); (2) including an explicit, quantitative treatment for the precipitation of secondary carbonates based on the method of Bickle et al. (2015). To that effect, we will determine the local mixing array between the different rock type based on new major and trace element data obtained on local carbonate and silicate bedrock samples (acetic acid and HCl leachates, residues, and bulk rocks; see Figure 1 below). These major modifications will relax a strong assumption of the previously used model - namely that the composition of the carbonate end member was an "adjustment variable".

Figure 1: local mixing array (mixing line is the blue dotted line) between the different rock types (the black square corresponds to one of the rock samples used to constrain this mixing array; the other rock samples are not visible at this scale) in the Sr/Ca vs. Na/Ca space based on new major and trace element data obtained on local carbonate and silicate bedrock samples (including quantitative treatment for the precipitation of secondary carbonates based on the method of Bickle et al. (2015)). The colored circles correspond to the "raw" composition of springs, the colored triangles to their

composition corrected for gypsum inputs, and the colored stars to their composition corrected for secondary carbonate precipitation. The colored lines reflect the evolution of the water composition upon precipitation of secondary carbonates. Symbols colors have the same meaning as in the submitted manuscript.

- The manner in which some of the end-member ratios are defined is concerning to me as there may be some statistical issues and/or circular logic applied. For some endmembers, linear regressions of solute data are extrapolated to yield constraints on their characteristic elemental ratios. For example, the text on line 638 describes regressing element to sulfate ratios versus Ca+Mg to sulfate ratios. Since sulfate is the denominator for both ratios, there is potential for spurious correlation that might affect the best fit line and thus the calculated end-member ratio.

We are not sure to understand fully the reviewer's concern. As for the extrapolation of "linear regressions of solute data" generally speaking, we believe that this is a pretty conventional method yielding robust results when correctly performed (as was done by the reviewer himself and his co-authors in previous studies; see the determination of the rain end member in Torres et al., GCA, 2015). As for the specific case of the gypsum end member in our work, again we do not see where the issue is. Our rationale is as follows: (1) some springs are only very marginally affected by silicate inputs (hence, **in the case of Séchilienne** by sulfide inputs; see Fig. 4d of the previous version of the manuscript) as shown by their $87Sr/86Sr$ ratios < 0.710 (these are springs S12-S21); (2) data from these springs do form linear trends in X/ SO4 vs. (Ca+Mg)/SO4 diagrams, which in this type of diagram implies that their composition result from a binary mixture; (3) the SO4-rich end member of this mixture is gypsum, the composition of which lies by definition at (Ca+Mg)/ SO4 = 1. Thus the extrapolation of these linear trends at (Ca+Mg)/ SO4 = 1 does constrain the X/ SO4 ratios of the gypsum end member. In the revised version of the manuscript, we will better explain this approach, in particular by modifying Fig. C2 (see Figure 2 below); in its new version, Fig. 2 will feature only four panels with $87Sr/86Sr$, Sr/ SO4, Ca/ SO4, and Mg/ SO4 vs. (Ca+Mg)/

[Figure]

SO4, with only data from springs S12-S21 plotted. We hope that some additional text and a more focused figure will help the reader understand this approach.

Figure 2 (new version of Fig. C2 of the submitted manuscript): Mixing diagrams of E/SO4 (with E = Sr, Mg, Ca) and 87Sr/86Sr vs. (Ca+Mg)/SO4 used to determine the E/SO4 (and then E/Sr ratios) ratios of the gypsum end member.

- Similarly, since the sulfate to Sr ratio of the silicate endmember is estimated from the data, the finding that the d34S correlates with the fractional contribution of S from this endmember might be forced to be the case as oppose to being an entirely independent test of the mixing model. Given the large range of d34S values observed in the solid-phase data, I am somewhat surprised that a single d34S value for sulfide mineral weathering can be applied to all of the water samples.

Again, we are not sure to understand the reviewer's concern here. Fig. C3 does show an independent test of the inversion outputs: The X-axis is not constrained by any isotope ratio, while the Y-axis is "purely isotopic". The isotope composition of sulfur is indeed used to constrain the structure of the inversion model (Fig. 4d), showing that at Séchilienne it can be safely assumed that sulfides are most likely physically associated to silicate minerals, and thus that a SO4/Sr ratio can be defined for the silicate end member (and not for the carbonate end member). But this is a rather qualitative constraint on the model structure, and not a quantitative constraint on the model output themselves which would induce some "circularity" in Fig. C3. As for the variability in solid phase d34S data, we would first like to emphasize that the correlation of Fig. C3 is far from being perfect, such that some of the scatter at least could well be attributed to this type of variability. However, it should also be reminded that small rock samples are expected to be much more variable than waters (and end members determined thereby) because of the naturally integrative nature of the latter.

Text: Line by line detailed comments:

- Line 111 - I do not know what "inertial circulation" means and would appreciate a brief

definition here.

"Inertial circulation" here refers to the inertial behavior of groundwaters in aquifer. This term means that these waters exhibit a slower transit through the rocks, resulting in a slow response of the flow rate at the outlet in response to rainfall. By contrast, a "reactive system" shows a quick response. Here, the behavior of the aquifer is influenced by rainfall over long durations, showing its inertial behavior.

We will add a sentence to explain this term.

- Line 395 - I am not sure that figure 5 is a histogram. Is it not a stacked bar plot? Also, one standard deviation of the mixing model results may not be a sufficient representation of the uncertainty. With the Monte-Carlo approach that is applied, there is no reason that one of the more "rare" solutions could not be the most accurate. Focusing on the median (and results nearby) will be sensitive to the assumption of a priori distribution shapes for each end-member.

The reviewer is right, this is a stacked bar plot and we will change the text accordingly. Showing uncertainty in a stacked bar plot is challenging, but we will add whiskers to the figure to express the level of uncertainty associated to these estimates (e.g. 16th and 84th percentiles).

- Figure 7 - A more complete definition of R and Z in the figure caption would be useful.

This will be added.

Answer to Reviewer #2.

- The evidence from the rocks: Nevers and colleagues did a nice job by using the 87Sr/86Sr, $\delta 34S$ of the rocks to constrain the interpretation on dissolved species in water. However, more data on chemical/minerology composition of rocks might be necessary, e.g. the pyrite and carbonate abundances. It might be interesting to know how deep is the reaction front of pyrite (if samples from several boreholes are available). Also, whether weathering is driven by carbonic or sulfuric acid (as shown in

Fig. 5) is related to the pyrite and carbonate abundances in rock. Could gypsum be a weathering product of pyrite weathering? Since the isotope values were reported, I suspect the solid samples might be still available to make the analysis.

We indeed have additional information on the local rock composition, and some text material will be added in the text, part 2.1 "Geological setting":

"Borehole logs are available within the instability (Lajaunie et al. 2019). Within these logs, observations have shown that the rock formations below the slope are relatively unstructured, and pyrite is heterogeneously distributed therein. Rock samples along this borehole seem to have been subjected to oxidizing conditions, but no clear sulfide reaction front is present at the scale of the instability".

In addition, unpublished petrological observations on thin sections from these boreholes, as well as associated mineralogical analyses based on X-ray diffraction (XRD) have shown the presence of pyrite disseminated within the rocks, but with no particular association with calcite. XRD analyses does not show any evidence for gypsum in the sampled rocks. In support of these observations, the work of Vallet et al., 2015 showed by inverse modeling that sulphates in waters from the unstable zone (UZ) originate essentially from pyrite. Information from this work will be added as supplementary material, featuring in particular thin sections photographs and XRD spectra.

According to the saturation indices calculated for the spring waters, the saturation allowing the precipitation of the gypsum was never reached. Thus, the simplest interpretation is that the presence of the signature of gypsum in the waters is linked to the dissolution of an "external" source, rather than to a weathering product of pyrite. The possibility that gypsum is a pyrite alteration product is unlikely and the manuscript will not be modified about this topic.

- The discussion on hydrology: I like the authors' approach in section 5.2 on how this source identification may help to refine the hydrogeological model. But I feel the authors could discuss more about how the hydrological process may affect the chemistry

of different water types. For example, is it possible that outflow S10 with low elemental concentrations represents an interflow, where pyrite has already been depleted in surrounding rocks and samples from G1 and G2 represent deeper groundwater where pyrite oxidation is occurring?

As for the specific question of the reviewer regarding spring S10, we emphasize that this outflow represents a sample taken in a "stable" area, above a lowly weathered and slightly fractured basement. In contrast, G1 and G2 are located in the unstable context of the slope, where the basement is destructured. Thus, the weathering degree of rocks and minerals (pyrite weathering at the origin of sulphate concentrations in the water) will be higher at G1 and G2 with respect to S10. This explains the lower sulfate contents at the level of S10. Furthermore, interpreting S10 as an interflow and G1 and G2 as originating from deeper groundwaters does not seem to us like the simplest scenario, as topographically, outflows G1 and G2 are higher than S10. The unstable zone actually consists in a rather superficial context for water circulation (Vallet et al., 2015a), characterized by the quasi-absence of deep groundwaters.

This information will be integrated in the text and we will extend the discussion on hydrogeological pathways in the discussion section.

Specific comments:

- Line 35: I agree that silicate weathering by sulfuric acid does not directly influence atmospheric CO2, but I will argue it will reduce the potential for CO2 sequestration by silicate weathering.

Yes, we agree. We will modify the sentence to reflect this fact.

- Line 51: some references should be added to guide readers.

These references will be added: Vengeon, 1998; Meric et al., 2005; LeRoux et al., 2011; Guglielmi et al., 2002; Vallet et al., 2015; Lajaunie et al., 2019.

- Line 278: The authors showed a more complicated mass balance approach later,

then is the correlation from atmospheric input necessary here? The atmospheric input could be another endmember in the mixing model.

As replied to reviewer 1, we believe that this would overly complicate the mixing model, without specifically addressing the issue related to this correction - and rightfully raised by reviewer 1: that a larger fraction of Cl (larger than Clcrit) might be accompanied with cations such as Na. Consequently, we will not include another end member in the mixing model, but we will refine the correction done beforehand.

- Line 364: I didn't find the label (a-d) in the figure. The gray bar in Fig. 4a (left upper panel) needs explanation.

Labels a-d will be added to the figure, and some legend will be added to the grey bar in Fig. 4a ("Jurassic carbonate").

- Line 420: I love this figure. I think some quantitative results should be summarized in the abstract.

Quantitative results will be summarized in the abstract.

- Line 460: It is better to use another color for the river

The color of the river will be changed.

- Line 535, 536: These two citations are not listed in references.

Citations will be listed in references.

Fletcher, R. C., Buss, H. L., and Brantley, S. L.: A spheroidal weathering model coupling porewater chemistry to soil thicknesses during steady-state denudation, Earth Planet. Sci Lett., 244, 444-457, https://doi.org/10.1016/j.epsl.2006.01.055, 2006

Behrens R., Bouchez J., Schuessler J. A., Dultz S., Hewawasam T. and von Blanckenburg F. (2015) Mineralogical transformations set slow weathering rates in low-porosity metamorphic bedrock on mountain slopes in a tropical climate. Chem. Geol. 411,

283–298, https://doi.org/ 10.1016/j.chemgeo.2015.07.008, 2015

- Line 538: In general, I agree with the authors. But the significance of such feedback really depends on the pyrite and carbonate abundances in bedrock.

Yes, we agree. We will add "provided that enough carbonate and pyrite is present in the bedrock".

- Table A1. Are the dissolved oxygen data available? Given the importance of pyrite oxidation, such data might be interesting.

There are no data available for dissolved oxygen.

- Figure C1: the y label is unreadable.

Y label will be arranged to be readable.

Answer to the Associate Editor comment.

Specific comments:

- 14 - Here we use a combination of major element chemistry. . ..

This will be done.

- 16 – the final two sentences here are very vague – it would be better to use this space to highlight some key results (or examples of being able to do what you say)

Details will be added to these sentences. We will move to this place part of the information provided in the second paragraph of the submitted abstract.

- 20 – Using a mixing model of XXXX (details), we are able to show. . ..

This will be done.

- 21 – where does it do this – in the failure itself? In the debris it creates? It would be useful to specify here.

The creation of favorable conditions for sulfuric acid production (by pyrite oxidation) occurs mainly in the fractures. Reactive surfaces could also be created in the debris it creates but in smaller proportions.

This information will be added:

"As a consequence of the model, we are able to show that the instability creates favorable and sustained conditions within the failure, through the opening of new fractures bringing fresh and reactive surfaces allowing for the production of sulfuric acid by pyrite oxidation".

- 23 – "but" => by?

This will be done.

- 26 – change "instable zones" to "large landslide complexes"

This will be done.

- 27 – instead of "physical and chemical erosion and climate", is it clearer to say "physical and chemical erosion and their impact on the carbon cycle and global climate"

We agree, we will change this.

- 36 – and indeed when sulfuric acid mixes with natural waters containing HCO3 at neutral pH or higher – this can release CO2.

We will add this.

- 38 – is this true (that carbonates are a minor fraction)? I think Hartmann's global maps show sedimentary rocks cover âĹij65% of the earth;s surface, and I imagine that carbonates could make up a big chunk of that, especially considering interbedded carbonates and shales, and carbonate cement in siliciclastic rocks.

This is true. We will tone down this statement.

- 108 – consider splitting this sentence.

This will be done as follows: "The high degree of fracturation of the massif and its heterogeneity lead to distinct and complicated hydrological flow paths. Water pathways are characterized by different transit times related to a duel permeability behavior that is typical of fractured rock aquifers where conductive fractures play a major role in the drainage".

- Figure 1 – can you show the cross section (d) location on b or c?

This cross section is already shown on Fig. 1.c.

- 118 – can you explain briefly what the 'gallery' is – its not a term I've heard before, and other readers may not be familiar with it either

We apologize, the word "gallery" was not the correct translation of the French term. The correct translation would be "underground tunnel". We will change to:

"An underground tunnel for the production of electricity in a local hydropower plant, named "Galerie EDF", built by Electricité de France (EDF), located at the base of the slope, acts as a major westward drain for groundwater".

- 160 – leach. $H_2O$ not H20

This will be done.

- 179 – Sulfur

This will be done.

- 183 – typo

"Sulfides sulfur" will be replaced by "Sulfur contained in sulfides".

- Figure 2 – add the notations to the figure legend so the readers can quickly see the water types (e.g. what is UZ BSZ etc.,)

Notations will be added.

- Figure 4 – please add a,b,c,d labels to panels. Can carbonate weathering by sulfuric acid also be identified on part c? on part d, what does silicate end member mean for the x-axis (sulfur isotopes) – I guess pyrite? On d, what was the choice of S and Sr concentrations to make the mixing hyperbola?

Labels (a, b, c, d) will be added. For part d, yes, the silicate end member corresponds to pyrite in terms of d34S. Several values of Sr and S concentrations were tested in order to obtain a hyperbola that best fits the measured values, which correspond to a mixture between the pyrite and gypsum end members. These values were constrained by the defined pyrite and gypsum end members.

Please also note the supplement to this comment:
https://esurf.copernicus.org/preprints/esurf-2020-42/esurf-2020-42-AC1-supplement.pdf

[Figure]

[Figure]

**Fig. 1.**

**Fig. 2.**

---

## Author Response (AR1)

Dear Dr. Hilton

We thank you for handling our submission, and we thank the reviewers for the quality of their comments. The manuscript will be thoroughly revised in order to address these comments. You will find in this letter a proposal for changes that we can make to improve our manuscript. The major modifications will be as follows:

- We will include in our analysis an additional Cl source, including in our inversion scheme a scenario where that road salts (NaCl) can be an additional solute source of Cl into the waters (SC1).

- We will use a more realistic composition for the carbonate end member (in terms of Ca/Sr and Mg/Sr ratios) and include an explicit treatment of secondary carbonate precipitation in the inversion scheme based on the method introduced by Bickle et al. (2015) (SC1).

 - We will add some discussion material about bedrock composition (including noteably new data on major and trace elements on bulk rock, leacheates of the carbonate component, and residue) (SC1 and RC1)

- Further clarification will be provided on the specifics of the hydrological pathways and associated reactions such as the origin of gyspum-sourced solutes in waters, and on the oxidative weathering of the pyrite.

We believe that these modifications will significantly improve our manuscript, and we hope make it ultimately suitable for publication in E-surf.

Please find below the reviewer comments (in black) our answers from the interactive discussion (in blue) and the line of the change -if any- (in green).

**Answer to Reviewer #1.**

Firstly, I am slightly confused about the rainwater correction that the authors employ in this study. I get the concept of using chloride critical values, but I am slightly concerned that the assumption of spatially and temporally uniform evapotranspiration is not appropriate for this specific study site. The data presented in Figure 3 show a wide range of chloride concentrations, including some below the chloride critical value. How does such a range arise if all samples are presumed to originate from rainwater with a constant concentration of Cl that undergoes the same amount of evapotranspiration? It is also well known that rainwater element to Cl ratios can be significantly different than seawater element to Cl ratios especially for sulfate (e.g., Stallard & Edmond 1981), which is an important ion in the present study. The authors also acknowledge that some Cl remains after they perform the correction, which implies that there is an additional solute source. However, this additional source is not included in their subsequent mixing model, which is concerning to me as this might be a source of error.

We agree with the reviewer that the way we were quantifying the contribution of non-rock sources to the spring cationic load can be improved. Indeed, some springs show a fairly high Cl concentration (> 100 µmol/L). In particular, an additional Cl source can be attributed road salt inputs (NaCl). The additional amount of Na associated with road-salt derived Cl would also account for some of the high Na contents measured in springs draining the MSZ.

Therefore, in the revised manuscript we will include a new scenario in the inversion scheme, in addition to the previous scenario where Cl was just contributed by the atmosphere (subtraction of the critical Cl*): in this new scenario all Cl is associated with Na and thus it is necessary to subtract from the Na budget the whole Cl amount for each spring. This will obviously translate in additional uncertainty in the output parameters ("R" and "Z" parameters).

Regarding the reviewer's suggestion that we should rather "includes atmospheric deposition as a solute source (as oppose to performing a fixed correction beforehand)", we emphasize that this would not necessarily solve the issue of non-cyclic sources of Cl - we would still need to make a strong assumption about the X/Cl ratios of these sources. This is why we prefer to keep this "fixed correction beforehand", and to modify it to address the - fully valid - concern of the reviewer. We also note that the new scenario will place an "upper bound" on the correction that should be performed on Na concentrations, as some Cl could still be derived from other sources than NaCl (on this topic, we have noticed during examination of the relationships between Cl concentration and other variables that some Cl-rich samples were also K-rich, which might indicate that another source of Cl exists, such as a KCl in road salts). As a consequence, we believe that the combination of our previous scenario and of this new one will constitute a sort of "sensitivity analysis" allowing us to explore the whole range of possibilities for the impact of Cl sources on the cationic load of the springs at Séchilienne.

The 3-component mixing model the authors utilize is based on Na, Sr, and Sr isotopic ratios as these are expected to behave more conservatively. I agree with the authors about this and trust that their mixing model yields reasonable estimates for the fraction of Sr sourced from each endmember (with the caveat that they ignore the source of excess Cl as mentioned above). However, the extrapolation of the mixing results for Sr to Ca and Mg effectively assumes conservative mixing (Equation 8), which negates the whole motivation for performing the mixing calculations with Na and Sr alone.

This is a fair point too. To address this comment, we will significantly modify our inversion scheme by:
   (1) using the actual composition of local carbonate rocks to convert Sr fractions into Mg and Ca fractions (based on new major and trace element data obtained on HCl and acetic acid leachates of local carbonate bedrock samples);
   (2) including an explicit, quantitative treatment for the precipitation of secondary carbonates based on the method of Bickle et al. (2015). To that effect, we will determine the local mixing array between the different rock type based on new major and trace element data obtained on local carbonate and silicate bedrock samples (acetic acid and HCl leachates, residues, and bulk rocks).

These major modifications will release a strong assumption of the previously used model - namely that the composition of the carbonate end member was an "adjustment variable".

[Figure]

*Figure: local mixing array between the different rock type based on new major and trace element data obtained on local carbonate and silicate bedrock samples (including quantitative treatment for the precipitation of secondary carbonates based on the method of Bickle et al. (2015))*

The manner in which some of the end-member ratios are defined is concerning to me as there may be some statistical issues and/or circular logic applied. For some endmembers, linear regressions of solute data are extrapolated to yield constraints on their characteristic elemental ratios. For example, the text on line 638 describes regressing element to sulfate ratios versus Ca+Mg to sulfate ratios. Since sulfate is the denominator for both ratios, there is potential for spurious correlation that might affect the best fit line and thus the calculated end-member ratio.

We are not sure to understand fully the reviewer's concern.
As for the extrapolation of "linear regressions of solute data" generally speaking, we believe that this is a pretty conventional method and it yields robust results when correctly done (as was done by the reviewer himself and his co-authors in previous studies; see the determination of the rain end member in Torres et al., GCA, 2015).
As for the specific case of the gypsum end member in our work, we do not see where the issue is. Our rationale is as follows: (1) some springs are only very marginally affected by silicate inputs (hence, **in the case of Séchilienne** by sulfide inputs; see Fig. 4d of the previous version of the manuscript) as shown by their [87]Sr/[86]Sr ratios < 0.710 (these are springs S12-S21); (2) data from these springs do form linear trends in X/ $SO_4$ vs. (Ca+Mg)/$SO_4$ diagrams, which in this type of diagram implies that their composition result from a binary mixture; (3)

the SO4-rich end member of this mixture is gypsum, the composition of which lies by definition at $(Ca+Mg)/SO_4 = 1$. Thus the extrapolation of these linear trends at $(Ca+Mg)/SO_4 = 1$ do constrain the $X/SO_4$ ratios of the gypsum end member.

In the revised version of the manuscript, we will better explain this approach, in particular by modifying Fig. C2; in its new version, Fig. C2 will feature only four panels with $87Sr/86Sr$, $Sr/SO_4$, $Ca/SO_4$, and $Mg/SO_4$ vs. $(Ca+Mg)/SO_4$, with only data from springs S12-S21 plotted. We hope that some additional text and a more focused figure will help the reader understand this approach.

[Figure]

*Figure C2 (new version): Mixing diagrams of E/SO₄ vs. (Ca+Mg)/SO₄ (with E = Sr, Mg, Ca, Na and $^{87}Sr/^{86}Sr$) used to determine the E/Sr rations of the gypsum endmember.*

Similarly, since the sulfate to Sr ratio of the silicate endmember is estimated from the data, the finding that the d34S correlates with the fractional contribution of S from this endmember might be forced to be the case as oppose to being an entirely independent test of the mixing model. Given the large range of d34S values observed in the solid-phase data, I am somewhat surprised that a single d34S value for sulfide mineral weathering can be applied to all of the water samples.

Again, we are not sure to understand the reviewer's concern here. Fig. C3 does show an independent test of the inversion outputs: The X-axis is not constrained by any isotope ratio, while the Y-axis is "purely isotopic". The isotope composition of sulfur is indeed used to constrain the structure of the inversion model (Fig. 4d), showing that at Séchilienne it can be safely assumed that sulfides are most likely physically associated to silicate minerals, and thus that the $SO_4$/Sr ratio can be defined for the silicate end member (and not for the carbonate end member). But this is rather qualitative constraint on the model structure, and not a quantitative constraint on the model output.
As for the variability in solid phase d34S data, we would first like to emphasize that the correlation of Fig. C3 is far from being perfect, and some of the scatter at least could be attributed to such variability. However, it should also be reminded that small rock samples are expected to be much more variable than waters (and end members determined thereby) because of the naturally integrative nature of the latter.

**Text: Line by line detailed comments:**

Line 111 - I do not know what "inertial circulation" means and would appreciate a brief definition here.

"Inertial circulation" here refers to the inertial behavior of groundwaters in aquifer. This term means that these waters exhibit a slower transit through the rocks, resulting in a slow response of the flowrate at the outlet in response to a rainfall. By contrast, a "reactive system" shows a very quick response. Here, the behavior of the aquifer is influenced by rainfalls over long durations, showing its inertial behavior.

We will add a sentence to explain this term.

Done – Line 122 to 124

Line 395 - I am not sure that figure 5 is a histogram. Is it not a stacked bar plot? Also, one standard deviation of the mixing model results may not be a sufficient representation of the uncertainty. With the Monte-Carlo approach that is applied, there is no reason that one of the more "rare" solutions could not be the most accurate. Focusing on the median (and results nearby) will be sensitive to the assumption of a priori distribution shapes for each end-member.

The reviewer is right, this is stacked bar plot and we will change the text accordingly. Showing uncertainty in a stacked bar plot is challenging, but we will add whiskers to the figure to express the level of uncertainty associated to these estimates (e.g. 16th and 84th percentiles).

Done

Figure 7 - A more complete definition of R and Z in the figure caption would be useful.

This will be added.

**Answer to Reviewer #2.**

The evidence from the rocks: Nevers and colleagues did a nice job by using the 87Sr/86Sr, δ34S of the rocks to constrain the interpretation on dissolved species in water. However, more data on chemical/minerology composition of rocks might be necessary, e.g. the pyrite and carbonate abundances. It might be interesting to know how deep is the reaction front of pyrite (if samples from several boreholes are available). Also, whether weathering is driven by carbonic or sulfuric acid (as shown in Fig. 5) is related to the pyrite and carbonate abundances in rock. Could gypsum be a weathering product of pyrite weathering? Since the isotope values were reported, I suspect the solid samples might be still available to make the analysis.

We can indeed add information on the local rock composition.

A sentence can be added in the text, part 2.1 Geological stetting:

"Borehole logs are available within the instability, (Lajaunie et al. 2019). Within these logs, observations have shown that the rock formations are relatively unstructured, and pyrite is heterogeneously distributed. Rock samples along this borehole seem to have been subjected to oxidizing conditions, but no clear "sulfide reaction front" is present at the scale of the instability".

Done – Line 103

In addition, unpublished petrological observations on thin sections from these boreholes, as well as associated mineralogical analyses based on X-ray diffraction (XRD) have shown the presence of pyrite disseminated within the rocks, but with no particular association with calcite. XRD analyses does not show any evidence for gypsum in the sampled rocks. In support of these observations, the work of Vallet et al., 2015 showed by inverse modeling that sulphates in waters from the unstable zone (UZ) originate essentially from pyrite. Information from this work can be added in appendices with thin sections photos and XRD spectrums.

Done

According to the saturation indices calculated for the spring waters, the saturation allowing the precipitation of the gypsum was never reached. Thus, the simplest interpretation is that the presence of the signature of gypsum in the waters is linked to the dissolution of an "external" source, rather than to a weathering product of pyrite. The possibility that gypsum is a pyrite alteration product is unlikely and the paper won't be modified about this topic.

The discussion on hydrology: I like the authors' approach in section 5.2 on how this source identification may help to refine the hydrogeological model. But I feel the authors could discuss more about how the hydrological process may affect the chemistry of different water types. For example, is it possible that outflow S10 with low elemental concentrations represents an interflow, where pyrite has already been depleted in surrounding rocks and samples from G1 and G2 represent deeper groundwater where pyrite oxidation is occurring?

As for the specific question of the reviewer regarding spring S10, we emphasize that this outflow represents a sample taken in a "stable" area, above a lowly weathered and slightly fractured basement. In contrast, G1 and G2 are located in the unstable context of the slope, where the basement is destructured. Thus, the weathering degree of rocks and minerals (pyrite weathering at the origin of sulphate concentrations in the water) will be higher at G1 and G2 with respect to S10. This explains the lower sulphate contents at the level of S10. Furthermore, interpreting S10 as an interflow and G1 and G2 as originating from deeper groundwaters does not seem to us like the simplest scenario, as topographically, outflows G1 and G2 are higher than S10. The unstable zone actually consists in a rather superficial context for water circulation (Vallet et al., 2015a), characterized by the quasi-absence of deep groundwaters.

These information will be integrated into the text and we will extend the discussion on hydrogeology.

Done – section 5.2

**Specific comments:**

Line 35: I agree that silicate weathering by sulfuric acid does not directly influence atmospheric CO2, but I will argue it will reduce the potential for CO2 sequestration by silicate weathering.

Yes, we agree. We will modify the sentence to reflect this.

Done – Line 36 to 38

Line 51: some references should be added to guide readers.

These references will be added: Vengeon, 1998; Meric et al., 2005; LeRoux et al., 2011; Guglielmi et al., 2002; Vallet et al., 2015; Lajaunie et al., 2019.

Done – Line 54-55

Line 278: The authors showed a more complicated mass balance approach later, then is the correlation from atmospheric input necessary here? The atmospheric input could be another endmember in the mixing model.

As replied to reviewer 1, we believe that this would overly complicate the mixing model, without specifically addressing the issue related to this correction - and rightfully raised by reviewer 1: that a larger fraction of Cl (larger than Clcrit) might be accompanied with cations such as Na. Consequently, we will not include another end member in the mixing model, but we will refine the correction done beforehand.

Line 364: I didn't find the label (a-d) in the figure. The gray bar in Fig. 4a (left upper panel) needs explanation.

Labels a-d will be added to the figure, and some legend will be added to the grey bar in Fig. 4a ("Jurassic carbonate").

Done

Line 420: I love this figure. I think some quantitative results should be summarized in the abstract.

Quantitative results will be summarized in the abstract.

Done

Line 460: It is better to use another color for the river

The color of the river will be changed.

Done

Line 535, 536: These two citations are not listed in references.

Citations will be listed in references.

Fletcher, R. C., Buss, H. L., and Brantley, S. L.: A spheroidal weathering model coupling porewater chemistry to soil thicknesses during steady-state denudation, Earth Planet. Sci Lett., 244, 444-457, https://doi.org/10.1016/j.epsl.2006.01.055, 2006

Behrens R., Bouchez J., Schuessler J. A., Dultz S., Hewawasam T. and von Blanckenburg F. (2015) Mineralogical transformations set slow weathering rates in low-porosity metamorphic bedrock on mountain slopes in a tropical climate. *Chem. Geol.* **411**, 283–298, https://doi.org/10.1016/j.chemgeo.2015.07.008, 2015

Done – Line 1424 and 1509

Line 538: In general, I agree with the authors. But the significance of such feedback really depends on the pyrite and carbonate abundances in bedrock.

Yes, we agree. We will add "provided that enough carbonate and pyrite is present in the bedrock".

Done – Line 966

Table A1. Are the dissolved oxygen data available? Given the importance of pyrite oxidation, such data might be interesting.

There are no data available for dissolved oxygen.

Figure C1: the y label is unreadable.

Y label will be arranged to be readable.

Done

**Answer to the Associate Editor comment.**

**Specific comments:**

14 - Here we use a combination of major element chemistry. . ..

This will be done.

Done – Line 14

16 – the final two sentences here are very vague – it would be better to use this space to highlight some key results (or examples of being able to do what you say)

Details will be added to the sentences: we are going to bring up part of the information provided in the second part of the current abstract with some more information and discussions.

Done

20 – Using a mixing model of XXXX (details), we are able to show. . ..

This will be done.

Done – Line 20

21 – where does it do this – in the failure itself? In the debris it creates? It would be useful to specify here.

The creation of favorable conditions for sulfuric acid production (by pyrite oxidation) occurs mainly in the fractures. Reactive surfaces could also be created in the debris it creates but in smaller proportions.

This information will be added:

"As a consequence of the model, we are able to show that the instability creates favorable and sustained conditions within the failure, through the opening of new fractures bringing fresh and reactive surfaces allowing for the production of sulfuric acid by pyrite oxidation".

Done – Line 20

23 – "but" => by?

This will be done.

Done – Line 24

26 – change "instable zones" to "large landslide complexes"

This will be done.

Done – Line 27

27 – instead of "physical and chemical erosion and climate", is it clearer to say "physical and chemical erosion and their impact on the carbon cycle and global climate"

We agree, we will change this.

Done – Line 28

36 – and indeed when sulfuric acid mixes with natural waters containing HCO3 at neutral pH or higher – this can release CO2.

We will add this.

Done – Line 38

38 – is this true (that carbonates are a minor fraction)? I think Hartmann's global maps show sedimentary rocks cover ~65% of the earth;s surface, and I imagine that carbonates could make up a big chunk of that, especially considering interbedded carbonates and shales, and carbonate cement in siliciclastic rocks.

This is true. We will tone down this statement.

Done – Line 41

108 – consider splitting this sentence.

This will be done as follows:
"The high degree of fracturation of the massif and its heterogeneity lead to distinct and complicated hydrological flow paths. Water pathways are characterized by different transit times related to a duel permeability behavior that is typical of fractured rock aquifers where conductive fractures play a major role in the drainage".

Done – Line 119 to 124

Figure 1 – can you show the cross section (d) location on b or c?

This cross section is already shown on Fig. 1.c.

118 – can you explain briefly what the 'gallery' is – its not a term I've heard before, and other readers may not be familiar with it either

Some explanation will be added to the sentence.
The word gallery is not the correct translation of the French term. The correct translation would be underground tunnel.

"An underground tunnel (water pipe used for the production of electricity in Hydropower plant), "named EDF Gallery", built by Electricité de France (EDF), located at the base of the slope, acts as a major westward drain for groundwater".

Changes done in text

160 – leach. H2O not H20

This will be done.

Done – Line 242

179 – Sulfur

This will be done.

Done – Line 262

183 – typo

"Sulfides sulfur" will be replaced by "Sulfur contained in sulfides".

Done – Line 283

Figure 2 – add the notations to the figure legend so the readers can quickly see the water types (e.g. what is UZ BSZ etc.,)

Notations will be added.

Done

Figure 4 – please add a,b,c,d labels to panels. Can carbonate weathering by sulfuric acid also be identified on part c? on part d, what does silicate end member mean for the x-axis (sulfur isotopes) – I guess pyrite? On d, what was the choice of S and Sr concentrations to make the mixing hyperbola?

Labels (a, b, c, d) will be added. For part d, yes, the silicate end member corresponds to pyrite in terms of d34S.

Done

Several theoretical values of Sr and S concentrations were tested in order to obtain a hyperbola that best translates the measured values, which correspond to a mixture between the pyrite and gypsum poles. These values were framed by the defined pyrite and gypsum poles.

---

## Author Response (AR2)

Dear Associate Editor,

Thank you for your reply and your suggestions on our manuscript. All requested changes and suggestions were made. The main text has been corrected where necessary. We strongly reduced Appendix D, and in particular we remove former Figure D3 and its legend as it was not used, and. we significantly shortened the legend of figure D4 of the appendix.

To further reduce the size of the Appendix, we moved former Appendix C and D as separate documents in the supplementary materials and removed from the main body of the manuscript. Table of results have also been displaced to supplementary materials and the numbering of the tables has been changed as follow: Tables D1, D2 and D3 are changed to Table S1, S2 and S3. Finally, datasets originally presented as Tables A1, A2 and B1 in appendix A and B of the manuscript have been removed and are now available as a data repository at 10.5281/zenodo.4606732.

Manuscript track-changes and supplementary materials track-changes are combined in one .pdf file in order to be submitted.

We hope that these minor changes will correspond to your expectations.

On behalf of the co-authors,

Pierre Nevers